# 3d $\mathcal{N} = 4$ Bootstrap and Mirror Symmetry

**Chi-Ming Chang[1,2], Martin Fluder[3,4,7], Ying-Hsuan Lin[4,5],**
**Shu-Heng Shao[6] and Yifan Wang[7,8,9]**

**1** Yau Mathematical Sciences Center, Tsinghua University, Beijing, 100084, China
**2** Center for Quantum Mathematics and Physics (QMAP),
University of California, Davis, CA 95616, USA
**3** Kavli Institute for the Physics and Mathematics of the Universe (WPI),
University of Tokyo, Kashiwa 277-8583, Japan
**4** Walter Burke Institute for Theoretical Physics, California Institute of Technology,
Pasadena, CA 91125, USA
**5** Hsinchu County Environmental Protection Bureau, Hsinchu, Taiwan
**6** School of Natural Sciences, Institute for Advanced Study, Princeton, NJ 08540, USA
**7** Joseph Henry Laboratories, Princeton University, Princeton, NJ 08544, USA
**8** Center of Mathematical Sciences and Applications,
Harvard University, Cambridge, MA 02138, USA
**9** Jefferson Physical Laboratory, Harvard University, Cambridge, MA 02138, USA

## Abstract

We investigate the non-BPS realm of 3d $\mathcal{N} = 4$ superconformal field theory by uniting the non-perturbative methods of the conformal bootstrap and supersymmetric localization, and utilizing special features of 3d $\mathcal{N} = 4$ theories such as mirror symmetry and a protected sector described by topological quantum mechanics (TQM). Supersymmetric localization allows for the exact determination of the conformal and flavor central charges, and the latter can be fed into the mini-bootstrap of the TQM to solve for a subset of the OPE data. We examine the implications of the $\mathbb{Z}_2$ mirror action for the SCFT single- and mixed-branch crossing equations for the moment map operators, and apply numerical bootstrap to obtain universal constraints on OPE data for given flavor symmetry groups. A key ingredient in applying the bootstrap analysis is the determination of the mixed-branch superconformal blocks. Among other results, we show that the simplest known self-mirror theory with $SU(2) \times SU(2)$ flavor symmetry saturates our bootstrap bounds, which allows us to extract the non-BPS data and examine the self-mirror $\mathbb{Z}_2$ symmetry thereof.

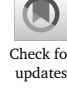

# 1  Introduction

Quantum field theory (QFT) in three spacetime dimensions has proven to be a promising arena for studying strongly coupled physics. It shares close resemblance to four-dimensional QFT in providing a rich array of non-perturbative phenomena such as confinement, chiral symmetry breaking, and mass gaps. 3d QFT is also particularly intriguing in its own right due to the presence of Chern-Simons couplings, fractional (anyonic) statistics, monopole operators, and highly nontrivial duality mechanisms such as the bosonization dualities (analogous to 2d), all of which play important roles in our understanding of the quantum phase transitions of gauge theories and condensed matter systems. A particularly interesting and subtle case is when the transition is second order and thus mediated by a conformal field theory (CFT). Knowledge of the operator spectrum and correlation functions in the CFT is crucial to completing the phase diagram.

There is a rich family of 3d CFTs constructed as the infrared fixed points of Chern-Simons matter theories. The existence of such fixed points is supported by substantial evidence including a non-renormalization theorem for the Chern-Simons level [1,2], perturbative analyses of the beta function for the matter couplings [3–5], exact computations of three-point functions and thermal free energies in the 't Hooft limit [6–10], the bulk gravity or higher spin duals under the AdS/CFT correspondence [5,6,11–14], as well as intricate duality webs and the 't Hooft anomaly matching thereof [6–8,15–18]. Their non-perturbative dynamics are explored to some extent by approaches including AdS/CFT, bosonization dualities, and Schwinger-Dyson equations.

Progress on the non-perturbative dynamics of CFT has been made by two powerful tools – the conformal bootstrap and supersymmetric localization. The conformal bootstrap method has been successful in solving CFT, including the famous 3d Ising CFT that describes critical phase transitions for water and magnets [19]. The inclusion of supersymmetry and the method of supersymmetric localization [20,21] have made analytic computations possible in superconformal field theory (SCFT). The unison of these two tools have proven powerful in [22–28].

In this paper, we apply the conformal bootstrap and supersymmetric localization to the specific context of 3d SCFT with $\mathcal{N} = 4$ supersymmetry. 3d $\mathcal{N} = 4$ SCFT boasts special properties such as mirror symmetry [29,30] and an exactly solvable subsector of topological quantum mechanics (TQM) [22,31–34] that present interesting interplay with bootstrap and localization. We elaborate on these special properties after briefly discussing some generalities.

Most known 3d $\mathcal{N} = 4$ SCFTs arise as the low energy descriptions of quiver gauge theories, and more generally of M2 branes probing transverse singular geometries in M-theory [12,13,35–43]. These theories typically have a vacuum moduli space that includes some combination of a Coulomb branch, a Higgs branch, and possibly mixed branches. In terms of the CFT operators, they are parametrized by the vevs of half-BPS operators subject to chiral ring relations. Some of these operators, such as monopoles, are of the disorder type: they are non-perturbative in nature and their correlation functions are inaccessible by traditional methods. Fortunately, certain integrated correlators of the current multiplets and stress-tensor multiplets can be obtained from supersymmetric localization on $S^3$ with mass and squashing deformations, and in particular encode the flavor and conformal central charges of the CFT.

The flavor symmetry of 3d $\mathcal{N} = 4$ SCFTs takes a product form $G_C \times G_H$, where the factors $G_C$ and $G_H$ are realized by different types of flavor current multiplets, whose bottom components are charged under the $SU(2)_C$ and $SU(2)_H$ R-symmetry, respectively (and uncharged under the other).[1] We study the single- and mixed-branch four-point functions of the moment map op-

---

[1]In this paper, we will not be concerned with the global properties of groups which do not affect the correlation functions of local operators.

erators (scalar primaries in the flavor current multiplets for $G_C$ and $G_H$), exploiting constraints from superconformal symmetry and crossing using the numerical bootstrap methods.[2] This is made possible by first determining the mixed-branch 3d $\mathcal{N} = 4$ superconformal blocks (the single-branch blocks were determined in [25, 44, 45]). By comparison with certain OPE data computable by supersymmetric localization, we observe that all known candidate theories are consistent with the numerical bounds. Furthermore, in some cases, such as $G_C = G_H = SU(2)$, the bound (see Figure 3) is saturated by a known theory, namely the SQED with 2 flavors, and we use the extremal functional method to extract the more general, non-BPS spectrum, that appears in its moment map OPE.

**Mirror symmetry**

Mirror symmetry in 3d [29, 30] bears close relation to 4d S-duality and 2d mirror symmetry. 3d mirror symmetry is an infrared duality: two UV gauge theory descriptions flow to the same conformal fixed point. In particular, the notions of Coulomb and Higgs branches swap between the two descriptions. The Coulomb branch is highly quantum as it embodies the complicated gauge theory dynamics that involves monopoles, yet the Higgs branch is protected against such effects by supersymmetric non-renormalization theorems [29]. The key insight of mirror symmetry is that the strongly coupled Coulomb branch in one UV description has an alternative simple description as the protected Higgs branch of another UV description, which leads to highly-nontrivial predictions on observables in the Coulomb branch of the first description. Mirror symmetry also has important consequences for non-supersymmetric dualities in 3d. By turning on deformations that break the supersymmetry, it gives nontrivial evidence for the 3d bosonization duality mentioned above [46, 47], the zero mass case of which was otherwise unsettled from just anomaly matching arguments.[3]

However, despite much progress in understanding mirror symmetry at the level of BPS data, little is known in the non-BPS realm, which is crucial for analyzing the non-supersymmetric bosonization duality from deformations. We make progress towards analyzing the non-BPS aspect of mirror symmetry by applying the bootstrap method to the simplest interacting 3d $\mathcal{N} = 4$ SCFT, namely the SQED with 2 flavors, given by the IR fixed point of SQED with two hypermultiplets of unit charge. In this particular case, the UV description is self-mirror in the sense that the Higgs and Coulomb branches are simply exchanged within a *single* UV description under the mirror map, and the theory enjoys an additional $\mathbb{Z}_2$ global symmetry that simultaneously acts on the $\mathcal{N} = 4$ superconformal algebra as an outer-automorphism and exchanges the two factors of $G_C \times G_H$. As usual, such symmetries will leave signatures on the fixed point OPE data which we explore by analyzing a system of mixed correlators involving both Coulomb and Higgs branch moment map operators. As we will see, this theory saturates the bootstrap bound, and by extracting the non-BPS OPE data we are able to provide nontrivial evidence for the (self-)mirror symmetry.

**Topological quantum mechanics**

Moreover, 3d $\mathcal{N} = 4$ SCFT is also special in that it contains a one-dimensional protected subsector, known as the topological quantum mechanics (TQM). In general, there are two TQMs, one associated with the Coulomb branch, and the other with the Higgs branch [22, 31–34].

---

[2]Due to supersymmetry, the four-point functions of the moment map operators contain all the dynamical information of the four-point functions of the operators in the flavor current multiplets.

[3]In this case, starting from the mirror symmetry between a free hypermultiplet and the $\mathcal{N} = 4$ SQED with one charged hypermultiplet, the authors of [46] considered supersymmetry-breaking deformations to deduce the non-supersymmetric bosonization duality between the Dirac fermion and the 3d scalar QED with a level-1 Chern-Simons term.

The operator algebra in the TQM is an associative algebra given by a non-commutative deformation of the Higgs or Coulomb branch chiral ring. The OPE data in the TQM can be solved or constrained using the conformal bootstrap, *i.e.* imposing the associativity and unitarity of the operator algebra, which is dubbed the mini-bootstrap because it is a closed subsystem of the full bootstrap equations. In particular, the single-branch four-point function of (twisted) moment map operators restricted to a line reduces to a four-point function in the TQM.[4] As we will see in Section 4, when the flavor symmetry group $G_C$ or $G_H$ is simple, the mini-bootstrap constrains the OPE data in this four-point function down to a two-dimensional convex quadrilateral parametrized by the flavor central charge and another OPE coefficient. The OPE data in the TQM can also be computed by supersymmetric localization [32–34].

**Brief summary of the results**

We briefly summarize the main results of this paper.

- We compute the conformal and flavor central charges for several classes of 3d $\mathcal{N} = 4$ SCFTs, by taking derivatives with respect to the mass and squashing parameters of the deformed $S^3$ free energy. The results are (2.51) and (2.52) for the SQED, (2.53) and (2.54) for the SQCD, (A.32) and (A.59) for the $T[SU(3)]$, (2.55) for the $T_3$, (A.46) and (A.65) for the $II_k(3)$ Chern-Simons matter theory. Some of these theories are dubbed as the minimal 3d $\mathcal{N} = 4$ SCFTs with flavor symmetry $G$ and their the central charges are summarized in Table 1.

- We compute the superconformal blocks for the four-point function of two Coulomb and two Higgs branch moment map operators.[5] The results for the $s$-channel superconformal blocks are in (3.24), (3.25) and (3.4.1), and the $t$-channel superconformal blocks can be found in (3.28), (3.29), (3.30) and (3.31).

- We explore the analytic bootstrap constraints in the protected topological quantum mechanics sector of the SCFT to solve or constrain the flavor central charge and certain OPE coefficients. The results are summarized in Table 5 and Figure 2. The minimal nilpotent theories sit at the kinks, and the maximal nilpotent theories sit at the boundaries of the allowed regions (see Section 2.3 and 2.4 for definitions of these theories).

- Finally, we employ the numerical bootstrap technology to map out the space of consistent unitary 3d $\mathcal{N} = 4$ SCFTs, for several flavor symmetry groups and assuming or not assuming the minimal/maximal nilpotent orbit condition on the protected operators. The key results are in Table 7, Figures 3 and 5, and we find free theories and $T[SU(2)]$ to sit at the boundary of consistency. We also present a diagnosis on bootstrap data for the self-mirror property, and explicitly verify it in $T[SU(2)]$.

**Organization of the paper**

We start by reviewing the notion of conformal and flavor central charges in 3d CFT in Section 2, and determine their values for an array of 3d $\mathcal{N} = 4$ SCFTs that realize minimal and maximal nilpotent orbits as their Higgs branch moduli spaces. Such data will later be compared with the numerical bootstrap bounds. In Section 3, we review the 3d $\mathcal{N} = 4$ superconformal algebra and its representations, and determine the superconformal blocks when the external operators are either Higgs or Coulomb branch chiral primaries. In Section 4, we present analytic

---

[4]The four-point function only involves one type of moment map operators, Coulomb or Higgs.

[5]The superconformal blocks for the four-point function of four Coulomb or four Higgs branch moment map operators were computed in [25, 44].

solutions to the mini-bootstrap problem for the protected sector of 3d $\mathcal{N} = 4$ SCFTs, and explain the relation to the deformation quantization of the Coulomb or Higgs branch. Equipped with the exact protected OPE data and the superconformal blocks, we setup and implement the bootstrap analysis in Section 5 to extract bounds on the non-BPS spectrum. We end by a summary of the main results and a discussion of future directions in Section 6. In Appendix A, we present the details of the computations of the conformal and flavor central charges using supersymmetric localization. In Appendix B, we derive the superconformal Casimir equations used for deriving the superconformal blocks. In Appendix C, we state our conventions for the Clebsch-Gordan coefficients and the crossing matrices (6j symbols). Finally, in Appendix D, we summarize the crossing equations rewritten in matrix form.

## 2 CFT central charges

For any CFT in $d$ dimensions, the two-point function of the stress tensor $T_{\mu\nu}$ takes the form

$$\left\langle T_{\mu\nu}(x) T_{\sigma\rho}(0) \right\rangle = \frac{C_T}{V_{\widehat{S}^{d-1}}^2} \frac{\mathcal{I}_{\mu\nu,\sigma\rho}(x)}{x^{2d}}, \tag{2.1}$$

where $V_{\widehat{S}^{d-1}} = 2\pi^{d/2}/\Gamma(d/2)$ is the volume of a $(d-1)$-dimensional unit sphere. Similarly, if the CFT has global symmetry $G$, the two-point functions of the conserved currents $J_\mu^a$ are given by

$$\left\langle J_\mu^a(x) J_\nu^b(0) \right\rangle = \frac{C_J}{V_{\widehat{S}^{d-1}}^2} \delta^{ab} \frac{I_{\mu\nu}(x)}{x^{2(d-1)}}. \tag{2.2}$$

The tensor structures $\mathcal{I}_{\mu\nu,\sigma\rho}(x)$ and $I_{\mu\nu}(x)$ in (2.1) and (2.2) are completely fixed by conformal symmetry

$$\begin{aligned}
\mathcal{I}_{\mu\nu,\sigma\rho}(x) &= \frac{1}{2}\left[ I_{\mu\sigma}(x) I_{\nu\rho}(x) + I_{\mu\rho}(x) I_{\nu\sigma}(x) \right] - \frac{1}{d} \delta_{\mu\nu} \delta_{\sigma\rho}, \\
I_{\mu\nu}(x) &= \delta_{\mu\nu} - 2\frac{x_\mu x_\nu}{x^2}.
\end{aligned} \tag{2.3}$$

The coefficients $C_T$ and $C_J$, with appropriate normalization, capture *dynamical* information about the particular CFT. We refer to them as the conformal and flavor central charges, respectively. We canonically normalize the stress tensor $T_{\mu\nu}$ by the Ward identity. For symmetry currents $J_\mu^a$ associated to a simple Lie group $G$, we adopt the normalization

$$J_\mu^a(x) J_\nu^b(0) \supset \frac{1}{V_{\widehat{S}^{d-1}}} i f^{abc} \frac{x_\mu}{|x|^d} J_\nu^c(0), \tag{2.4}$$

with $f^{abc}$ totally antisymmetric, purely imaginary and satisfying

$$\frac{1}{2h^\vee} f^{aed} f^{bde} = \delta^{ab}, \tag{2.5}$$

where $h^\vee$ denotes the dual Coxeter number of $G$.

For the convenience of the reader, we record the values of $C_T$ and $C_J$ for free fields in $d = 3$; namely our convention is such that $C_T = \frac{3}{2}$ for either a real massless scalar or a Majorana fermion. Thus, a free $\mathcal{N} = 4$ hypermultiplet has conformal central charge

$$C_T(\text{hyper}) = 8 \times \frac{3}{2} = 12. \tag{2.6}$$

The flavor central charge $C_J$ for the $SU(2)$ flavor symmetry of a free $\mathcal{N} = 4$ hypermultiplet can be worked out explicitly from the OPE (see the next section), and takes the value

$$C_J^{SU(2)}(\text{hyper}) = 4. \tag{2.7}$$

Here and in the following, we denote the flavor central charge with respect to the symmetry group $G$ of a SCFT $\mathcal{T}$ by $C_J^G(\mathcal{T})$.

We now outline the derivation of a formula that allows us to explicitly compute the conformal and flavor central charges for 3d $\mathcal{N} = 4$ superconformal field theories. Namely, on the one hand, we relate the conformal central charge, $C_T$, of a given theory to its metric deformed (squashed) $S^3$ partition function. On the other hand, we derive a precise correspondence between the flavor central charge $C_J$ of a theory and its mass-deformed $S^3$ partition function. Such formulae were originally derived in [48,49] for 3d $\mathcal{N} = 2$ SCFTs, and we re-derive them here in order to precisely connect with our choice of $\mathcal{N} = 4$ convention. By means of supersymmetric localization [20,21], these formulae allow for the explicit evaluation of the central charges in a large class of theories with (UV) Lagrangian descriptions. We refer to Appendix A for the required ingredients for the localization computation as well as some examples.

## 2.1 $C_T$ from squashed $S^3$ partition function

The 3d $\mathcal{N} = 4$ stress tensor multiplet $\mathcal{A}[0]_1^{(0;0)}$ consists of conformal primaries [50,51]

$$
\begin{array}{ccccccccc}
[0]_1^{(0;0)} & \rightarrow & [1]_{3/2}^{(1;1)} & \rightarrow & [2]_2^{(2;0)} \oplus [2]_2^{(0;2)} \oplus [0]_2^{(0;0)} & \rightarrow & [3]_{5/2}^{(1;1)} & \rightarrow & [4]_3^{(0,0)} \\
\Phi & \rightarrow & \Psi_\alpha^{A\dot{A}} & \rightarrow & j_\mu^{AB} \oplus j_\mu^{\dot{A}\dot{B}} \oplus U & \rightarrow & S_{\mu\alpha}^{A\dot{A}} & \rightarrow & T_{\mu\nu},
\end{array}
\tag{2.8}
$$

where each entry, $[2\ell]_\Delta^{(R_{\mathrm{H}};R_{\mathrm{C}})}$, labels the representation content of the corresponding conformal primary under the bosonic subgroup of $OSp(4|4)$; $\ell$ and $\Delta$ denote the spacetime spin and scaling dimension, while $R_{\mathrm{H}}$ and $R_{\mathrm{C}}$ label the $SU(2)_{\mathrm{H}}$ and $SU(2)_{\mathrm{C}}$ spins, respectively. We use $\mu$ and $\alpha$ for spacetime vector and spinor indices, and reserve $A, B$ (resp. $\dot{A}, \dot{B}$) for $SU(2)_{\mathrm{H}}$ (resp. $SU(2)_{\mathrm{C}}$) doublet indices. The arrows in (2.8) represent (anti)commutators with the supercharge

$$Q_\alpha^{A\dot{A}} \in [1]_{1/2}^{(1;1)}, \tag{2.9}$$

which generates the entire multiplet starting from the superconformal primary $\Phi$, which is a dimension $\Delta = 1$ scalar uncharged under the R-symmetries.

By supersymmetry, the two-point functions of the conformal primaries in (2.8) are all determined in terms of the stress tensor $T_{\mu\nu}$,

$$\left\langle T_{\mu\nu}(x) T_{\sigma\rho}(0) \right\rangle = \frac{C_T}{16\pi^2} \frac{\mathcal{I}_{\mu\nu,\sigma\rho}(x)}{|x|^6}. \tag{2.10}$$

In particular, the R-symmetry currents, normalized as in (2.4) for the $SU(2)_{\mathrm{H}} \times SU(2)_{\mathrm{C}}$ symmetry, have two-point functions,[6]

$$\langle j_\mu^{AB}(x) j_\nu^{CD}(0) \rangle = \frac{C_T}{96\pi^2} \frac{\epsilon^{C(A}\epsilon^{B)D} I_{\mu\nu}}{|x|^4}, \quad \langle j_\mu^{\dot{A}\dot{B}}(x) j_\nu^{\dot{C}\dot{D}}(0) \rangle = \frac{C_T}{96\pi^2} \frac{\epsilon^{\dot{C}(\dot{A}}\epsilon^{\dot{B})\dot{D}} I_{\mu\nu}}{|x|^4}. \tag{2.11}$$

As explained in [48], an efficient way to compute the conformal central charge $C_T$ for interacting SCFTs (with at least $\mathcal{N} = 2$ supersymmetry) is to couple the SCFT to a certain

---

[6]Note that the $SU(2)_{\mathrm{H}}$ currents are given by $J_\mu^{AB} \equiv J_\mu^a (\sigma_a)^{AB}$ and our normalization requires $f_{abc} = i\sqrt{2}\epsilon_{abc}$ in equation (2.4). Similarly for the $SU(2)_{\mathrm{C}}$ currents.

family of supersymmetric curved background, where the spacetime manifold is a squashed $S^3$, i.e.

$$\omega_1^2|z_1|^2 + \omega_2^2|z_2|^2 = 1, \quad z_i \in \mathbb{C}, \tag{2.12}$$

with squashing parameters $\omega_i$. Conformal invariance further implies that the partition function of the SCFT only depends on

$$b^2 \equiv \frac{\omega_1}{\omega_2}. \tag{2.13}$$

On such backgrounds, the corresponding partition function $Z_b$ is protected by supersymmetric non-renormalization theorems, and thus can be computed reliably given a UV supersymmetric gauge theory description. Localization techniques then reduce the path integral to a finite dimensional matrix model over the Coulomb branch moduli of the gauge theory [21,52–57].[7]

Thus, the central charge $C_T$ is conveniently encoded in the dependence of the partition function $Z_b$ on the squashing parameter $b$. Defining the squashed $S^3$ free energy by

$$F_b = -\log Z_b, \tag{2.14}$$

the precise formula relating $C_T$ to $F_b$ is given as follows [48][8]

$$\boxed{C_T = \frac{48}{\pi^2} \left.\frac{\partial^2 F_b}{\partial b^2}\right|_{b=1}.} \tag{2.15}$$

## 2.2 $C_J$ from mass deformed $S^3$ partition function

Having the explicit formula for the conformal central charges in hand, let us now turn to the flavor central charges. The 3d $\mathcal{N} = 4$ Higgs branch flavor-symmetry multiplet $\mathcal{B}[0]_1^{(2;0)}$ consists of conformal primaries

$$\begin{array}{ccccc}
[0]_1^{(2;0)} & \rightarrow & [1]_{3/2}^{(1;1)} & \rightarrow & [2]_2^{(0;0)} \oplus [0]_2^{(0;2)} \\
L_{AB}^a & \rightarrow & \varphi_{\alpha A\dot{A}}^a & \rightarrow & J_\mu^i \oplus N_{\dot{A}\dot{B}}^a,
\end{array} \tag{2.16}$$

where $a$ denotes the adjoint index of the flavor symmetry group $G$. The Coulomb branch flavor symmetry multiplet $\mathcal{B}[0]_1^{(0;2)}$ is similar but with the quantum numbers $R_H$ and $R_C$ interchanged.

For the simplicity of discussion, we assume that $G$ is simple in this section. By supersymmetry, the two-point functions of the operators in (2.16) are all determined in terms of the current two-point function

$$\left\langle J_\mu^a(x) J_\nu^b(0) \right\rangle = \frac{C_J}{16\pi^2} \delta^{ab} \frac{I_{\mu\nu}(x)}{x^4}. \tag{2.17}$$

In particular, it follows from the $\mathcal{N} = 4$ algebra, $\{Q_\alpha^{A\dot{A}}, Q_\beta^{B\dot{B}}\} = -i\epsilon^{AB}\epsilon^{\dot{A}\dot{B}}\gamma_{\alpha\beta}^\mu \partial_\mu$ that the two-point functions of $L^a$ and $N^a$, normalized as

$$J_\mu^a = \epsilon_{\dot{A}\dot{B}} Q^{A\dot{A}} \gamma_\mu Q^{B\dot{B}} L_{AB}^a, \quad N_{\dot{A}\dot{B}}^a = Q^{A(\dot{A}} Q^{B\dot{B})} L_{AB}^a, \tag{2.18}$$

where $\gamma_i$ are the Pauli matrices, depend on $C_J$ as follows

$$\left\langle L_{AB}^a(x) L_{CD}^b(0) \right\rangle = \frac{C_J}{64\pi^2} \delta^{ab} \frac{\epsilon_{C(A}\epsilon_{B)D}}{x^2}, \quad \left\langle N_{\dot{A}\dot{B}}^a(x) N_{\dot{C}\dot{D}}^b(0) \right\rangle = \frac{C_J}{32\pi^2} \delta^{ab} \frac{\epsilon_{C(A}\epsilon_{B)D}}{x^4}. \tag{2.19}$$

Similar to the case of $C_T$, we can obtain the flavor central charge $C_J$ for flavor symmetry $G$ by coupling the SCFT to certain supergravity backgrounds that involve mass deformations for $G$, and compute the supersymmetric partition function.

---

[7]See also [58] for a nice review on 3d supersymmetric localization, and appendix A for the relevant ingredients for our purposes.

[8]The notation here is related to that of [48] by $\tau_{rr} = \frac{C_T}{24}$ and $j_\mu^R = \frac{1}{\sqrt{2}}(j_\mu^{12} + j_\mu^{\dot{1}\dot{2}})$, where $j_\mu^R$ denotes the $U(1)_R$ current for the $\mathcal{N} = 2$ subalgebra. This is in order for the supercharges in the $\mathcal{N} = 2$ subalgebra to have $U(1)_R$ charge $\pm 1$.

**Massive supersymmetric background on $S^3$**

The Higgs branch multiplet $\mathcal{B}[0]_1^{(2;0)}$ couples to background vector fields with field content

$$D_{AB}^a, \ \lambda_{A\dot{A}}^a, \ A_\mu^a, \ \phi_{\dot{A}\dot{B}}^a. \tag{2.20}$$

To linearized order, the mass deformation of the SCFT amounts to a linear deformation of the action by

$$\delta S = \int d^3x \sqrt{g} \left( D^{aAB} L_{AB}^a + \phi^{a\dot{A}\dot{B}} N_{\dot{A}\dot{B}}^a + A_\mu^a J^{a\mu} + \text{fermionic terms} \right). \tag{2.21}$$

For our purpose, it is convenient to take the spacetime manifold to be $S^3$.

To specify the background, we begin with the round $S^3$ with no mass deformation. The $\mathfrak{osp}(4|4)$ symmetry of the SCFT on the round $S^3$ is generated by Killing spinors satisfying (see [32])

$$\nabla_\mu \xi_{A\dot{A}} = \gamma_\mu \xi'_{A\dot{A}}, \quad \nabla_\mu \xi'_{A\dot{A}} = -\frac{1}{4r^2} \gamma_\mu \xi_{A\dot{A}}, \tag{2.22}$$

where $r$ is the radius of $S^3$. In the mass-deformed theory, the maximal subalgebra we can preserve on $S^3$ is

$$\mathfrak{su}(2|1) \oplus \mathfrak{su}(2|1) \subset \mathfrak{osp}(4|4), \tag{2.23}$$

where the embedding of $\mathfrak{u}(1) \times \mathfrak{u}(1)$ is specified by two generators, one for each of $\mathfrak{su}(2)$ and $\mathfrak{su}(2)$,

$$h_A{}^B \in \mathfrak{su}(2), \quad \bar{h}^{\dot{A}}{}_{\dot{B}} \in \mathfrak{su}(2), \tag{2.24}$$

satisfying

$$h_A{}^B h_B{}^C = \delta_A^C, \quad \bar{h}^{\dot{A}}{}_{\dot{B}} \bar{h}^{\dot{B}}{}_{\dot{C}} = \delta_{\dot{C}}^{\dot{A}}, \tag{2.25}$$

and the supercharges generating the subalgebra (2.23) are specified by the restriction

$$\xi'_{A\dot{A}} = \frac{i}{2r} h_A{}^B \xi_{B\dot{B}} \bar{h}^{\dot{B}}{}_{\dot{A}}. \tag{2.26}$$

Without loss of generality, and akin to [32], we choose

$$h = -\sigma^2, \quad \bar{h} = -\sigma^3, \tag{2.27}$$

where $\sigma^i$ are the Pauli matrices. The fields in the background (abelian) vector multiplet then take the following values in order to preserve the $\mathfrak{su}(2|1) \oplus \mathfrak{su}(2|1)$ supersymmetry:

$$A_\mu^a = \lambda_{A\dot{A}}^a = 0, \quad D_{AB}^a = -\frac{1}{r} M^a \sigma_{AB}^2, \quad \phi_{\dot{A}\dot{B}}^a = M^a \sigma_{\dot{A}\dot{B}}^3, \tag{2.28}$$

where $a$ is chosen to take values $a \in \{1, 2, \ldots, n\}$, such that $T^a$ generate the Cartan subalgebra of $G$, and $M^a$ denote the mass parameters for the flavor symmetry group $G$ of the SCFT.

For this background, we define $Z(M^a)$ to be the mass-deformed partition function on $S^3$ and $F(M^a) \equiv -\log Z(M^a)$ to be the free energy of the SCFT. Following the argument in [49], while keeping track of our choice of normalization of the operators, the flavor central charge can be extracted from the $S^3$ free energy as

$$F(M^a)|_{M^2} = \frac{\pi^2 C_J^G}{16} \delta_{ab} M^a M^b, \tag{2.29}$$

where the superscript $G$ is used to emphasize the fact that the flavor central charge $C_J^G$ is with respect to the flavor symmetry group $G$.

If $G$ is a subgroup of a larger simple symmetry group $G'$ of the SCFT, then

$$C_J^{G'} = I_{G \hookrightarrow G'} C_J^G, \tag{2.30}$$

where $I_{G \hookrightarrow G'}$ denotes the Dynkin index of the embedding.[9]

**The formula for $C_J$ from the gauge theory free energy**

In practice, the mass-deformed free energy of an $\mathcal{N} = 4$ SCFT is computed by localization using its UV gauge theory (Lagrangian) description, and the mass parameters $M^a$ for the flavor symmetry are related to the gauge invariant masses of hypermultiplets.[10] However, the full symmetry $G$ can be emergent from the RG flow, and consequently only a subset of mass-deformations is accessible from the UV description. Therefore, in order to compute $C_J$ for the full symmetry group $G$ using (2.29), we need to know the precise relation between the UV and IR mass parameters as well as how the UV symmetry group $G_{\text{UV}}$ embeds into the IR symmetry group $G$.

We start by recalling the matter sector of a general 3d $\mathcal{N} = 4$ gauge theory, which is described by $n$ hypermultiplets

$$q_A^i, \ \psi_{\alpha \dot{A}}^i, \tag{2.32}$$

with $i = 1, 2, \ldots, 2n$ transforming in the fundamental representation of $USp(2n)$. The Euclidean action of hypermultiplets coupled to vector multiplets (both dynamical and background) on $S^3$ is given by

$$
\begin{aligned}
S_{\text{hyper}} = \int d^3 x \sqrt{g} \Big( & \epsilon^{AB} \Omega_{ij} D_\mu q_A^i D^\mu q_B^j - i \epsilon^{\dot{A}\dot{B}} \Omega_{ij} \psi_{\dot{A}}^i \slashed{D} \psi_{\dot{B}}^j + \frac{3}{4r^2} \epsilon^{AB} \Omega_{ij} q_A^i q_B^j \\
& - \frac{1}{2} \phi^{a\dot{A}\dot{B}} \phi_{\dot{A}\dot{B}}^b \epsilon^{AB} q_A T^a T^b q_B - iq_A D^{iAB} T^i q_B + i\psi_{\dot{A}} \phi^{a\dot{A}\dot{B}} T^a \psi_{\dot{B}} - 2iq^A \lambda_{A\dot{B}}^a T^a \psi^{\dot{B}} \Big),
\end{aligned}
\tag{2.33}
$$

where $T_{ij}^a$ are generators of $\mathfrak{usp}(2n)$. Note that the scalars are subject to the reality condition[11]

$$(q_A^i)^* = q_i^A = \epsilon^{AB} \Omega_{ij} q_B^j. \tag{2.34}$$

On the supersymmetric background (2.28), the hypermultiplet action becomes

$$
\begin{aligned}
S_{\text{hyper}} = \int d^3 x \sqrt{g} \Big( & \epsilon^{AB} \Omega_{ij} \nabla_\mu q_A^i \nabla^\mu q_B^j - i \epsilon^{\dot{A}\dot{B}} \Omega_{ij} \psi_{\dot{A}}^i \slashed{\nabla} \psi_{\dot{B}}^j + \frac{3}{4r^2} \epsilon^{AB} \Omega_{ij} q_A^i q_B^j \\
& + \epsilon^{AB} q_{(A}^i (\mathcal{M}^2)_{ij} q_{B)}^j + \frac{i}{r} (\sigma_2)^{AB} q_A^i \mathcal{M}_{ij} q_B^j + i(\sigma_3)^{\dot{A}\dot{B}} \psi_{\dot{A}}^i \mathcal{M}_{ij} \psi_{\dot{B}}^j \Big),
\end{aligned}
\tag{2.35}
$$

where

$$\mathcal{M}^i{}_j = M_{\text{UV}}^a T^{ai}{}_j, \tag{2.36}$$

---

[9] The Dynkin index of an embedding $G \subset G'$ between simple Lie groups is defined by

$$I_{G \hookrightarrow G'} \equiv \frac{\sum_i T(\mathbf{r}_i)}{T(\mathbf{r})}, \tag{2.31}$$

where $\mathbf{r}$ denotes a representation of $G'$ which decomposes into $\oplus_i \mathbf{r}_i$ under $G' \to G$, and $T(\cdot)$ denotes the quadratic index of the representation. This definition does not depend on the choice of $\mathbf{r}$.

[10] The IR mass parameters $M^a$ associated to Coulomb branch symmetries come from the FI parameters in the UV gauge theory instead. The generalization is straightfoward.

[11] $q^i \equiv q_1^i$, $\tilde{q}_i \equiv q_i^2$ for $i = 1, \cdots, n$ have the same $R_3^H$ charge and represent $\mathcal{N} = 2$ chiral multiplet scalars [31].

with $a$ restricted to $a = 1, 2, \ldots, n$ labelling the Cartans of $\mathfrak{usp}(2n)$, encodes the UV mass parameters (or vector multiplet scalar vevs).

When a subgroup $H \subset USp(2n)$ is gauged, the residual flavor symmetry is given by the commutant which we will refer to as $G_{\text{UV}}$. In the cases we are interested in here, the hypermultiplets transform in a single representaton $(R_{G_{\text{UV}}}, R_{\text{H}})$ of $G_{\text{UV}} \times H \subset USp(2n)$, with indices $(i, \hat{\imath})$ and $\dim(R_{G_{\text{UV}}}) \dim(R_{\text{H}}) = 2n$. The mass matrix decomposes as

$$\mathcal{M}^{i\hat{\imath}}{}_{j\hat{\jmath}} = M_{\text{UV}}^a T^{ai}{}_j \mathcal{K}^{\hat{\imath}}{}_{\hat{\jmath}} + \phi^{\hat{a}} \hat{T}^{\hat{a}\hat{\imath}}{}_{\hat{\jmath}} \mathcal{I}^i{}_j, \tag{2.37}$$

where $\mathcal{I}^i{}_j$ and $\mathcal{K}^{\hat{\imath}}{}_{\hat{\jmath}}$ are invariant tensors of $G_{\text{UV}}$ and $H$, respectively, satisfying

$$\mathcal{I}^i{}_j \mathcal{I}^j{}_k = \delta^i{}_k, \quad \mathcal{K}^{\hat{\imath}}{}_{\hat{\jmath}} \mathcal{K}^{\hat{\jmath}}{}_{\hat{k}} = \delta^{\hat{\imath}}{}_{\hat{k}}. \tag{2.38}$$

We still denote the adjoint index of $G_{\text{UV}}$ by $a$ and use $\hat{a}$ for the adjoint index for the gauge group $H$. In the gauge theory, $\phi^{\hat{a}}$ are dynamical while $M_{\text{UV}}^a$ are mass parameters.

We claim that the UV and IR mass parameters for $G_{\text{UV}}$ are simply identified as

$$M_{\text{UV}}^a = M^a. \tag{2.39}$$

This is equivalent to checking the normalization of the conserved current multiplets in the UV gauge theory description. Comparing (2.33) with (2.21), we identify the currents for $G_{\text{UV}}$[12]

$$J_\mu^a = 2 T^{ai}{}_j \left[ i \epsilon^{AB} q_{Ai} \partial_\mu q_B^j - \frac{1}{2} \epsilon^{\dot{A}\dot{B}} \psi_{\alpha \dot{A} i} \gamma_\mu^{\alpha\beta} \psi_{\beta \dot{B}}{}^j \right], \tag{2.40}$$

and the moment map operator

$$L_{AB}^a = -T^{aij} q_{(Ai} q_{B)j}. \tag{2.41}$$

In flat space, the two-point functions of free massless hypermultiplet fields in (2.35) are given by

$$\langle q_A^i(x) q_B^j(y) \rangle = \frac{1}{8\pi} \frac{\Omega^{ij} \epsilon_{AB}}{|x-y|}, \quad \langle \psi_{\dot{A}}^i(x) \psi_{\dot{B}}^j(y) \rangle = \frac{i}{8\pi} \epsilon_{\dot{A}\dot{B}} \Omega^{ij} \frac{\gamma_\mu(x^\mu - y^\mu)}{|x-y|^3}. \tag{2.42}$$

Performing explicit Wick contractions, it is easy to see that (2.40) indeed satisfies (2.4). We can also compute the UV flavor central charge (central charge for the free theory) from the two-point function of (2.40),

$$\langle J_\mu^a(x) J_\nu^b(y) \rangle = \frac{4 \delta^{ab}}{16\pi^2} I_{\mu\nu} \dim(R_{\text{H}}), \tag{2.43}$$

which gives[13]

$$C_J^{G_{\text{UV}}}(\text{hyper}_n) = 4 \dim(R_{\text{H}}). \tag{2.44}$$

To summarize, for UV flavor symmetry $G_{\text{UV}} \subset G_{\text{UV}} \times H \subset USp(2n)$ carried by gauged hypermultiplets, we define $\mathcal{M}_{\text{UV}}$ as the reduced mass matrix such that the full mass matrix for the hypermultiplets is

$$\mathcal{M} = \mathcal{M}_{\text{UV}} \otimes \mathcal{K} \oplus \text{ vector multiplet scalar vevs}, \tag{2.45}$$

where $\mathcal{K}$ is a normalized invariant tensor for the gauge group $H$ introduced in (2.37). Then the flavor central charge of the SCFT for $G_{\text{UV}}$ is determined by the gauge theory supersymmetric free energy via

$$\boxed{F|_{\mathcal{M}^2} = \frac{\pi^2 C_J^{G_{\text{UV}}}}{16} \text{Tr} \, \mathcal{M}_{\text{UV}}^2.} \tag{2.46}$$

---

[12]Here and below we focus on the gauge-invariant currents and suppress all gauge indices $\iota, J$ which have been contracted with the invariant tensor $\mathcal{I}^\iota{}_J$.

[13]Note that for $n$ free hypermultiplets, $C_J^{USP(2n)}(\text{hyper}_n) = 4$. The extra factor of $\dim(R_{\text{H}})$ comes from the embedding index for the subgroup $G_{\text{UV}} \subset G_{\text{UV}} \times H \subset USp(2n)$ (see (2.31)).

Table 1: Conformal and flavor central charges $C_T$ and $C_J$ for candidate minimal SCFTs with Higgs branches given by the minimal nilpotent orbits of $G$, as computed by localization.

| $G$ | Candidate minimal theory |
|-----|--------------------------|
| $SU$ | SQED |
| $SO$ | $SU(2)$ SQCD |
| $USp$ | Free hypers |
| $E_6$ | $T_3$ |

| $G$ | $C_J$ | $C_T$ |
|-----|-------|-------|
| $SU(2)$ | $\frac{16}{3}$ | $\frac{64}{3} + \frac{16}{\pi^2}$ |
| $SU(3)$ | $6$ | $54 - \frac{192}{\pi^2}$ |
| $SU(4)$ | $\frac{32}{5}$ | $\frac{256}{5} - \frac{48}{\pi^2}$ |
| $SU(5)$ | $\frac{20}{3}$ | $100 - \frac{3712}{9\pi^2}$ |
| $SO(5)$ | $4$ | $54 - \frac{64}{\pi^2}$ |
| $SO(6)$ | $\frac{32}{5}$ | $\frac{272}{5} + \frac{48}{\pi^2}$ |
| $SO(7)$ | $8$ | $156 - \frac{12416}{15\pi^2}$ |
| $SO(8)$ | $\frac{64}{7}$ | $\frac{736}{7} - \frac{200}{\pi^2}$ |
| $USp(2n)$ | $4$ | $12n$ |
| $E_6$ | $\frac{192}{13}$ | $160.246$ |

## 2.3 Minimal theories with flavor symmetry $G$

Given a simple Lie group $G$, there is a notion of minimality for 3d $\mathcal{N} = 4$ SCFTs with flavor symmetry $G$: Assuming that the symmetry $G$ is generated by the Higgs branch current multiplets, $G$ naturally acts on the Higgs branch moduli space of the theory. A special class of such Higgs branch geometries are given by (the closure of) the nilpotent orbits of $G$, among which the minimal nilpotent orbit $\mathcal{O}_{\min}(G)$ has complex dimension $2(h^{\vee}(G) - 1)$ and is the one-instanton moduli space of $G$ on $\mathbb{R}^4$. A "minimal" theory is one whose Higgs branch moduli space is (the closure of) the minimal nilpotent orbit, and whose conformal central charge $C_T$ is the smallest.

The minimal nilpotent orbit $\mathcal{O}_{\min}(G)$ has a simple description as a holomorphic-symplectic variety embedded in the Lie algebra $\mathfrak{g}$ of $G$,

$$\mathcal{O}_{\min}(G) = \{\mathfrak{g} \,|\, \mathcal{I}_2 = 0\}, \tag{2.47}$$

where $\mathcal{I}_2$ denotes the Joseph ideal in the polynomial algebra of $\mathfrak{g}$ defined by

$$(\mathbf{adj} \otimes \mathbf{adj})_S = \mathbf{2adj} \oplus \mathcal{I}_2. \tag{2.48}$$

For this reason, we refer to the condition in (2.47) as the Joseph relations. We expect such conditions to impose strong constraints. Indeed as we will see in Section 4, the Higgs branch protected operator algebra (1d TQM) [22, 31–34] can be solved completely and uniquely (except in the case of $G = A_1$) by certain generalized higher-spin algebra for $G$.

Below we present a list of candidate *minimal* theories for each simple Lie group $G$, and provide their conformal and flavor central charges computed using the formulae presented in the previous sections. The results are also summarized in Table 1. We leave the details of the computations to Appendices A.3 and A.4.

### $n$ Free hypermultiplets

Consider $n$ free hypermultiplets with the following $\mathbb{Z}_2$ symmetry gauged:[14]

$$\mathbb{Z}_2 : \ q_A^i \to -q_A^i. \tag{2.49}$$

This is precisely the ADHM gauge theory for the 1-instanton moduli space of $USp(2n)$ on $\mathbb{R}^4$. The conformal and flavor central charges are then given by (see (A.9) and (A.48))

$$
\begin{aligned}
C_T(\text{hyper}_n) &= 12n, \\
C_J^{USp(2n)}(\text{hyper}_n) &= 4.
\end{aligned}
\tag{2.50}
$$

### $\mathcal{N} = 4$ SQED with $n \geq 2$ unit-charge hypermultiplets

The 3d $\mathcal{N} = 4$ SQED coupled to $n$ unit-charge hypermultiplets flows in the IR to an interacting SCFT for $n \geq 2$. This is precisely the ADHM gauge theory for the one-instanton moduli space of $SU(n)$ on $\mathbb{R}^4$. The theory has a mirror-dual description as an affine $A_{n-1}$ type quiver theory, which is a cyclic quiver consisting of $n$ $U(1)$ gauge-nodes[15] and $n$ bifundamental hypermultiplets. From the SQED description, we can extract the conformal and flavor central charges, given by

$$C_T(\text{SQED}_n) = \frac{12\left(2n^2\psi^{(1)}\left(\frac{n}{2}+1\right)+\left(\pi^2 n+4\right)n+4\right)}{\pi^2(n+1)}, \tag{2.51}$$

where $\psi^{(1)}(z) \equiv \frac{d}{dz}\frac{\Gamma'(z)}{\Gamma(z)}$ is the $z$-derivative of the digamma function, and

$$C_J^{SU(n)}(\text{SQED}_n) = \frac{8n}{n+1}. \tag{2.52}$$

There is an interesting family of $U(1)_k \times U(1)_{-k}$ Chern-Simons matter theories with $SU(n)$ flavor symmetry: the $\text{II}_k(n+1)$ theories of [59]. We compute the values of $C_T$ and $C_J$ for the $\text{II}_k(3)$ series in Appendix A.3 and A.4. The minimal $k = 2$ theories have enhanced $SU(n+1)$ symmetry and are dual to the SQED with $n+1$ unit-charge hypermultiplets.

### $\mathcal{N} = 4$ SQCD with $SU(2)$ gauge group and $n \geq 5$ fundamental half-hypermultiplets

The 3d $\mathcal{N} = 4$ SQCD with $SU(2)$ gauge group coupled to $n$ fundamental half-hypermultiplets flows in the IR to an interacting SCFT for $n \geq 5$. This is precisely the ADHM gauge theory for the one-instanton moduli space of $SO(n)$ on $\mathbb{R}^4$.

When $n$ is even, the theory has a known mirror-dual description in terms of a 3d $\mathcal{N} = 4$ affine $D_{n/2}$ shaped quiver. From the SQCD description, we can extract the conformal and flavor central charges, given by

$$
\begin{aligned}
&C_T(\text{SQCD}_n) = \\
&\frac{12\left(n\left(\pi^2(n-4)(n-2)+12n-20\right)(n-4)+2(n-2)(3n-2)(n-4)^2\psi^{(1)}\left(\frac{n-2}{2}\right)+48\right)}{\pi^2(n-4)(n-2)(n-1)},
\end{aligned}
\tag{2.53}
$$

and

$$C_J^{SO(n)}(\text{SQCD}_n) = \frac{16(n-4)}{n-1}. \tag{2.54}$$

---

[14]One can gauge the $\mathbb{Z}_2$ symmetry by coupling to a nontrivial 3d $\mathbb{Z}_2$ gauge theory, but this will not make a difference on the local operator spectrum. In fact, even if we do not gauge $\mathbb{Z}_2$, although the full (free) theory does not satisfy the minimal nilpotent orbit condition, the OPE of $\mathbb{Z}_2$ singlets is still identical to the gauged versions, and suffice for mere comparisons with bootstrap.

[15]The diagonal $U(1)$ vector multiplet decouples.

**$E_{6,7,8}$ theories**

The 4d $\mathcal{N} = 2$ $E_n$ Minahan-Nemeschansky theories [60,61] have Higgs branches realized by the one-instanton moduli spaces of $E_n$. Their $S^1$ reduction naturally gives rise to 3d $\mathcal{N} = 4$ $E_n$ theories that inherit the 4d Higgs branch. Unlike their 4d parents, these 3d theories do have (mirror) Lagrangian descriptions given by quivers of affine $E_{6,7,8}$ types, which we can use to compute the conformal and flavor central charges.

In the case of $E_6$ (also known as $T_3$ theory), we carry out the computation in Appendix A.3, and obtain

$$
\begin{aligned}
C_T(T_3) &= 160.246, \\
C_J^{E_6}(T_3) &= 4.
\end{aligned}
\tag{2.55}
$$

In later sections, we give an independent derivation of $C_J$ by solving the 1d TQM associated to the Higgs branch.

**$G_2, F_4$ theories**

These are the most mysterious 3d $\mathcal{N} = 4$ SCFTs in this class whose existence are purely conjectural. There are no known string/M-theory constructions for them. However, if they exist, then we can determine $C_J$ unambiguously by solving the 1d TQM associated to the Higgs branch of a would-be $G_2$ or $F_4$ minimal theory (see Section 4 and Table 5).

## 2.4 $T[G]$ theories and maximal nilpotent orbit

Let us now discuss a special class of theories, dubbed $T[G]$. They were originally introduced in [43] as the S-dual [62] (under the S-tranformation of the full S-duality group $SL(2,\mathbb{Z})$) of particular (Dirichlet for the 4d $\mathcal{N} = 2$ vector multiplets and Neumann for the adjoint hypermultipelts) boundary conditions of the 4d $\mathcal{N} = 4$ supersymmetric Yang-Mills theory with gauge group $G$.[16] More precisely, the 4d S-dual configuration consists of the 3d $T[G]$ theory coupled to the 4d bulk gauge fields.

The $T[G]$ theories are 3d $\mathcal{N} = 4$ SCFTs in the infrared, and carry (IR) $G^\vee \times G$ ($G^\vee$ is the GNO or Langlands dual of $G$ [64]) flavor symmetry, with $G^\vee$ and $G$ realized on the Coulomb branch and Higgs branch, respectively. The $G^\vee$ and $G$ act faithfully only through their adjoint forms, and thus the distinction (for observables that only involve local operators) between them is only relevant when they have different Lie algebras. The Higgs branch is given by the maximal nilpotent orbit $\mathcal{N}_{\mathfrak{g}}$ of $G$ with complex dimension

$$
d \equiv \dim_{\mathbb{C}} \text{Higgs} = \dim(\mathfrak{g}) - \text{rank}(\mathfrak{g}).
\tag{2.56}
$$

The Coulomb branch is given by the maximal nilpotent orbit $\mathcal{N}_{\mathfrak{g}^\vee}$ of $G^\vee$ with the same complex dimension.[17] Under 3d $\mathcal{N} = 4$ mirror symmetry [29,30], $T[G]$ is mapped to $T[G^\vee]$ and vice versa, with Coulomb and Higgs branches interchanged.

These facts can be observed by realizing that the theory $T[G]$ is given by the S-dual to Dirichlet boundary conditions in $\mathcal{N} = 4$ super-Yang-Mills theory with gauge group $G$ (see [43] for more details). Consider the $\mathcal{N} = 4$ super-Yang-Mills theory with gauge group $G$ coupled to $T[G]$. The coupling lifts the Higgs branch and we are left with the Coulomb branch. Then, upon performing an S-transformation, we obtain $\mathcal{N} = 4$ super-Yang-Mills with gauge group $G^\vee$ and Dirichlet boundary conditions, whose moduli space is given by the nilpotent cone $\mathcal{N}_{\mathfrak{g}^\vee}$ of $G^\vee$ (found by solving Nahm's equation) [65]. We conclude that this is the Coulomb branch of

---

[16]Recall that for $G = G_2$ and $G = F_4$, even though the Langlands dual pairs have identical Lie algebras, the S-duality group is not a subgroup of $SL(2,\mathbb{Z})$ [63]. In particular, the S-transformation involves a nontrivial involution on the vacuum moduli space.

[17]The full vacuum moduli space also contains mixed branches.

$T[G]$. Similarly, one can couple the 4d theory to Dirichlet boundary conditions on the left and on the right of the 3d $T[G]$ theory domain wall. This ungauges (freezes) both gauge groups $G$ and $G^\vee$. Then, being a symmetric setup under exchange of left and right, one concludes that $T[G]$ is mirror dual to $T[G^\vee]$.

In the case of the $T[SU(N)]$ theory, there is a convenient description in terms of a D3-D5-NS5 brane system in type IIB string theory [43, 66]. As the example of $T[SU(2)]$ theory plays a major role in the following, let us elaborate some more on the brane constructions of $T[SU(N)]$ theories (we refer to the original paper for a more exhaustive discussion [43]). We start with a set of D3 branes, spanned by $x^0, \dots x^3$ inside 10d Minkowski space (with coordinates $x^0, \dots x^9$), whose worldvolume theory leads to the $\mathcal{N} = 4$ super-Yang-Mills theory in the bulk. To describe half-BPS boundary conditions, we may add D5 branes, spanned by $x^0, x^1, x^2$, and $x^4, x^5, x^6$, as well as NS5 branes, spanned by $x^0, x^1, x^2$, and $x^7, x^8, x^9$, on which the D3 branes may end (along $x^3$), see Table 2. In Figure 1, we illustrate this for the examples of the $T[SU(5)]$-theory.

The brane construction makes the mirror self-duality manifest: Famously, S-duality – or rather the S-transformation inside the full S-duality group – acts by exchanging NS5- and D5 branes. Thus, starting with the brane setup in Figure 1, applying S-duality, and then repeatedly performing Hanany-Witten moves [66] (now moving four NS5 branes to the left of all the D5's and one to the right), it is straightforward to see, that the resulting brane configuration precisely reduces to the one we started with.

Table 2: The D3-D5-NS5 brane system in type IIB string theory, describing boundary conditions of 4d $\mathcal{N} = 4$ super-Yang-Mills theory (for unitary groups).

|     | 0 | 1 | 2 | 3 | 4 | 5 | 6 | 7 | 8 | 9 |
|-----|---|---|---|---|---|---|---|---|---|---|
| D3  | × | × | × | × |   |   |   |   |   |   |
| D5  | × | × | × |   | × | × | × |   |   |   |
| NS5 | × | × | × |   |   |   |   | × | × | × |

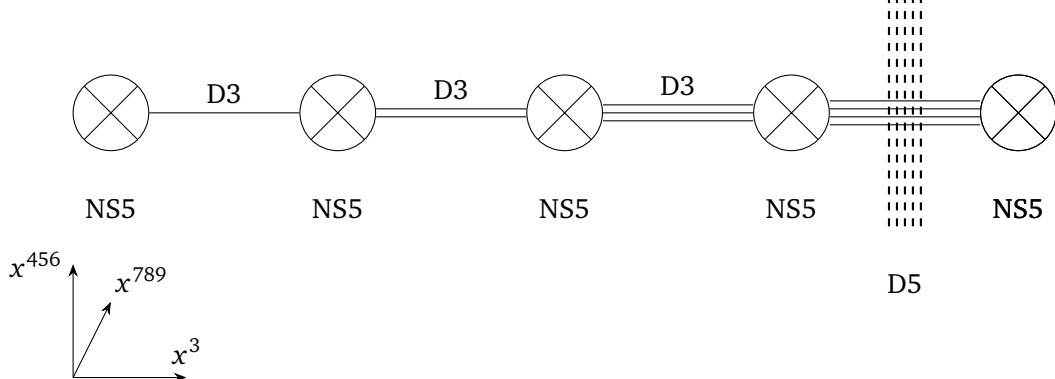

Figure 1: The D3-D5-NS5 brane system in type IIB string theory, describing the 3d $\mathcal{N} = 4$ $T[SU(5)]$ theory. Here, the D5 branes can be moved beyond the right-most NS5 brane by standard Hanany-Witten moves. The SCFT is reached by taking the limit in which the separations along $x^3$ are taken to zero, *i.e.* we are moving onto the origin of the moduli space.

By separating the NS5 branes in the $x_3$ direction (which corresponds to tunning the FI parameters in the field theory), this brane construction naturally provides a UV Lagrangian

description for the $T[SU(N)]$ theories, given by the quiver

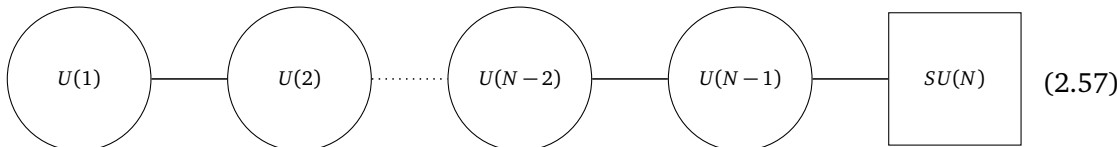

$$(2.57)$$

where (as usual) the circular nodes are gauge-groups connected by bifundamental hypermultiplets, while the squares denote flavor symmetry factors. In the IR, the $SU(N)$ flavor symmetry of the Higgs branch is manifest, while the $SU(N)$ flavor symmetry of the Coulomb branch comes from IR enhancement due to monopole operator.[18]

Similarly, by including various types of $O3$ planes in the brane setup [67], one may write down the UV quiver gauge theory for $T[SO(2N)]$,

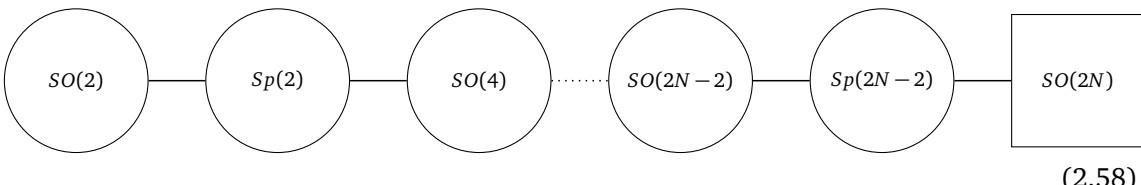

$$(2.58)$$

and for $T[Sp(2N)]$ theories [43],

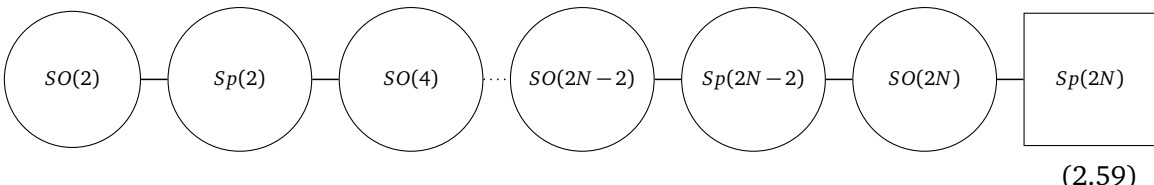

$$(2.59)$$

We remark that for the remaining cases, there are no – or only "bad" in the sense of [43] – quiver descriptions. However, below we conjecture the general $S^3$ partition function even for those cases.

The Higgs branch chiral ring is generated by the moment map operators, which are dimension-1 scalars in the adjoint representation, denoted collectively by the matrix-valued operator $N$. They are subject to the chiral ring relations

$$\epsilon_i(N) = 0, \qquad (2.60)$$

where $\epsilon_i$ label the fundamental Casimirs of $\mathfrak{g}$. For $G = SU(n)$, this is just the statement that $\mathrm{Tr}\, N^i = 0$ for $i = 2, 3, \ldots, n$. In general, this means that the singlet representation is absent from symmetric tensor products of the moment map operators. Compared to the minimal nilpotent orbit theories, these theories sit at the opposite end of 3d theories carrying $G$ flavor symmetry: by tuning Higgs branch vevs, one can flow from $T[G]$ to any of the 3d $\mathcal{N} = 4$ theories with nilpotent sub-orbits as Higgs branches.

Let $\mathfrak{g}$ be the Lie algebra of a simple Lie group $G$. The $S^3$ partition function for $T[G]$ theories was

$$Z^{T[G]}(\mathcal{M}; \tilde{\mathcal{M}}) = \frac{\sum_{w \in W(G)} (-1)^{l(w)} e^{2\pi i (w(\mathcal{M}) \cdot \tilde{\mathcal{M}})}}{i^{\frac{d}{2}} \prod_{\alpha \in \Delta^+} 2 \sinh(\pi(\alpha \cdot \mathcal{M})) \prod_{\alpha \in \Delta^+} 2 \sinh(\pi(\alpha \cdot \tilde{\mathcal{M}}))}. \qquad (2.61)$$

Here, $d$ is the Higgs branch dimension (2.56), $\mathcal{M}, \tilde{\mathcal{M}}$ are Cartan elements of the Lie algebra $\mathfrak{g}$ representing the mass and FI parameters for the $G \times G^\vee$ flavor symmetry, $\Delta^+$ denotes the

---

[18]Note that for the $T[SU(2)]$ theory (*i.e.* the SQED with 2 flavors), there is an alternative description as a Chern-Simons theory [59]. There, the $U(1) \times U(1) \subset SU(2) \times SU(2)$ flavor symmetry is manifest, and furthermore the Higgs and Coulomb branches are put on equal footing.

set of positive roots, $W(G)$ is the Weyl group of $G$, and $l(w)$ denotes the length of the Weyl element $w$. The product $(\cdot)$ is the standard Killing form on $\mathfrak{h}$ or $\mathfrak{h}^*$.

This conjectured form of the $S^3$ partition function of the $T[G]$ theory is motivated as a generalization of the case of $G = A_{n-1}$, which was proven by induction in [68, 69]. Furthermore, it passes a variety of non-trivial checks: first, it is manifestly mirror symmetric, *i.e.* we end up with the conjectured answer for $T[G^\vee]$ upon exchanging the mass and FI parameters, corresponding to a mirror symmetry transformation; second, it behaves appropriately under the addition of line defects (Wilson and vortex lines).

Let us briefly elaborate on this and sketch the logic: It is possible to decorate the $T[G]$ theories with a set of line defects. For instance, given a UV Lagrangian description, we can compute the partition function with the addition of (flavor) Wilson lines (along the large circle in $S^3$) in a representation $\mathcal{R}$ of $G$ [70]. This will not enter the supersymmetric localization computation of the $S^3$ partition function apart from a multiplicative factor, given by the addition of a Schur character in the mass-parameters of the representation $\mathcal{R}$ of $G$ [21], *i.e.* the Wilson loop vev is given by

$$\langle W_{\mathcal{R}} \rangle_{T[G]} = \operatorname{tr}_{\mathcal{R}} \exp(2\pi i \mathcal{M}). \tag{2.62}$$

On the other hand, we can consider adding (flavor) vortex lines $V_{\mathcal{R}}$ to $T[G^\vee]$, which are labeled by a representation $\mathcal{R}$ of $(G^\vee)^\vee = G$. As opposed to Wilson lines, vortex lines are disorder operators characterized by a vortex-like singularity for the fields in a theory near a curve in spacetime.[19] It is known that vortex lines are mirror dual to Wilson lines [72–74]. In particular, this means that their VEVs ought to agree upon adding them to mirror dual theories. The vortex lines act on the integrand of the $S^3$ partition function by a sum over shifts of the mass/FI parameters, *i.e.* up to an overall factor,

$$V_{\mathcal{R}} \circ Z(m) \propto \sum_{\rho \in \mathcal{R}} Z(m + \mathrm{i}(m, \rho)), \tag{2.63}$$

where the sum is over the weights $\rho$ of the representation $\mathcal{R}$, and we have collectively denoted by $m$ the masses (or equivalently FI-parameters).[20] Thus, we can first evaluate the integral, obtaining $Z$ in (2.61), and then act with the vortex line operators by shifts [72, 74, 77]. By doing so and using the fact that weights are invariant under the action of the Weyl group, one can show that indeed

$$\langle V_{\mathcal{R}} \rangle_{T[G^\vee]} = \langle W_{\mathcal{R}} \rangle_{T[G]}, \tag{2.64}$$

adding further evidence for the conjecture in (2.61).

Now, let us proceed to compute the conformal central charge $C_J$ for $T[G]$ theories of a single flavor factor $G$ using (2.46). We first remark that we only need the denominator of (2.61), since terms from the numerator necessarily mix $G$ and $G^\vee$. Let us expand the partition function in small $\mathcal{M}$ (and finite $\tilde{\mathcal{M}}$) and only keep track of the $\sinh(\pi\alpha \cdot \mathcal{M})$ factor in the denominator. We find

$$\frac{Z^{T[G]}(\mathcal{M}; \tilde{\mathcal{M}})|_{\mathcal{M}^2}}{Z^{T[G]}(\mathcal{M}; \tilde{\mathcal{M}})|_0} = -\frac{\pi^2}{6} \sum_{\alpha \in \Delta^+} (\alpha \cdot \mathcal{M})^2, \tag{2.65}$$

where $A|_{\mathcal{M}^p}$ means the $p$-th term in the small $\mathcal{M}$ expansion of $A$. Next, we use the following completeness relation,

$$(\lambda \cdot \mu) = \frac{1}{h^\vee} \sum_{\alpha \in \Delta^+} (\lambda \cdot \alpha)(\alpha \cdot \mu), \tag{2.66}$$

---

[19]Such operators are somewhat analogously defined to 4d Gukov-Witten type surface defects [71]. Alternatively, they can be defined by coupling the 3d theory to a 1d (supersymmetric) quantum mechanics, see *e.g.* [72].

[20]In fact, they can be viewed as dimensional reductions of Macdonald q-difference operators [75–77], which can be defined for general root systems.

where $\lambda$ and $\mu$ are arbitrary weights, to arrive at a compact formula for $C_J$ of $T[G]$ with respect to the symmetry group $G$:

$$C_J^G(T[G]) = \frac{8}{3}h^\vee(G). \tag{2.67}$$

The $C_T$ of $T[G]$ for classical simple Lie group $G$ can be computed using the formula (2.15) and the quiver descriptions (2.57), (2.58), and (2.59). In Appendix A.3, we explicitly evaluated the $C_T$ of $T[SU(3)]$. The result is

$$C_T^{SU(3)}(T[SU(3)]) = 75.5329. \tag{2.68}$$

## 2.5 $\mathcal{N} = 8$ theories

There are three classes of $\mathcal{N} = 8$ theories: the $U(N)_k \times U(N)_{-k}$ ABJM theories for $k = 1, 2$, the $U(N+1)_2 \times U(N)_{-2}$ ABJ theories, and the $SU(2)_k \times SU(2)_{-k}$ BLG theories [12,13,35–40]. Viewing these as $\mathcal{N} = 4$ theories, the $SO(8)$ R-symmetry decomposes into the $SU(2)_C \times SU(2)_H$ R-symmetry and the $SU(2)_C^f \times SU(2)_H^f$ flavor symmetry. Among these theories, the $U(1)_k \times U(1)_{-k}$ ABJM theories for $k = 1, 2$ and the $U(2)_2 \times U(1)_{-2}$ ABJ theory have minimal nilpotent Coulomb and Higgs branches. The central charges of them were computed in [22], and we summarize the results in Table 3.

Table 3: The conformal and flavor central charges of the $\mathcal{N} = 8$ theories with minimal nilpotent Coulomb and Higgs branches when viewed as $\mathcal{N} = 4$ theories.

| theories | $C_T$ | $C_J^{SU(2)}$ |
|---|---|---|
| $U(1)_1 \times U(1)_{-1}$ ABJM | 24 | 4 |
| $U(1)_2 \times U(1)_{-2}$ ABJM | 24 | 4 |
| $U(2)_2 \times U(1)_{-2}$ ABJ | 32 | $\frac{16}{3}$ |

The central charges of another infinite class of $\mathcal{N} = 8$ theories were computed in [22]. They are the $U(2)_k \times U(2)_{-k}$ ABJM theories for $k = 1, 2$, and the $SU(2)_k \times SU(2)_{-k}$ BLG theories for all positive integer $k$. The formulae are

$$c_T = 32\left(2 - \frac{I_4}{I_2}\right), \quad I_n = \int_{-\infty}^{\infty} dy\, y\, \frac{\tanh^n(\pi y)}{\sinh(\pi k y)},$$
$$C_T = \frac{3}{2}c_T, \quad C_J^{SU(2)} = \frac{1}{4}c_T. \tag{2.69}$$

# 3 Superconformal blocks

## 3.1 Superconformal algebra, representations, and Casimir operator

The three-dimensional $\mathcal{N} = 4$ superconformal algebra is $OSp(4|4)$, which contains the bosonic conformal algebra $SO(3, 2)$ with generators $P_\mu$, $K_\mu$, $D$, and the R-symmetry group $SU(2)_C \times SU(2)_H$ with generators $R_I^C$ and $R_I^H$. The fermionic generators of $OSp(4|4)$ are the Poincaré Q-supercharges $Q_{\alpha A \dot{A}}$ and superconformal S-supercharges $S^{\alpha A \dot{A}}$. The (anti)commutators in the superconformal

algebra that involve fermionic generators are

$$
\begin{aligned}
&\{Q_{\alpha A\dot{A}},Q_{\beta B\dot{B}}\} = \epsilon_{AB}\epsilon_{\dot{A}\dot{B}}(\sigma^{\mu})_{\alpha\beta}P_{\mu}, \quad \{S^{\alpha A\dot{A}},S^{\beta B\dot{B}}\} = -\epsilon^{AB}\epsilon^{\dot{A}\dot{B}}(\sigma^{\mu})^{\alpha\beta}K_{\mu}, \\
&[K_{\mu},Q_{\alpha A\dot{A}}] = \epsilon_{AB}\epsilon_{\dot{A}\dot{B}}(\sigma_{\mu})_{\alpha\beta}S^{\beta B\dot{B}}, \quad [P_{\mu},S^{\alpha A\dot{A}}] = \epsilon^{AB}\epsilon^{\dot{A}\dot{B}}(\sigma_{\mu})^{\alpha\beta}Q_{\beta B\dot{B}}, \\
&[M_{\mu\nu},Q_{\alpha A\dot{A}}] = (m_{\mu\nu})^{\beta}{}_{\alpha}Q_{\beta A\dot{A}}, \quad [M_{\mu\nu},S^{\alpha A\dot{A}}] = -(m_{\mu\nu})^{\alpha}{}_{\beta}S^{\beta A\dot{A}}, \\
&[R^{\mathrm{C}}_{I},Q_{\alpha A\dot{A}}] = \frac{1}{2}(\sigma_{I})^{B}{}_{A}Q_{\alpha B\dot{A}}, \quad [R^{\mathrm{C}}_{I},S^{\alpha A\dot{A}}] = -\frac{1}{2}(\sigma_{I})^{A}{}_{B}S^{\alpha B\dot{A}}, \\
&[R^{\mathrm{H}}_{\dot{I}},Q_{\alpha A\dot{A}}] = \frac{1}{2}(\sigma_{\dot{I}})^{\dot{B}}{}_{\dot{A}}Q_{\alpha A\dot{B}}, \quad [R^{\mathrm{H}}_{\dot{I}},S^{\alpha A\dot{A}}] = -\frac{1}{2}(\sigma_{\dot{I}})^{\dot{A}}{}_{\dot{B}}S^{\alpha A\dot{B}}, \\
&\{S^{\alpha A\dot{A}},Q_{\beta B\dot{B}}\} = i\delta^{\alpha}_{\beta}\delta^{A}_{B}\delta^{\dot{A}}_{\dot{B}}D + \delta^{A}_{B}\delta^{\dot{A}}_{\dot{B}}(m_{\mu\nu})^{\alpha}{}_{\beta}M^{\mu\nu} - \delta^{\alpha}_{\beta}\delta^{\dot{A}}_{\dot{B}}(\sigma_{I})^{A}{}_{B}R^{\mathrm{C}}_{I} - \delta^{\alpha}_{\beta}\delta^{A}_{B}(\sigma_{\dot{I}})^{\dot{A}}{}_{\dot{B}}R^{\mathrm{H}}_{\dot{I}},
\end{aligned}
\tag{3.1}
$$

where $(\sigma_{\mu})^{\alpha}{}_{\beta}$, $(\sigma_{I})^{A}{}_{B}$, $(\sigma_{\dot{I}})^{\dot{A}}{}_{\dot{B}}$ are Pauli matrices. The indices are raised and lowered by $\epsilon^{\alpha\beta}$, $\epsilon^{AB}$, $\epsilon^{\dot{A}\dot{B}}$ and $\epsilon_{\alpha\beta}$, $\epsilon_{AB}$, $\epsilon_{\dot{A}\dot{B}}$ in the convention that upper left indices contract with lower right indices; for example,

$$
(\sigma_{\mu})^{\alpha\beta} = \epsilon^{\beta\gamma}(\sigma_{\mu})^{\alpha}{}_{\gamma}, \quad (\sigma_{\mu})_{\alpha\beta} = (\sigma_{\mu})^{\gamma}{}_{\beta}\epsilon_{\gamma\alpha}.
\tag{3.2}
$$

The spacetime rotation matrices $m_{\mu\nu}$ are defined by

$$
(m_{\mu\nu})^{\alpha}{}_{\beta} = -\frac{i}{4}\left[(\sigma_{\mu})^{\alpha\gamma}(\sigma_{\nu})_{\gamma\beta} - (\sigma_{\nu})^{\alpha\gamma}(\sigma_{\mu})_{\gamma\beta}\right] = \frac{1}{2}\epsilon_{\mu\nu\rho}(\sigma^{\rho})^{\alpha}{}_{\beta}.
\tag{3.3}
$$

The $SU(2)_{\mathrm{C}}$ and $SU(2)_{\mathrm{H}}$ are exchanged under a $\mathbb{Z}_2$ outer automorphism (see Section 3.2).

The unitary representations (superconformal multiplets) can be constructed by successively acting with $Q_{\alpha A\dot{A}}$, $M_{23}-iM_{31}$, $R^{\mathrm{C}}_1-iR^{\mathrm{C}}_2$, and $R^{\mathrm{H}}_1-iR^{\mathrm{H}}_2$ on the highest weight states, which are states annihilated by $K_{\mu}$, $S^{\alpha A\dot{A}}$, $M_{23}+iM_{31}$, $R^{\mathrm{C}}_1+iR^{\mathrm{C}}_2$, and $R^{\mathrm{H}}_1+iR^{\mathrm{H}}_2$. The superconformal multiplets of $OSp(4|4)$ are classified into long multiplets as well as $\mathcal{A}$- and $\mathcal{B}$-type short multiplets [50, 51, 78]. They are labeled by

$$
\mathcal{X}[2\ell]^{(2j_{\mathrm{C}};2j_{\mathrm{H}})}_{\Delta}
\tag{3.4}
$$

and satisfy the unitarity conditions

$$
\begin{aligned}
\mathcal{L}: &\quad \Delta > \ell + j_{\mathrm{C}} + j_{\mathrm{H}} + 1, \\
\mathcal{A}: &\quad \Delta = \ell + j_{\mathrm{C}} + j_{\mathrm{H}} + 1, \\
\mathcal{B}: &\quad \Delta = j_{\mathrm{C}} + j_{\mathrm{H}}, \quad \ell = 0,
\end{aligned}
\tag{3.5}
$$

where $\mathcal{X} = \mathcal{L}, \mathcal{A}, \mathcal{B}$, and $\Delta, \ell \in \frac{\mathbb{Z}}{2}$, $j_{\mathrm{C}} \in \frac{\mathbb{Z}}{2}$, $j_{\mathrm{H}} \in \frac{\mathbb{Z}}{2}$ are the dimension, spin and R-charges of the highest weight state of $\mathcal{X}[2\ell]^{(2j_{\mathrm{C}};2j_{\mathrm{H}})}_{\Delta}$. The superconformal multiplets that will be important in this paper are summarized in Table 4.

The quadratic Casimir operator $C$ of $OSp(4|4)$ is constructed out of bilinears of the generators,

$$
\begin{aligned}
C &= C_b + \frac{1}{2}[S^{\alpha A\dot{A}},Q_{\alpha A\dot{A}}] - R^{\mathrm{C}}_I R^{\mathrm{C}}_I - R^{\mathrm{H}}_{\dot{I}} R^{\mathrm{H}}_{\dot{I}}, \\
C_b &= \frac{1}{2}M_{\mu\nu}M_{\mu\nu} - D^2 - \frac{1}{2}\{P_{\mu},K_{\mu}\},
\end{aligned}
\tag{3.6}
$$

and commutes with all the generators in the superconformal algebra. The Casimir operator acting on a highest weight operator $\mathcal{O}_{\Delta,\ell,j_{\mathrm{C}},j_{\mathrm{H}}}$ gives

$$
[C,\mathcal{O}_{\Delta,\ell,j_{\mathrm{C}},j_{\mathrm{H}}}] = \rho(\Delta,\ell,j_{\mathrm{C}},j_{\mathrm{H}})\mathcal{O}_{\Delta,\ell,j_{\mathrm{C}},j_{\mathrm{H}}},
\tag{3.7}
$$

where the eigenvalue $\rho(\Delta,\ell,j_{\mathrm{C}},j_{\mathrm{H}})$ is given by

$$
\begin{aligned}
\rho(\Delta,\ell,j_{\mathrm{C}},j_{\mathrm{H}}) &= \rho_b(\Delta,\ell,j_{\mathrm{C}},j_{\mathrm{H}}) + 4\Delta - j_{\mathrm{C}}(j_{\mathrm{C}} + 1) - j_{\mathrm{H}}(j_{\mathrm{H}} + 1), \\
\rho_b(\Delta,\ell,j_{\mathrm{C}},j_{\mathrm{H}}) &= \Delta(\Delta - 3) + \ell(\ell + 1).
\end{aligned}
\tag{3.8}
$$

Table 4: Short $\mathcal{N} = 4$ multiplets that contain conserved currents.

| Name | Alias | Superconformal primary | Other important primaries |
|---|---|---|---|
| $\mathcal{B}[0]_0^{(0;0)}$ | Identity multiplet | Identity operator | |
| $\mathcal{B}[0]_1^{(2;0)}$ $\mathcal{B}[0]_1^{(0;2)}$ | Flavor current multiplet | Moment map operator | Flavor current |
| $\mathcal{A}[0]_1^{(0;0)}$ | Stress tensor multiplet | Dimension-1 R-singlet scalar | R-symmetry current Stress tensor |
| $\mathcal{A}[2\ell]_{\ell+1}^{(0;0)}$ | Higher spin multiplet | spin-$\ell$ R-singlet current | Higher spin current |

## 3.2  $\mathbb{Z}_2$ outer automorphism

The 3d $\mathcal{N} = 4$ superconformal algebra has a $\mathbb{Z}_2$ outer automorphism that exchanges $SU(2)_C$ with $SU(2)_H$. This outer automorphism has a close relation to mirror symmetry of 3d $\mathcal{N} = 4$ theories. Mirror symmetry is a duality between two UV QFTs in the sense that they flow to the same IR CFT. The Coulomb and the Higgs branch R-symmetries in the UV are embedded in the opposite way into the IR R-symmetries. The outer automorphism is the action of mirror symmetry that exchanges which $SU(2)$ R-symmetry we call Coulomb, and the other Higgs. Note that the $\mathbb{Z}_2$ outer automorphism is generally not a global symmetry of the theory.

The supercharges $Q_{\alpha A \dot{A}}$ are in the $(\mathbf{2},\mathbf{1}) \oplus (\mathbf{1},\mathbf{2}) = \mathbf{4}$ representation of $SU(2)_C \times SU(2)_H = SO(4)_R$. We will use the index M for the $\mathbf{4}$ of $SO(4)_R$, and write the supercharge as $Q_{\alpha M}$. The stress tensor $T_{\mu\nu}$ is a conformal primary but a superconformal descendant of the bottom component scalar $\mathcal{O}_{1,0,0,0}$ of the multiplet $\mathcal{A}[0]_1^{(0;0)}$. More explicitly, they are related by

$$T_{\mu\nu} = \mathcal{N}\, \epsilon^{M_1 M_2 M_3 M_4} (\sigma_\mu)^{\alpha_1 \alpha_2} (\sigma_\nu)^{\beta_1 \beta_2} Q_{\alpha_1 M_1} Q_{\alpha_2 M_2} Q_{\beta_1 M_3} Q_{\beta_2 M_4} \mathcal{O}_{1,0,0,0}, \tag{3.9}$$

where $\mathcal{N}$ is a normalization constant. The $\mathbb{Z}_2$ outer automorphism exchanges $SU(2)_C$ with $SU(2)_H$, and hence is a reflection in $O(4) = SO(4)_R \rtimes \mathbb{Z}_2$. In particular, the $\mathbb{Z}_2$ outer automorphism flips the sign of $\epsilon^{M_1 M_2 M_3 M_4}$. This implies that in a self-mirror theory where this $\mathbb{Z}_2$ is a true global symmetry, the superconformal primary $\mathcal{O}_{1,0,0,0}$ of the stress tensor multiplet is $\mathbb{Z}_2$ odd, while the stress tensor itself $T_{\mu\nu}$ is of course $\mathbb{Z}_2$ even.[21]

## 3.3  Single-branch superconformal blocks

The moment map operators in $\mathcal{B}[0]_1^{(2;0)}$ form a spin-1 representation of $SU(2)_C$, and can be written as a rank-2 symmetric traceless tensor in spinor indices, $\mathcal{O}_C^{AB}(x)$. It contracts with auxiliary $SU(2)_C$ spinors $Y^A$ as $\mathcal{O}_C(x, Y) = \mathcal{O}_C^{AB}(x) Y_A Y_B$. The four-point function of $\mathcal{O}_C(x, Y)$ is a homogeneous function of degree $(-4, 8)$,

$$\left\langle \mathcal{O}_C(x_1, Y_1) \mathcal{O}_C(x_2, Y_2) \mathcal{O}_C(x_3, Y_3) \mathcal{O}_C(x_4, Y_4) \right\rangle = \frac{(Y_1 \cdot Y_2)^2 (Y_3 \cdot Y_4)^2}{x_{12}^2 x_{34}^2} G_C(u, v; w), \tag{3.10}$$

where the cross ratios $u$, $v$ and $w$ are defined as

$$u = \frac{|x_{12}|^2 |x_{34}|^2}{|x_{13}|^2 |x_{24}|^2}, \quad v = \frac{|x_{14}|^2 |x_{23}|^2}{|x_{13}|^2 |x_{24}|^2}, \quad w = \frac{(Y_1 \cdot Y_2)(Y_3 \cdot Y_4)}{(Y_1 \cdot Y_4)(Y_2 \cdot Y_3)}. \tag{3.11}$$

---

[21]One can verify this in $\mathcal{N} = 8$ theories where the $\mathbb{Z}_2$ self-mirror symmetry is embedded in the $SO(8)_R$ symmetry.

The function $G(u, v; w)$ can be expanded in terms of the single-branch superconformal blocks as

$$G_C(u, v, w) = \sum_{\mathcal{X}=\mathcal{L},\mathcal{A},\mathcal{B}} \sum_{\Delta,\ell,j_C} \lambda^2_{\mathcal{X}[2\ell]^{(2j_C;0)}_\Delta} \mathcal{A}_{\mathcal{X}[2\ell]^{(2j_C;0)}_\Delta}(u, v, w). \tag{3.12}$$

Similarly, the moment map operators in $\mathcal{B}[0]^{(0;2)}_1$ are denoted by $\mathcal{O}_H(x, \widetilde{Y})$, where $\widetilde{Y}^{\dot{A}}$ is an auxiliary $SU(2)_H$ spinor. The four-point function of $\mathcal{O}_H(x, \widetilde{Y})$ admits a similar single-branch superconformal block expansion as

$$\langle \mathcal{O}_H(x_1, \widetilde{Y}_1)\mathcal{O}_H(x_2, \widetilde{Y}_2)\mathcal{O}_H(x_3, \widetilde{Y}_3)\mathcal{O}_H(x_4, \widetilde{Y}_4) \rangle = \frac{(\widetilde{Y}_1 \cdot \widetilde{Y}_2)^2(\widetilde{Y}_3 \cdot \widetilde{Y}_4)^2}{x_{12}^2 x_{34}^2} G_H(u, v; \widetilde{w}),$$

$$G_H(u, v, \widetilde{w}) = \sum_{\mathcal{X}=\mathcal{L},\mathcal{A},\mathcal{B}} \sum_{\Delta,\ell,j_H} \lambda^2_{\mathcal{X}[2\ell]^{(0;2j_H;)}_\Delta} \mathcal{A}_{\mathcal{X}[2\ell]^{(0;2j_H)}_\Delta}(u, v, \widetilde{w}). \tag{3.13}$$

By the $\mathbb{Z}_2$ outer automorphism, the single-branch superconformal blocks $\mathcal{A}_{\mathcal{X}[2\ell]^{(2j;0)}_\Delta}$ and $\mathcal{A}_{\mathcal{X}[2\ell]^{(0;2j)}_\Delta}$ are the same functions, *i.e.*

$$\mathcal{A}_{\mathcal{X}[2\ell]^{(2j;0)}_\Delta}(u, v, w) = \mathcal{A}_{\mathcal{X}[2\ell]^{(0;2j)}_\Delta}(u, v, w) \equiv \mathcal{A}_{\mathcal{X}[2\ell]^{2j}_\Delta}(u, v, w), \tag{3.14}$$

and they were originally computed in [25, 44].

## 3.4 Mixed-branch superconformal blocks

Now, consider the four-point function of two $\mathcal{O}^C$ and two $\mathcal{O}^H$ operators,

$$\langle \mathcal{O}^C(x_1, Y_1)\mathcal{O}^C(x_2, Y_2)\mathcal{O}^H(x, \widetilde{Y}_3)\mathcal{O}^H(x, \widetilde{Y}_4) \rangle$$

$$= \left( \frac{(Y_1 \cdot Y_2)(\widetilde{Y}_3 \cdot \widetilde{Y}_4)}{|x_{12}||x_{34}|} \right)^2 G_s(u, v) = \left( \frac{(Y_2 \cdot Y_3)(\widetilde{Y}_1 \cdot \widetilde{Y}_4)}{|x_{23}||x_{14}|} \right)^2 G_t(v, u), \tag{3.15}$$

which can be expanded in terms of either the $s$-channel mixed-branch superconformal blocks or the $t$-channel mixed-branch superconformal blocks

$$G_s(u, v) = \sum_{\mathcal{X}=\mathcal{L},\mathcal{A},\mathcal{B}} \sum_{\Delta,\ell,j_C,j_H} \lambda_{C,C,\mathcal{X}[2\ell]^{(2j_C;2j_H)}_\Delta} \lambda_{H,H,\mathcal{X}[2\ell]^{(0;0)}_\Delta} \mathcal{A}^s_{\mathcal{X}[2\ell]^{(0;0)}_\Delta}(u, v),$$

$$G_t(u, v) = \sum_{\mathcal{X}=\mathcal{L},\mathcal{A},\mathcal{B}} \sum_{\Delta,\ell,j_C,j_H} \lambda^2_{C,H,\mathcal{X}[2\ell]^{(2j_C;2j_H)}_\Delta} \mathcal{A}^t_{\mathcal{X}[2\ell]^{(2j_C;2j_H)}_\Delta}(u, v). \tag{3.16}$$

The $s$- and $t$-channel mixed-branch superconformal blocks can be computed by solving the super-Casimir equations. We first strip off the auxiliary variables $Y$ and $\widetilde{Y}$ in (3.15), and find

$$\langle \mathcal{O}^C_{++}(x_1)\mathcal{O}^C_{--}(x_2)\mathcal{O}^H_{++}(x_3)\mathcal{O}^H_{--}(x_4) \rangle = \frac{1}{|x_{12}|^2|x_{34}|^2} G_s(u, v) = \frac{1}{|x_{23}|^2|x_{14}|^2} G_t(v, u). \tag{3.17}$$

The commutators of the Casimir operator (3.6) with $\mathcal{O}^C_{++}\mathcal{O}^C_{--}$ and $\mathcal{O}^H_{++}\mathcal{O}^C_{--}$ give differential operators acting on the functions $G_s(u, v)$ and $G_t(u, v)$,

$$\langle [C, \mathcal{O}^C_{++}(x_1)\mathcal{O}^C_{--}(x_2)]\mathcal{O}^H_{++}(x_3)\mathcal{O}^H_{--}(x_4) \rangle = \frac{1}{|x_{12}|^2|x_{34}|^2} \mathcal{C}^s G_s(u, v),$$

$$\langle [C, \mathcal{O}^H_{++}(x_1)\mathcal{O}^C_{--}(x_2)]\mathcal{O}^C_{++}(x_3)\mathcal{O}^H_{--}(x_4) \rangle = \frac{1}{|x_{12}|^2|x_{34}|^2} \mathcal{C}^t G_t(u, v). \tag{3.18}$$

Expanding the above equations in terms of the mixed-branch superconformal blocks gives the Casimir equations

$$
\begin{aligned}
\mathcal{C}^s \mathcal{A}^s_{\mathcal{X}[2\ell]_\Delta^{(2j_C;2j_H)}}(u,v) &= \rho(\Delta,\ell,j_C,j_H)\mathcal{A}^s_{\mathcal{X}[2\ell]_\Delta^{(2j_C;2j_H)}}(u,v), \\
\mathcal{C}^t \mathcal{A}^t_{\mathcal{X}[2\ell]_\Delta^{(2j_C;2j_H)}}(u,v) &= \rho(\Delta,\ell,j_C,j_H)\mathcal{A}^s_{\mathcal{X}[2\ell]_\Delta^{(2j_C;2j_H)}}(u,v).
\end{aligned}
\tag{3.19}
$$

The differential operators $\mathcal{C}^s$ and $\mathcal{C}^t$ are computed in Appendix B. We solve the above Casimir equations by expanding the mixed-branch superconformal blocks in terms of the $\mathcal{N}=2$ superconformal blocks, *i.e.*

$$
\begin{aligned}
\mathcal{A}^s_{\mathcal{X}[2\ell]_\Delta^{(2j_C;2j_H)}} =\; & f^s_{0,0}\mathcal{G}^{\mathcal{N}=2}_{\Delta,\ell} + f^s_{\frac{1}{2},-\frac{1}{2}}\mathcal{G}^{\mathcal{N}=2}_{\Delta+\frac{1}{2},\ell-\frac{1}{2}} + f^s_{\frac{1}{2},\frac{1}{2}}\mathcal{G}^{\mathcal{N}=2}_{\Delta+\frac{1}{2},\ell+\frac{1}{2}} \\
& + f^s_{1,-1}\mathcal{G}^{\mathcal{N}=2}_{\Delta+1,\ell-1} + f^s_{1,0}\mathcal{G}^{\mathcal{N}=2}_{\Delta+1,\ell} + f^s_{1,1}\mathcal{G}^{\mathcal{N}=2}_{\Delta+1,\ell+1} \\
& + f^s_{\frac{3}{2},-\frac{1}{2}}\mathcal{G}^{\mathcal{N}=2}_{\Delta+\frac{3}{2},\ell-\frac{1}{2}} + f^s_{\frac{3}{2},\frac{1}{2}}\mathcal{G}^{\mathcal{N}=2}_{\Delta+\frac{3}{2},\ell+\frac{1}{2}} + f^s_{2,0}\mathcal{G}^{\mathcal{N}=2}_{\Delta+2,\ell}, \\
\mathcal{A}^t_{\mathcal{X}[2\ell]_\Delta^{(2j_C;2j_H)}} =\; & f^t_{0,0}\mathcal{G}^{\mathcal{N}=2}_{\Delta,\ell} + f^t_{\frac{1}{2},-\frac{1}{2}}\mathcal{G}^{\mathcal{N}=2}_{\Delta+\frac{1}{2},\ell-\frac{1}{2}} + f^t_{\frac{1}{2},\frac{1}{2}}\mathcal{G}^{\mathcal{N}=2}_{\Delta+\frac{1}{2},\ell+\frac{1}{2}} \\
& + f^t_{1,-1}\mathcal{G}^{\mathcal{N}=2}_{\Delta+1,\ell-1} + f^t_{1,0}\mathcal{G}^{\mathcal{N}=2}_{\Delta+1,\ell} + f^t_{1,1}\mathcal{G}^{\mathcal{N}=2}_{\Delta+1,\ell+1} \\
& + f^t_{\frac{3}{2},-\frac{1}{2}}\mathcal{G}^{\mathcal{N}=2}_{\Delta+\frac{3}{2},\ell-\frac{1}{2}} + f^t_{\frac{3}{2},\frac{1}{2}}\mathcal{G}^{\mathcal{N}=2}_{\Delta+\frac{3}{2},\ell+\frac{1}{2}} + f^t_{2,0}\mathcal{G}^{\mathcal{N}=2}_{\Delta+2,\ell},
\end{aligned}
\tag{3.20}
$$

where the $\mathcal{N}=2$ superconformal blocks $\mathcal{G}^{\mathcal{N}=2}_{\Delta,\ell}$ were computed in [79]. Their expansion in terms of the bosonic conformal blocks is given by

$$
\mathcal{G}^{\mathcal{N}=2}_{\Delta,\ell} = \mathcal{G}_{\Delta,\ell} + f^{\mathcal{N}=2}_{1,1}\mathcal{G}_{\Delta+1,\ell+1} + f^{\mathcal{N}=2}_{1,-1}\mathcal{G}_{\Delta+1,\ell-1} + f^{\mathcal{N}=2}_{2,0}\mathcal{G}_{\Delta+2,\ell},
\tag{3.21}
$$

with the following coefficients[22]

$$
\begin{aligned}
f^{\mathcal{N}=2}_{1,1} &= \frac{(\ell+1)(\Delta+\ell)^2}{2(2\ell+1)(\Delta+\ell)(\Delta+\ell+1)}, \\
f^{\mathcal{N}=2}_{1,-1} &= \frac{\ell(\Delta-\ell-1)^2}{2(2\ell+1)(\Delta-\ell-1)(\Delta-\ell)}, \\
f^{\mathcal{N}=2}_{2,0} &= \frac{\Delta^2(\Delta+\ell)^2(\Delta-\ell-1)^2}{4(2\Delta+1)(2\Delta-1)(\Delta+\ell)(\Delta+\ell+1)(\Delta-\ell-1)(\Delta-\ell)}.
\end{aligned}
\tag{3.22}
$$

The bosonic conformal blocks are normalized as

$$
\mathcal{G}_{\Delta,\ell}(u,v) \to \frac{(\epsilon)_\epsilon}{(\ell+\epsilon)_\epsilon} u^{\frac{\Delta-\ell}{2}}(1-v)^\ell
\tag{3.23}
$$

in the limit $u\to 0$ then $v\to 1$, where $(x)_n = \Gamma(x+n)/\Gamma(x)$ is the Pochhammer symbol and $\epsilon = (d-2)/2 = 1/2$.

### 3.4.1 $s$-channel

It is straightforward to solve the $s$-channel Casimir equation. The long multiplet block is given by

$$
\mathcal{A}^s_{\mathcal{L}[2\ell]_\Delta^{(0;0)}} = \mathcal{G}^{\mathcal{N}=2}_{\Delta,\ell} + f^s_{1,-1}\mathcal{G}^{\mathcal{N}=2}_{\Delta+1,\ell-1} + f^s_{1,1}\mathcal{G}^{\mathcal{N}=2}_{\Delta+1,\ell+1} + f^s_{2,0}\mathcal{G}^{\mathcal{N}=2}_{\Delta+2,\ell},
\tag{3.24}
$$

---

[22]Note that due to the difference in the normalization of the bosonic block, (3.23) versus (65) of [79], the coefficients (3.22) are different from the coefficients given by (67) of [79].

where the coefficients are

$$f_{1,1}^s = -\frac{(\ell+1)(\Delta+\ell)}{2(2\ell+1)(\Delta+\ell+1)}, \quad f_{1,-1}^s = -\frac{\ell(-\Delta+\ell+1)}{2(2\ell+1)(\ell-\Delta)},$$

$$f_{2,0}^s = \frac{(\Delta+1)^2(-\Delta+\ell+1)(\Delta+\ell)}{4(2\Delta+1)(2\Delta+3)(\ell-\Delta)(\Delta+\ell+1)}. \tag{3.25}$$

The short multiplet blocks are

$$\mathcal{A}^s_{\mathcal{A}[2\ell]_{\ell+1}^{(0;0)}} = \mathcal{G}^{\mathcal{N}=2}_{\ell+1,\ell} - \frac{1}{4}\mathcal{G}^{\mathcal{N}=2}_{\ell+2,\ell+1},$$

$$\mathcal{A}^s_{\mathcal{B}[0]_0^{(0;0)}} = 1. \tag{3.26}$$

### 3.4.2  $t$-channel

The $t$-channel Casimir equation admits the following classes of solutions,

$$\mathcal{A}^t_{\mathcal{X}[2\ell]_\Delta^{(2j_C;2j_H)}} = \mathcal{G}^{\mathcal{N}=2}_{\Delta,\ell} \qquad \text{with} \quad j_C(j_C+1)+j_H(j_H+1) = 2\Delta,$$

$$\mathcal{A}^t_{\mathcal{X}[2\ell]_\Delta^{(2j_C;2j_H)}} = \mathcal{G}^{\mathcal{N}=2}_{\Delta+\frac{1}{2},\ell+\frac{1}{2}} \quad \text{with} \quad j_C(j_C+1)+j_H(j_H+1) = \Delta-\ell-\frac{1}{2},$$

$$\mathcal{A}^t_{\mathcal{X}[2\ell]_\Delta^{(2j_C;2j_H)}} = \mathcal{G}^{\mathcal{N}=2}_{\Delta+\frac{1}{2},\ell-\frac{1}{2}} \quad \text{with} \quad j_C(j_C+1)+j_H(j_H+1) = \Delta+\ell+\frac{1}{2}, \tag{3.27}$$

$$\mathcal{A}^t_{\mathcal{X}[2\ell]_\Delta^{(2j_C;2j_H)}} = \mathcal{G}^{\mathcal{N}=2}_{\Delta+1,\ell} \qquad \text{with} \quad j_C(j_C+1)+j_H(j_H+1) = 0,$$

$$\mathcal{A}^t_{\mathcal{X}[2\ell]_\Delta^{(2j_C;2j_H)}} = \mathcal{G}^{\mathcal{N}=2}_{\Delta+1,\ell-1} \quad \text{with} \quad j_C(j_C+1)+j_H(j_H+1) = 2\ell.$$

There are several constraints on the possible superconformal blocks. First, the dimension, spin and R-charges are constrained by the unitarity bounds (3.5). Second, by the Lorentz and R-symmetry selection rules, the operators that appear in the $\mathcal{O}^H_{++} \times \mathcal{O}^C_{--}$ OPE must have $j_C = -j_H = 1$, and $\ell \in \mathbb{Z}$. Hence, they must all be superconformal multiplets $\mathcal{X}[2\ell]_\Delta^{(2j_C;2j_H)}$ with $\ell$, $j_C$ and $j_H$ being all integers or all half integers. Third, the superconformal primaries that appear in the $\mathcal{O}^H_{++} \times \mathcal{O}^C_{--}$ OPE must be in the $(j_C, j_H) = (1,1)$ representation of $SU(2)_C \times SU(2)_H$; hence, the first line of (3.27) can only take value $\Delta = 2$ and $\ell = 0$.

The superconformal block belonging to the first line of (3.27) is

$$\mathcal{A}^t_{\mathcal{B}[0]_2^{(2;2)}} = \mathcal{G}^{\mathcal{N}=2}_{2,0}. \tag{3.28}$$

The superconformal blocks belonging to the second line of (3.27) are

$$\mathcal{A}^t_{\mathcal{A}[2\ell]_{\ell+2}^{(1;1)}} = \mathcal{G}^{\mathcal{N}=2}_{\ell+\frac{5}{2},\ell+\frac{1}{2}} \qquad \text{with} \quad \ell \in \mathbb{Z}_{\geq 0} + \frac{1}{2},$$

$$\mathcal{A}^t_{\mathcal{L}[2\ell]_{\ell+4}^{(3;1)}} = \mathcal{A}^t_{\mathcal{L}[2\ell]_{\ell+4}^{(1;3)}} = \mathcal{G}^{\mathcal{N}=2}_{\ell+\frac{11}{2},\ell+\frac{1}{2}} \quad \text{with} \quad \ell \in \mathbb{Z}_{\geq 0} + \frac{1}{2}, \tag{3.29}$$

$$\mathcal{A}^t_{\mathcal{L}[2\ell]_{\ell+8}^{(3;3)}} = \mathcal{G}^{\mathcal{N}=2}_{\ell+\frac{17}{2},\ell+\frac{1}{2}} \qquad \text{with} \quad \ell \in \mathbb{Z}_{\geq 0} + \frac{1}{2}.$$

The superconformal blocks belonging to the third line of (3.27) are

$$\mathcal{A}^t_{\mathcal{A}[1]_{\frac{7}{2}}^{(3;1)}} = \mathcal{A}^t_{\mathcal{A}[1]_{\frac{7}{2}}^{(1;3)}} = \mathcal{G}^{\mathcal{N}=2}_{4,0},$$

$$\mathcal{A}^t_{\mathcal{A}[3]_{\frac{11}{2}}^{(3;3)}} = \mathcal{G}^{\mathcal{N}=2}_{6,1}, \tag{3.30}$$

$$\mathcal{A}^t_{\mathcal{L}[1]_{\frac{13}{2}}^{(3;3)}} = \mathcal{G}^{\mathcal{N}=2}_{7,0}.$$

Finally, the superconformal blocks belonging to the fourth line of (3.27) are

$$
\begin{aligned}
\mathcal{A}^t_{\mathcal{A}[2\ell]^{(0;0)}_{\ell+1}} &= \mathcal{G}^{\mathcal{N}=2}_{\ell+2,\ell}, \\
\mathcal{A}^t_{\mathcal{A}[2]^{(2;0)}_3} = \mathcal{A}^t_{\mathcal{A}[2]^{(0;2)}_3} &= \mathcal{G}^{\mathcal{N}=2}_{4,0}, \\
\mathcal{A}^t_{\mathcal{A}[4]^{(2;2)}_5} &= \mathcal{G}^{\mathcal{N}=2}_{6,1}, \\
\mathcal{A}^t_{\mathcal{A}[6]^{(4;0)}_6} = \mathcal{A}^t_{\mathcal{A}[6]^{(0;4)}_6} &= \mathcal{G}^{\mathcal{N}=2}_{7,2}, \\
\mathcal{A}^t_{\mathcal{A}[8]^{(4;2)}_8} = \mathcal{A}^t_{\mathcal{A}[8]^{(2;4)}_8} &= \mathcal{G}^{\mathcal{N}=2}_{9,3}, \\
\mathcal{A}^t_{\mathcal{A}[12]^{(4;4)}_{11}} &= \mathcal{G}^{\mathcal{N}=2}_{12,5}, \\
\mathcal{A}^t_{\mathcal{L}[2\ell]^{(0;0)}_\Delta} &= \mathcal{G}^{\mathcal{N}=2}_{\Delta+1,\ell} \quad \text{with} \quad \Delta > \ell+1,\, \ell \in \mathbb{Z}_{\geq 0}, \\
\mathcal{A}^t_{\mathcal{L}[2]^{(2;0)}_\Delta} = \mathcal{A}^t_{\mathcal{L}[2]^{(0;2)}_\Delta} &= \mathcal{G}^{\mathcal{N}=2}_{\Delta+1,0} \quad \text{with} \quad \Delta > 3, \\
\mathcal{A}^t_{\mathcal{L}[4]^{(2;2)}_\Delta} &= \mathcal{G}^{\mathcal{N}=2}_{\Delta+1,1} \quad \text{with} \quad \Delta > 5, \\
\mathcal{A}^t_{\mathcal{L}[6]^{(4;0)}_\Delta} = \mathcal{A}^t_{\mathcal{L}[6]^{(0;4)}_\Delta} &= \mathcal{G}^{\mathcal{N}=2}_{\Delta+1,2} \quad \text{with} \quad \Delta > 6, \\
\mathcal{A}^t_{\mathcal{L}[8]^{(4;2)}_\Delta} = \mathcal{A}^t_{\mathcal{L}[8]^{(2;4)}_\Delta} &= \mathcal{G}^{\mathcal{N}=2}_{\Delta+1,3} \quad \text{with} \quad \Delta > 8, \\
\mathcal{A}^t_{\mathcal{L}[12]^{(4;4)}_\Delta} &= \mathcal{G}^{\mathcal{N}=2}_{\Delta+1,5} \quad \text{with} \quad \Delta > 11.
\end{aligned}
\tag{3.31}
$$

# 4 The protected sector

In [22, 31, 32], the authors showed that every three-dimensional $\mathcal{N} = 4$ SCFT admits a protected sector as a topological quantum mechanics (TQM), which lives on a straight line in the three dimensional Euclidean space $\mathbb{R}^3$. The operators in the TQM are either the Coulomb or Higgs branch chiral ring operators with suitable twisting, where the translation along the straight line is accompanied with certain $SU(2)_\mathrm{C}$ or $SU(2)_\mathrm{H}$ rotations. The correlation functions of these operators depend only on the ordering of the operators on the straight line but not their positions. This implies that the OPE along the straight line forms an associative algebra, which is called the protected associative algebra. Crossing symmetry of the four-point function amounts to associativity of the protected associative algebra.

The multiplication of the Coulomb or Higgs branch chiral ring operators in the protected associative algebra is a non-commutative deformation of the commutative chiral ring multiplication. It was observed in [31] that the leading order deformation is determined by the Poisson bracket of the chiral ring; hence, the protected associative algebra is a deformation quantization of the chiral ring.

When the Coulomb or Higgs branch of a given theory coincides with the minimal nilpotent orbit of a complex simple Lie algebra $\mathfrak{g}$ (except $\mathfrak{g} = \mathfrak{sl}_2$), the protected associative algebra is unique [31]. We will study these theories by two equivalent approaches in Sections 4.1 and 4.2. In Section 4.1, we consider the crossing equations of the four-point function of the moment map operators in the TQM, and derive analytical bounds on the flavor central charges and other OPE coefficients. We dub this approach the "mini-bootstrap". We find that the minimal nilpotent theories sit at the kinks of the allowed regions, where the values of the charges nicely agree with the ones computed in Section 2 and Appendix A by localization. In Section 4.2, we study the deformation quantization of the chiral ring of the minimal nilpotent theories. Using associativity, we fix the coefficients in the protected associative algebra up to quadratic order. We also compute the four-point function of the moment map operators using the protected

associative algebra, and find agreement with the four-point function computed using mini-bootstrap.

## 4.1 Mini-bootstrap

Consider a three dimensional $\mathcal{N} = 4$ SCFT with a simple flavor group $G$, which is generated by the flavor currents in the flavor current multiplets. Without loss of generality, we assume these flavor current multiplets are of type $\mathcal{B}[0]_1^{(2;0)}$ (as opposed to $\mathcal{B}[0]_1^{(0;2)}$). The moment map operators $\mathcal{O}_a^C(x, Y)$ are in the adjoint representation of the flavor group $G$, and have the four-point function

$$\left\langle \mathcal{O}_a^C(x_1, Y_1) \mathcal{O}_b^C(x_2, Y_2) \mathcal{O}_c^C(x_3, Y_3) \mathcal{O}_d^C(x_4, Y_4) \right\rangle = \frac{(Y_1 \cdot Y_2)^2 (Y_3 \cdot Y_4)^2}{x_{12}^2 x_{34}^2} G_{abcd}(u, v; w). \quad (4.1)$$

The four-point function simplifies when all the four-points are along a straight line and with a suitable twisting by the R-symmetry rotation. More precisely, we consider

$$\mathcal{O}_a^C(s) \equiv \mathcal{O}_a^C(x = (0, 0, s), Y = (1, s/2r)), \quad (4.2)$$

so that the four-point function becomes

$$\left\langle \mathcal{O}_a^C(s_1) \mathcal{O}_b^C(s_2) \mathcal{O}_c^C(s_3) \mathcal{O}_d^C(s_4) \right\rangle = G_{abcd}(u_w, v_w; w). \quad (4.3)$$

In the above, $r$ is a dimensionful parameter to make the combination $\frac{s}{2r}$ dimensionless; we define

$$Y_i \cdot Y_j = \frac{1}{2r}(s_i - s_j), \quad w = \frac{s_{12}s_{34}}{s_{14}s_{23}}, \quad u = u_w \equiv \frac{w^2}{(1+w)^2}, \quad v = v_w \equiv \frac{1}{(1+w)^2}. \quad (4.4)$$

The function $G_{abcd}(u_w, v_w; w)$ has a very simple superconformal block expansion. In general, the fusion of two flavor current multiplets $\mathcal{B}[0]_1^{(2;0)}$ gives

$$\mathcal{B}[0]_1^{(2;0)} \times \mathcal{B}[0]_1^{(2;0)} = \mathcal{B}[0]_0^{(0;0)} + \mathcal{B}[0]_1^{(2;0)} + \mathcal{B}[0]_2^{(4;0)} + \sum_{\ell=0}^{\infty} \mathcal{A}[2\ell]_{\ell+1}^{(0;0)}$$
$$+ \sum_{\ell=0}^{\infty} \mathcal{A}[2\ell]_{\ell+2}^{(2;0)} + \mathcal{L}[2\ell]_{\Delta}^{(0;0)}. \quad (4.5)$$

However, in the particular configuration (4.3), the long multiplet blocks and the $\mathcal{A}$-type short multiplet blocks vanish identically, and the $\mathcal{B}$-type short multiplet blocks only depend on the ordering of the $s_i$. More explicitly, we have

$$G_{abcd}(u_w, v_w; w)$$
$$= P_{\mathbf{1}}^{abcd} \mathcal{A}_{\mathcal{B}[0]_0^{(0;0)}}(u_w, v_w; w) + P_{\mathbf{adj}}^{abcd} \lambda_{\mathcal{B}[0]_1^{(2;0)}, \mathbf{adj}}^2 \mathcal{A}_{\mathcal{B}[0]_1^{(2;0)}}(u_w, v_w; w)$$
$$+ \left( P_{\mathbf{1}}^{abcd} \lambda_{\mathcal{B}[0]_2^{(4;0)}, \mathbf{1}}^2 + P_{\mathbf{2adj}}^{abcd} \lambda_{\mathcal{B}[0]_2^{(4;0)}, \mathbf{2adj}}^2 \right.$$
$$\left. + \sum_i P_{\mathcal{R}_{(S,i)}}^{abcd} \lambda_{\mathcal{B}[0]_2^{(4;0)}, \mathcal{R}_{(S,i)}}^2 \right) \mathcal{A}_{\mathcal{B}[0]_2^{(4;0)}}(u_w, v_w; w) \quad (4.6)$$
$$= \delta^{ab} \delta^{cd} - 2 P_{\mathbf{adj}}^{abcd} \lambda_{\mathcal{B}[0]_1^{(2;0)}, \mathbf{adj}}^2 \operatorname{sgn}(s_{12} s_{34} s_{13} s_{24})$$
$$+ 6 \left( P_{\mathbf{1}}^{abcd} \lambda_{\mathcal{B}[0]_2^{(4;0)}, \mathbf{1}}^2 + P_{\mathbf{2adj}}^{abcd} \lambda_{\mathcal{B}[0]_2^{(4;0)}, \mathbf{2adj}}^2 + \sum_i P_{\mathcal{R}_{(S,i)}}^{abcd} \lambda_{\mathcal{B}[0]_2^{(4;0)}, \mathcal{R}_{(S,i)}}^2 \right),$$

where the $P_{\mathbf{adj}}^{abcd}$, $P_{\mathbf{2adj}}^{abcd}$ and $P_{\mathcal{R}_{(S,i)}}^{abcd}$ are projection matrices defined in Appendix C, and we have assumed that in the $\mathcal{O}_a^C \times \mathcal{O}_b^C$ OPE, the identity multiplet must be in the trivial representation of the flavor group, and the flavor current multiplets must be in the adjoint representation.

Defining

$$G_{\mathcal{R}_i}(u,v;w) = \frac{1}{\dim(\mathcal{R}_i)} P_{\mathcal{R}_i}^{dcba} G_{abcd}(u,v;w), \tag{4.7}$$

the crossing equation of the four-point function (4.1) can be written as

$$F_i{}^j G_{\mathcal{R}_j}(u,v;w) = \frac{u}{vw^2} G_{\mathcal{R}_i}(v,u;w^{-1}), \tag{4.8}$$

where $F_i{}^j$ is the crossing matrix defined in Appendix C. After the specialization (4.2), the bootstrap equation reduces to

$$F_i{}^j G_{\mathcal{R}_j} = G_{\mathcal{R}_i}, \tag{4.9}$$

where the $G_{\mathcal{R}_i}$ can be computed by use of (4.6) and (4.7),

$$G_{\mathbf{1}} = \dim(G) + 6\lambda^2_{\mathcal{B}[0]_2^{(4;0)},\mathbf{1}}, \quad G_{\mathbf{adj}} = -2\lambda^2_{\mathcal{B}[0]_1^{(2;0)},\mathbf{adj}}, \quad G_{\mathbf{2adj}} = 6\lambda^2_{\mathcal{B}[0]_2^{(4;0)},\mathbf{2adj}},$$

$$G_{\mathcal{R}_{(S,i)}} = 6\lambda^2_{\mathcal{B}[0]_2^{(4;0)},\mathcal{R}_{(S,i)}}, \quad G_{\mathcal{R}_{(A,i)}} = 0. \tag{4.10}$$

For simple Lie groups, we find that the coefficients $\lambda^2_{\mathcal{B}[0]_2^{(4;0)},\mathbf{1}}$ and $\lambda^2_{\mathcal{B}[0]_2^{(4;0)},\mathcal{R}_{(S,i)}}$ are determined by the coefficients $\lambda^2_{\mathcal{B}[0]_1^{(2;0)},\mathbf{adj}}$ and $\lambda^2_{\mathcal{B}[0]_2^{(4;0)},\mathbf{2adj}}$ via the crossing equation (4.9). Unitarity gives the positivity conditions

$$\lambda^2_{\mathcal{B}[0]_2^{(4;0)},\mathbf{1}}, \ \lambda^2_{\mathcal{B}[0]_2^{(4;0)},\mathcal{R}_{(S,i)}} \geq 0, \tag{4.11}$$

which give bounds on $\lambda^2_{\mathcal{B}[0]_1^{(2;0)},\mathbf{adj}}$ and $\lambda^2_{\mathcal{B}[0]_2^{(4;0)},\mathbf{2adj}}$. The bounds for flavor groups $G = SU(2)$, $SU(3)$, $G_2$, and $E_6$ are given in Figure 2, where the shaded region is ruled out.

The maximal nilpotent orbit condition is

$$\lambda^2_{\mathcal{B}[0]_2^{(4;0)},\mathbf{1}} = 0, \tag{4.12}$$

which combines with the positivity conditions (4.11) gives a line segment in the $\lambda^2_{\mathcal{B}[0]_1^{(2;0)},\mathbf{adj}}$-$\lambda^2_{\mathcal{B}[0]_2^{(4;0)},\mathbf{2adj}}$ plane. As discussed in Section 2, the $T[G]$ theories are examples of theories on this line segment.

The minimal nilpotent orbit condition is

$$\lambda^2_{\mathcal{B}[0]_2^{(4;0)},\mathbf{1}} = \lambda^2_{\mathcal{B}[0]_2^{(4;0)},\mathcal{R}_{(S,i)}} = 0. \tag{4.13}$$

For $G = SU(2)$, the equation (4.9) and the conditions (4.13) give a linear relation

$$\lambda^2_{\mathcal{B}[0]_2^{(4;0)},\mathbf{2adj}} = \frac{1}{5}\left(1 + \lambda^2_{\mathcal{B}[0]_1^{(2;0)},\mathbf{adj}}\right). \tag{4.14}$$

As discussed in Section 2, the examples of theories on this line are the $\mathbb{Z}_2$ gauge field coupled to a free hypermultiplet, the SQED with $N_f = 2$, and the $U(2)_2 \times U(1)_{-2}$ ABJ theory. For other simple Lie groups, the equation (4.9) and the conditions (4.13) uniquely determine $\lambda^2_{\mathcal{B}[0]_1^{(2;0)},\mathbf{adj}}$ and $\lambda^2_{\mathcal{B}[0]_2^{(4;0)},\mathbf{2adj}}$. The results are given in Table 5. The OPE coefficient $\lambda^2_{\mathcal{B}[0]_1^{(2;0)},\mathbf{adj}}$ is related to the flavor central charge $C_J$ by a general formula

$$\lambda^2_{\mathcal{B}[0]_1^{(2;0)},\mathbf{adj}} = \frac{8h^\vee}{C_J}. \tag{4.15}$$

Table 5: Values of the flavor central charge $C_J$ and the OPE coefficient $\lambda^2_{\mathcal{B}[0]^{(4;0)}_2,\mathbf{2adj}}$ fixed by the minimal nilpotent orbit condition and solved by mini-bootstrap.

| $G$ | $A_{n-1}, n \geq 3$ | $B_n$ | $C_n$ | $D_n$ | $E_6$ | $E_7$ | $E_8$ | $F_4$ | $G_2$ |
|---|---|---|---|---|---|---|---|---|---|
| $C_J$ | $\frac{8n}{n+1}$ | $\frac{8(2n-3)}{n}$ | $4$ | $\frac{32(n-2)}{2n-1}$ | $\frac{192}{13}$ | $\frac{384}{19}$ | $\frac{960}{31}$ | $12$ | $\frac{64}{9}$ |
| $\lambda^2_{\mathcal{B}[0]^{(4;0)}_2,\mathbf{2adj}}$ | $\frac{2(n+1)(n+2)}{3n(n+3)}$ | $\frac{n(2n-1)}{2(n+1)(2n-3)}$ | $1$ | $\frac{2n^2-3n+1}{4n^2-6n-4}$ | $\frac{13}{27}$ | $\frac{95}{216}$ | $\frac{217}{540}$ | $\frac{14}{27}$ | $\frac{2}{3}$ |

## 4.2 Deformation quantization

Let $v^a$ be a vector in the adjoint representation of a simple Lie algebra $\mathfrak{g}$. The coordinate ring $\mathcal{A}$ of the minimal nilpotent orbit of $\mathfrak{g}$ is

$$\mathcal{A} = \mathbb{C}[v^a]/\{v^a v^a = 0, \; v^a v^b (\Gamma_{\mathcal{R}_{(S,i)}})^{I_{(S,i)}}_{ab} = 0\}, \tag{4.16}$$

where $\mathcal{R}_{(S,i)}$ are the representations appeared in the symmetric tensor product $(\mathbf{adj} \otimes \mathbf{adj})_S$ that are not $\mathbf{2adj}$, and $(\Gamma_{\mathcal{R}_{(S,i)}})^{I_{(S,i)}}_{ab}$ are the associated Clebsch-Gordan coefficients. The Poisson bracket on the minimal nilpotent orbit is

$$\{v^a, v^b\}_{\mathrm{PB}} = f^{abc} v^c. \tag{4.17}$$

We consider the deformation quantization algebra of the chiral ring $\mathcal{A}$. The algebra is a commutative ring $\mathcal{A}[r^{-1}]$ equipped with an associative star product

$$\star : \mathcal{A}[r^{-1}] \otimes \mathcal{A}[r^{-1}] \to \mathcal{A}[r^{-1}]. \tag{4.18}$$

The star product satisfies the conditions

$$[f(v), g(v)]_\star = \frac{1}{2r} \{f(v), g(v)\}_{\mathrm{PB}},$$
$$f(v) \star g(v) = f(v)g(v) + \frac{1}{4r} \{f(v), g(v)\}_{\mathrm{PB}} + \mathcal{O}(r^{-2}). \tag{4.19}$$

The star product of two $v^a$ takes the general form as

$$v^a \star v^b = (\Gamma_{\mathbf{2adj}})^{ab}_\alpha v^\alpha + \frac{1}{4r} f^{abc} v^c + \frac{1}{4r^2} \lambda_2 \delta^{ab},$$
$$v^a \star v^\alpha = (\Gamma_{\mathbf{3adj}})^{a\alpha}_A v^A + \frac{1}{2r} (\Gamma_{\mathbf{2adj}})^{bc}_\alpha f^{abd} v^d v^c + \frac{1}{4r^2} \widetilde{\lambda}_2 (\Gamma_{\mathbf{2adj}})^{ab}_\alpha v^b, \tag{4.20}$$

where $\lambda_2$ and $\widetilde{\lambda}_2$ are coefficients to be determined, $v^\alpha$ and $v^A$ are defined by

$$v^\alpha \equiv (\Gamma_{\mathbf{2adj}})^\alpha_{ab} v^a v^b,$$
$$v^A \equiv (\Gamma_{\mathbf{3adj}})^A_{a\alpha} v^a v^\alpha, \tag{4.21}$$

and $(\Gamma_{\mathbf{3adj}})^A_{a\alpha}$ for $A = 1, \cdots \dim(\mathbf{3adj})$ are the Clebsch-Gordan coefficients of the decomposition $\mathbf{adj} \otimes \mathbf{2adj} \supset \mathbf{3adj}$ with the normalization

$$(\Gamma_{\mathbf{3adj}})^A_{a\alpha} (\Gamma_{\mathbf{3adj}})^B_{a\alpha} = \delta^{AB}. \tag{4.22}$$

Contracting the first equation of (4.20) with $(\Gamma_{\mathbf{2adj}})^\alpha_{ab}$, we find

$$v^\alpha = (\Gamma_{\mathbf{2adj}})^\alpha_{ab} v^a \star v^b. \tag{4.23}$$

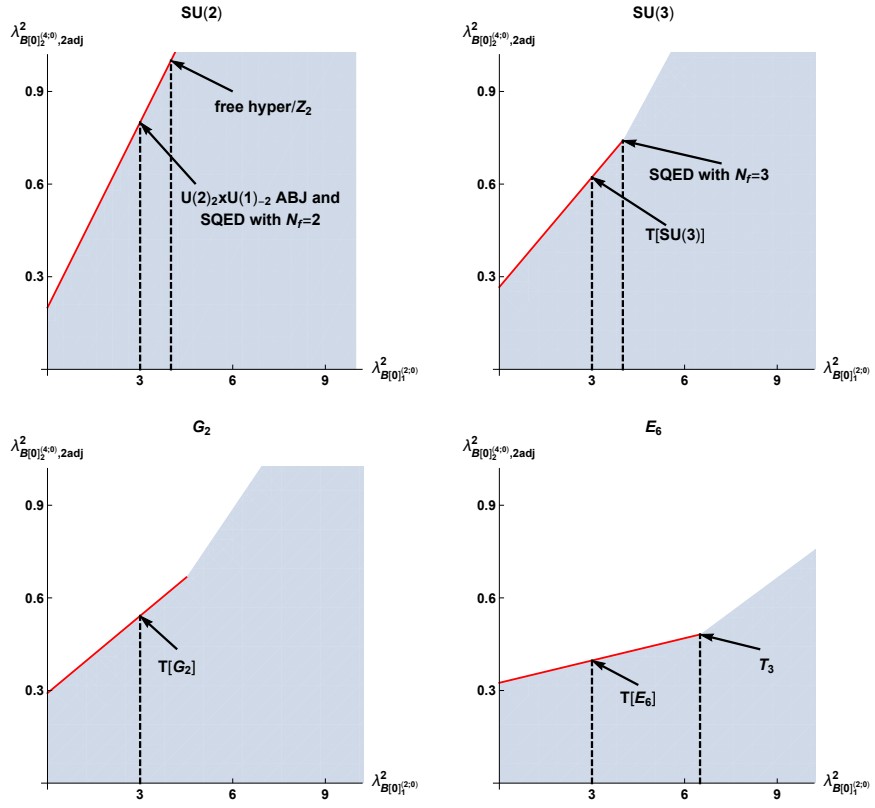

Figure 2: Analytic bounds from the mini-bootstrap equations (4.10) and the positivity conditions (4.11). The shaded regions are ruled out. The solid red lines are the maximal nilpotent condition (4.12). The dashed lines show the values of the OPE coefficient $\lambda^2_{\mathcal{B}[0]^{(2;0)}_1,\mathbf{adj}}$ determined via (4.15) by the central charges computed in Section 2 for various theories.

Let us try to fix $\lambda_2$ and $\widetilde{\lambda}_2$ by imposing associativity. Consider

$$
\begin{aligned}
v^a \star (v^b \star v^c) &= (v^a \star v^b) \star v^c \\
&= (\Gamma_{\mathbf{2adj}})^{ab}_\beta \left( (\Gamma_{\mathbf{3adj}})^{c\beta}_A v^A - \frac{1}{2r}(\Gamma_{\mathbf{2adj}})^{de}_\beta f^{cdf} v^f v^e + \frac{1}{4r^2}\widetilde{\lambda}_2 (\Gamma_{\mathbf{2adj}})^{cd}_\beta v^d \right) \\
&\quad + \frac{1}{4r} f^{abd} (\Gamma_{\mathbf{2adj}})^{dc}_\beta v^\beta + \frac{1}{16r^2} f^{abd} f^{dce} v^e + \frac{1}{4r^2}\lambda_2 \delta_{ab} v^c,
\end{aligned}
\tag{4.24}
$$

where we have used the commutators

$$
\begin{aligned}
[v^a, v^\alpha]_\star &= \frac{1}{2r}\{v^a, v^\alpha\} = \frac{1}{2r}(\Gamma_{\mathbf{2adj}})^\alpha_{bc}\{v^a, v^b v^c\} \\
&= \frac{1}{r}(\Gamma_{\mathbf{2adj}})^\alpha_{bc} f^{abd} v^d v^c = \frac{1}{r}(\Gamma_{\mathbf{2adj}})^\alpha_{bc} f^{abd} (\Gamma_{\mathbf{2adj}})^{dc}_\beta v^\beta.
\end{aligned}
\tag{4.25}
$$

Contracting (4.24) with the Clebsch-Gordan coefficients (C.4), the $\mathcal{O}(r^{-2})$ order of the equa-

tion gives

$$\widetilde{\lambda}_2(\Gamma_1)_{bc}(\Gamma_{\mathbf{2adj}})^{ab}_\beta(\Gamma_{\mathbf{2adj}})^{ce}_\beta + \frac{1}{4}(\Gamma_1)_{bc}f^{abd}f^{dce} + \lambda_2(\Gamma_1)_{ae} = \lambda_2\text{dim}(G)(\Gamma_1)_{ae},$$

$$\widetilde{\lambda}_2(\Gamma_{\mathbf{2adj}})^\alpha_{bc}(\Gamma_{\mathbf{2adj}})^{ab}_\beta(\Gamma_{\mathbf{2adj}})^{ce}_\beta + \frac{1}{4}(\Gamma_{\mathbf{2adj}})^\alpha_{bc}f^{abd}f^{dce} + \lambda_2(\Gamma_{\mathbf{2adj}})^\alpha_{ae} = \widetilde{\lambda}_2(\Gamma_{\mathbf{2adj}})^{ae}_\alpha,$$

$$\widetilde{\lambda}_2(\Gamma_{\mathcal{R}_{(S,i)}})^{I(S,i)}_{bc}(\Gamma_{\mathbf{2adj}})^{ab}_\beta(\Gamma_{\mathbf{2adj}})^{ce}_\beta + \frac{1}{4}(\Gamma_{\mathcal{R}_{(S,i)}})^{I(S,i)}_{bc}f^{abd}f^{dce} + \lambda_2(\Gamma_{\mathcal{R}_{(S,i)}})^{I(S,i)}_{ae} = 0, \quad (4.26)$$

$$\widetilde{\lambda}_2(\Gamma_{\mathbf{adj}})^f_{bc}(\Gamma_{\mathbf{2adj}})^{ab}_\beta(\Gamma_{\mathbf{2adj}})^{ce}_\beta + \frac{1}{4}(\Gamma_{\mathbf{adj}})^f_{bc}f^{abd}f^{dce} + \lambda_2(\Gamma_{\mathbf{adj}})^f_{ae} = \frac{1}{4}(\Gamma_{\mathbf{adj}})^f_{bc}f^{bcd}f^{ade},$$

$$\widetilde{\lambda}_2(\Gamma_{\mathcal{R}_{(A,i)}})^{I(A,i)}_{bc}(\Gamma_{\mathbf{2adj}})^{ab}_\beta(\Gamma_{\mathbf{2adj}})^{ce}_\beta + \frac{1}{4}(\Gamma_{\mathcal{R}_{(A,i)}})^{I(A,i)}_{bc}f^{abd}f^{dce} + \lambda_2(\Gamma_{\mathcal{R}_{(A,i)}})^{I(A,i)}_{ae} = 0,$$

which can be rewritten in terms of the crossing matrices defined in (C.6) as

$$\widetilde{\lambda}_2 F_{\mathbf{1}}{}^{\mathbf{2adj}} + \frac{h^\vee}{2}F_{\mathbf{1}}{}^{\mathbf{adj}} + \lambda_2 = \lambda_2\text{dim}(G),$$

$$\widetilde{\lambda}_2 F_{\mathbf{2adj}}{}^{\mathbf{2adj}} + \frac{h^\vee}{2}F_{\mathbf{2adj}}{}^{\mathbf{adj}} + \lambda_2 = \widetilde{\lambda}_2,$$

$$\widetilde{\lambda}_2 F_{\mathcal{R}_{(S,i)}}{}^{\mathbf{2adj}} + \frac{h^\vee}{2}F_{\mathcal{R}_{(S,i)}}{}^{\mathbf{adj}} + \lambda_2 = 0, \quad (4.27)$$

$$\widetilde{\lambda}_2 F_{\mathbf{adj}}{}^{\mathbf{2adj}} + \frac{h^\vee}{2}F_{\mathbf{adj}}{}^{\mathbf{adj}} + \lambda_2 = \frac{h^\vee}{2},$$

$$\widetilde{\lambda}_2 F_{\mathcal{R}_{(A,i)}}{}^{\mathbf{2adj}} + \frac{h^\vee}{2}F_{\mathcal{R}_{(A,i)}}{}^{\mathbf{adj}} + \lambda_2 = 0.$$

For SU(2), these equations reduce to

$$3 - 6\lambda_2 + 5\widetilde{\lambda}_2 = 0. \quad (4.28)$$

For the other simply Lie groups, $\lambda_2$ and $\widetilde{\lambda}_2$ are uniquely determined. The results are summarized in Table 6.

Table 6: The leading coefficients in the deformation quantization algebra of the chiral ring of the minimal nilpotent theories.

| $G$ | $A_{n-1}, n \geq 3$ | $B_n$ | $C_n$ | $D_n$ | $E_6$ | $E_7$ | $E_8$ | $F_4$ | $G_2$ |
|---|---|---|---|---|---|---|---|---|---|
| $\lambda_2$ | $-\frac{n}{4(n+1)}$ | $-\frac{2n-3}{4n}$ | $-\frac{1}{8}$ | $-\frac{n-2}{2n-1}$ | $-\frac{6}{13}$ | $-\frac{12}{19}$ | $-\frac{30}{31}$ | $-\frac{3}{8}$ | $-\frac{2}{9}$ |
| $\widetilde{\lambda}_2$ | $-\frac{n+2}{n+3}$ | $-\frac{3(2n-1)}{4(n+1)}$ | $-\frac{3}{4}$ | $-\frac{3n-3}{2n+1}$ | $-\frac{4}{3}$ | $-\frac{5}{3}$ | $-\frac{7}{3}$ | $-\frac{7}{6}$ | $-\frac{8}{9}$ |

For each homogeneous polynomial $p(v)$, there is an associated chiral ring operator $\mathcal{O}_{f(v)}$. The correlation functions of these operators in the topological quantum mechanics can be computed by taking the constant term of the start product as

$$\left\langle \mathcal{O}_{p_1(v)}(s_1)\cdots\mathcal{O}_{p_n(v)}(s_n)\right\rangle = (2r)^{\frac{1}{2}(d_1+\cdots+d_n)}\text{C.T.}(p_1(v)\star\cdots\star p_n(v)) \quad \text{with} \quad s_1 \geq \cdots \geq s_n, \quad (4.29)$$

where $d_i$ is the degree of the polynomial $p_i(v)$, and C.T.$(\cdots)$ denotes the constant term of $(\cdots)$. The moment map operators are given by

$$\mathcal{O}_a = \mathcal{O}_{\widetilde{v}^a}, \quad \widetilde{v}^a = \frac{1}{\sqrt{\lambda_2}}v^a. \quad (4.30)$$

Using the star products (4.20), we compute the four-point function of the moment map operator $\mathcal{O}_a$,

$$
\begin{aligned}
\langle \mathcal{O}_a(s_1)\mathcal{O}_b(s_2)\mathcal{O}_c(s_3)\mathcal{O}_d(s_4)\rangle &= (2r)^4 \text{C.T.}(\widetilde{v}_a \star \widetilde{v}_b \star \widetilde{v}_c \star \widetilde{v}_d) \\
&= \delta^{ab}\delta^{cd} + \frac{h^\vee}{2\lambda_2}P_{\mathbf{adj}}^{abcd} + \frac{(2r)^4}{\dim(\mathbf{2adj})}\text{C.T.}(v^\alpha \star v^\alpha)P_{\mathbf{2adj}}^{abcd}.
\end{aligned}
\tag{4.31}
$$

Comparing (4.31) with (4.6) and using (4.15), we find the relation

$$
C_J = -32\lambda_2,
\tag{4.32}
$$

which perfectly agrees with Table 5 and Table 6.

## 5 Superconformal bootstrap

### 5.1 Coulomb-Higgs mixed correlators and $\mathbb{Z}_2$ outer automorphism

This subsection sets up the bootstrap equations for the four-point function of flavor current multiplets. We begin by considering the mixed correlator system that involves one flavor current in each of $SU(2)_C$ and $SU(2)_H$, with some focus on the effect of the $\mathbb{Z}_2$ outer automorphism that exchanges $SU(2)_C$ and $SU(2)_H$ on the bootstrap, and then set up the bootstrap system for the most general flavor symmetry group.

#### 5.1.1 $U(1) \times U(1)$ flavor symmetry

Let us consider the four-point functions involving $U(1)$ flavor current multiplets $\mathcal{B}[0]_1^{(2;0)}$ and $\mathcal{B}[0]_1^{(0;2)}$ (0, 2, or 4 of each):

$$
\begin{aligned}
&\langle \mathcal{O}^{\text{C}}(x_1, Y_1)\mathcal{O}^{\text{C}}(x_2, Y_2)\mathcal{O}^{\text{H}}(x, \widetilde{Y}_3)\mathcal{O}^{\text{H}}(x, \widetilde{Y}_4)\rangle \\
&= \left(\frac{(Y_1 \cdot Y_2)(\widetilde{Y}_3 \cdot \widetilde{Y}_4)}{|x_{12}||x_{34}|}\right)^2 G_s(u,v) = \left(\frac{(Y_2 \cdot Y_3)(\widetilde{Y}_1 \cdot \widetilde{Y}_4)}{|x_{23}||x_{14}|}\right)^2 G_t(v,u), \\
&\langle \mathcal{O}^{\text{C}}(x_1, Y_1)\mathcal{O}^{\text{C}}(x_2, Y_2)\mathcal{O}^{\text{C}}(x, Y_3)\mathcal{O}^{\text{C}}(x, Y_4)\rangle = \left(\frac{(Y_1 \cdot Y_2)(Y_3 \cdot Y_4)}{|x_{12}||x_{34}|}\right)^2 G_{\text{C}}(u,v,w), \\
&\langle \mathcal{O}^{\text{H}}(x_1, \widetilde{Y}_1)\mathcal{O}^{\text{H}}(x_2, \widetilde{Y}_2)\mathcal{O}^{\text{H}}(x, \widetilde{Y}_3)\mathcal{O}^{\text{H}}(x, \widetilde{Y}_4)\rangle = \left(\frac{(\widetilde{Y}_1 \cdot \widetilde{Y}_2)(\widetilde{Y}_3 \cdot \widetilde{Y}_4)}{|x_{12}||x_{34}|}\right)^2 G_{\text{H}}(u,v,\widetilde{w}).
\end{aligned}
\tag{5.1}
$$

The four-point functions can be expanded in terms of the various superconformal blocks as

$$
\begin{aligned}
G_s(u,v) &= \sum_{\mathcal{X}=\mathcal{L},\mathcal{A},\mathcal{B}}\sum_{\Delta,\ell} \lambda_{\text{C,C},\mathcal{X}[2\ell]_\Delta^{(0;0)}}\lambda_{\text{H,H},\mathcal{X}[2\ell]_\Delta^{(0;0)}} \mathcal{A}_{\mathcal{X}[2\ell]_\Delta^{(0;0)}}^s(u,v), \\
G_t(u,v) &= \sum_{\Delta,\ell,j_{\text{C}},j_{\text{H}}} \lambda_{\text{C,H},\mathcal{L}[2\ell]_\Delta^{(2j_{\text{C}};2j_{\text{H}})}}^2 \mathcal{A}_{\Delta,\ell,j_{\text{C}},j_{\text{H}}}^t(u,v), \\
G_{\text{C}}(u,v,w) &= \sum_{\mathcal{X}=\mathcal{L},\mathcal{A},\mathcal{B}}\sum_{\Delta,\ell,j_{\text{C}}} \lambda_{\text{C,C},\mathcal{X}[2\ell]_\Delta^{(2j_{\text{C}};0)}}^2 \mathcal{A}_{\mathcal{X}[2\ell]_\Delta^{(2j_{\text{C}};0)}}(u,v,w), \\
G_{\text{H}}(u,v,\widetilde{w}) &= \sum_{\mathcal{X}=\mathcal{L},\mathcal{A},\mathcal{B}}\sum_{\Delta,\ell,j_{\text{H}}} \lambda_{\text{H,H},\mathcal{X}[2\ell]_\Delta^{(0;2j_{\text{H}})}}^2 \mathcal{A}_{\mathcal{X}[2\ell]_\Delta^{(0;2j_{\text{H}})}}(u,v,\widetilde{w}),
\end{aligned}
\tag{5.2}
$$

where by the $\mathbb{Z}_2$ outer automorphism,

$$
\mathcal{A}_{\mathcal{X}[2\ell]_\Delta^{(0;2j)}}(u,v,w) = \mathcal{A}_{\mathcal{X}[2\ell]_\Delta^{(2j;0)}}(u,v,w).
\tag{5.3}
$$

Putting the above together gives the crossing equations

$$v \sum_{\mathcal{X}=\mathcal{L},\mathcal{A},\mathcal{B}} \sum_{\Delta,\ell} \lambda_{C,C,\mathcal{X}[2\ell]_\Delta^{(0;0)}} \lambda_{H,H,\mathcal{X}[2\ell]_\Delta^{(0;0)}} \mathcal{A}^s_{\mathcal{X}[2\ell]_\Delta^{(0;0)}}(u,v)$$

$$= u \sum_{\Delta,\ell,j_C,j_H} \lambda^2_{C,H,\mathcal{L}[2\ell]_\Delta^{(2j_C;2j_H)}} \mathcal{A}^t_{\Delta,\ell,j_C,j_H}(v,u),$$

$$\sum_{\mathcal{X}=\mathcal{L},\mathcal{A},\mathcal{B}} \sum_{\Delta,\ell,j_C} \lambda^2_{C,C,\mathcal{X}[2\ell]_\Delta^{(2j_C;0)}} \mathcal{F}_{\mathcal{X}[2\ell]_\Delta^{(2j_C;0)}}(u,v,w) = 0,$$

$$\sum_{\mathcal{X}=\mathcal{L},\mathcal{A},\mathcal{B}} \sum_{\Delta,\ell,j_H} \lambda^2_{H,H,\mathcal{X}[2\ell]_\Delta^{(0;2j_H)}} \mathcal{F}_{\mathcal{X}[2\ell]_\Delta^{(0;2j_H)}}(u,v,\widetilde{w}) = 0,$$

(5.4)

where the functions $\mathcal{F}_{\mathcal{X}[2\ell]_\Delta^{(2j_C;0)}}(u,v,w)$ and $\mathcal{F}_{\mathcal{X}[2\ell]_\Delta^{(0;2j_H)}}(u,v,\widetilde{w})$ are

$$\mathcal{F}_{\mathcal{X}[2\ell]_\Delta^{(2j;0)}}(u,v,w) = \mathcal{F}_{\mathcal{X}[2\ell]_\Delta^{(0;2j)}}(u,v,w)$$

$$\equiv vw \mathcal{A}_{\mathcal{X}[2\ell]_\Delta^{(0;2j)}}(u,v,w) - \frac{u}{w} \mathcal{A}_{\mathcal{X}[2\ell]_\Delta^{(0;2j)}}(v,u,w^{-1}).$$

(5.5)

These equations comprise a mixed system by the fact that $\lambda_{C,C,\mathcal{X}[2\ell]_\Delta^{(0;0)}}$ and $\lambda_{H,H,\mathcal{X}[2\ell]_\Delta^{(0;0)}}$ are common coefficients. A more explicit form of (5.4) is given in (D.1).

### 5.1.2 Mirror symmetry and the $\mathbb{Z}_2$ outer automorphism

We are particularly interested in theories where the $\mathbb{Z}_2$ outer automorphism that exchanges the $SU(2)_H$ with $SU(2)_H$ is a true global symmetry of the theory. For $j \equiv j_C = j_H$, let us denote a $\mathbb{Z}_2$ even/odd intermediate multiplet as $\mathcal{X}[2\ell]_{\Delta,\pm}^{(j,j)}$, respectively. When the $\mathbb{Z}_2$ outer automorphism is a global symmetry, the OPE coefficients are related as

$$\lambda_{C,C,\mathcal{X}[2\ell]_{\Delta,\pm}^{(j,j)}} = \pm \lambda_{H,H,\mathcal{X}[2\ell]_{\Delta,\pm}^{(j,j)}},$$

$$\lambda_{C,C,\mathcal{X}[2\ell]_\Delta^{(2j_C,2j_H)}} = \lambda_{H,H,\mathcal{X}[2\ell]_\Delta^{(2j_H,2j_C)}}, \qquad \text{for} \quad j_H \neq j_C.$$

(5.6)

Assuming the $\mathbb{Z}_2$ symmetry, the crossing equation (5.4) becomes

$$v \sum_{\mathcal{X}} \sum_{\Delta,\ell} \left( \lambda^2_{C,C,\mathcal{X}[2\ell]_{\Delta,+}^{(0;0)}} - \lambda^2_{C,C,\mathcal{X}[2\ell]_{\Delta,-}^{(0;0)}} \right) \mathcal{A}^s_{\mathcal{X}[2\ell]_\Delta^{(0;0)}}(u,v)$$

$$= u \sum_{\Delta,\ell,j_C,j_H} \lambda^2_{C,H,\mathcal{L}[2\ell]_\Delta^{(2j_C;2j_H)}} \mathcal{A}^t_{\Delta,\ell,j_C,j_H}(v,u),$$

$$\sum_{\mathcal{X}} \sum_{\Delta,\ell} \sum_{j_C \neq 0} \lambda^2_{C,C,\mathcal{X}[2\ell]_\Delta^{(2j_C;0)}} \mathcal{F}_{\mathcal{X}[2\ell]_\Delta^{2j_C}}(u,v,w)$$

$$+ \sum_{\mathcal{X}} \sum_{\Delta,\ell} \left( \lambda^2_{C,C,\mathcal{X}[2\ell]_{\Delta,+}^{(0;0)}} + \lambda^2_{C,C,\mathcal{X}[2\ell]_{\Delta,-}^{(0;0)}} \right) \mathcal{F}_{\mathcal{X}[2\ell]_\Delta^{2j_C}}(u,v,w) = 0.$$

(5.7)

A more explicit form of (5.7) is given in (D.2).

Every solution to (5.7) can clearly be lifted to a solution to the more general crossing equation (5.4) (not assuming $\mathbb{Z}_2$ symmetry), so the bootstrap constraints imposed by the $\mathbb{Z}_2$ symmetric (5.7) are no weaker than those imposed by the general (5.4). On the other hand, every solution $\lambda$ to the more general crossing equation (5.4) induces a $\mathbb{Z}_2$ symmetric solution $\tilde{\lambda}$ to (5.7), by

$$\tilde{\lambda}_{C,C,\mathcal{X}[2\ell]_{\Delta,\pm}^{(0;0)}} = \frac{1}{2} \left( \lambda_{C,C,\mathcal{X}[2\ell]_\Delta^{(0;0)}} \pm \lambda_{H,H,\mathcal{X}[2\ell]_\Delta^{(0;0)}} \right),$$

$$\tilde{\lambda}^2_{C,C,\mathcal{X}[2\ell]_\Delta^{(2j;0)}} = \frac{1}{2} \left( \lambda^2_{C,C,\mathcal{X}[2\ell]_\Delta^{(2j;0)}} + \lambda^2_{H,H,\mathcal{X}[2\ell]_\Delta^{(0;2j)}} \right), \qquad \text{for} \quad j \neq 0,$$

$$\tilde{\lambda}_{C,H,\mathcal{L}} = \lambda_{C,H,\mathcal{L}}.$$

(5.8)

The lesson here is that we can always construct a $\mathbb{Z}_2$ symmetric correlator that solves the crossing equations (5.7) from a non-symmetric one. Therefore, the crossing equations (5.7) can be used to constrain the quantities on the right hand sides of (5.8) even for theories without the $\mathbb{Z}_2$ symmetry.[23]

But then, how do we test whether a theory we want to bootstrap truly has this additional $\mathbb{Z}_2$ global symmetry?

The answer is as follows. If the $\mathbb{Z}_2$ outer automorphism is a true symmetry, then in an interacting theory without further global symmetry (in addition to the $\mathbb{Z}_2$), we do not expect the $\mathbb{Z}_2$ even spectrum to coincide with the $\mathbb{Z}_2$ odd spectrum. In other words, we expect a collection of the $\mathbb{Z}_2$ even states to *not* have a $\mathbb{Z}_2$ odd "partner" with the same $\Delta, \ell, \mathbf{r}$, and vice versa. By contrast, the induced solution (5.8) to the crossing equations (5.7) from a generic solution to the crossing equations (5.4) has no relation between $\lambda_{\mathrm{CC}}$ and $\lambda_{\mathrm{HH}}$, so the $\mathbb{Z}_2$ even and odd contributions appear *together* for most if not all $\Delta, \ell, \mathbf{r}$.

In conclusion, the hallmark of a genuinely $\mathbb{Z}_2$ symmetric four-point function is that *a collection of the $\mathbb{Z}_2$ even states do not have $\mathbb{Z}_2$ odd "partners" with the same $\Delta, \ell, \mathbf{r}$, or vice versa.*

As will be described in the next subsection, the spectrum of multiplets contributing to the four-point function can be fully determined by the bootstrap if we extremize an OPE coefficient. Testing the extremal spectrum against the above criterion allows us to verify whether the $\mathbb{Z}_2$ outer automorphism is a genuine symmetry of the extremal theory. In the field theory context, the $\mathbb{Z}_2$ outer automorphism becomes a genuine symmetry when the theory has a UV construction that is self-mirror.

### 5.1.3 $G_{\mathrm{C}} \times G_{\mathrm{H}}$ flavor symmetry

Let us consider the four-point functions analogous to (5.1), but for $G_{\mathrm{C}}$ flavor current multiplets $\mathcal{B}[0]_1^{(2;0)}$ and $G_{\mathrm{H}}$ flavor current multiplets $\mathcal{B}[0]_1^{(0;2)}$. The crossing equations are

$$
\frac{1}{u} \sum_{\mathcal{X}} \sum_{\Delta,\ell} \lambda_{\mathrm{C},\mathrm{C},\mathcal{X}[2\ell]_\Delta^{(0;0)},\mathbf{1}} \lambda_{\mathrm{H},\mathrm{H},\mathcal{X}[2\ell]_\Delta^{(0;0)},\mathbf{1}} \mathcal{A}^s_{\mathcal{X}[2\ell]_\Delta^{(0;0)}}(u,v)
$$

$$
= \frac{1}{v} \sum_{\Delta,\ell,j_{\mathrm{C}},j_{\mathrm{H}}} \lambda^2_{\mathrm{C},\mathrm{H},\mathcal{L}[2\ell]_\Delta^{(j_{\mathrm{C}};j_{\mathrm{H}})}} \mathcal{A}^t_{\Delta,\ell,j_{\mathrm{C}},j_{\mathrm{H}}}(v,u),
$$

$$
\sum_{\mathcal{X}} \sum_{\Delta,\ell,j_{\mathrm{C}}} \sum_{\mathbf{r}'\in\mathbf{adj}_{\mathrm{C}}\otimes\mathbf{adj}_{\mathrm{C}}} \lambda^2_{\mathrm{C},\mathrm{C},\mathcal{X}[2\ell]_\Delta^{(2j_{\mathrm{C}};0)},\mathbf{r}'_{\mathrm{C}}} \mathcal{F}_{\mathcal{X}[2\ell]_\Delta^{(2j_{\mathrm{C}};0)},\mathbf{r}_{\mathrm{C}}}^{\mathbf{r}'_{\mathrm{C}}}(u,v,w) = 0,
$$

$$
\sum_{\mathcal{X}} \sum_{\Delta,\ell,j_{\mathrm{H}}} \sum_{\mathbf{r}'_{\mathrm{H}}\in\mathbf{adj}_{\mathrm{H}}\otimes\mathbf{adj}_{\mathrm{H}}} \lambda^2_{\mathrm{H},\mathrm{H},\mathcal{X}[2\ell]_\Delta^{(0;2j_{\mathrm{H}})},\mathbf{r}'_{\mathrm{H}}} \mathcal{F}_{\mathcal{X}[2\ell]_\Delta^{(0;2j_{\mathrm{H}})},\mathbf{r}_{\mathrm{H}}}^{\mathbf{r}'_{\mathrm{H}}}(u,v,w) = 0,
$$

(5.10)

where $\mathbf{r}_{\mathrm{C}}, \mathbf{r}_{\mathrm{H}}$ denote representations of $G_{\mathrm{C}}, G_{\mathrm{H}}$, and we have defined

$$
\mathcal{F}_{\mathcal{X}[2\ell]_\Delta^{(2j;0)},\mathbf{r}}^{\mathbf{r}'}(u,v,w) = \mathcal{F}_{\mathcal{X}[2\ell]_\Delta^{(0;2j)},\mathbf{r}}^{\mathbf{r}'}(u,v,w)
$$

$$
\equiv F_{\mathbf{r}}^{\mathbf{r}'} \frac{w}{u} \mathcal{A}_{\mathcal{X}[2\ell]_\Delta^{(0;2j)}}(u,v,w) - \delta_{\mathbf{r}}^{\mathbf{r}'} \frac{1}{vw} \mathcal{A}_{\mathcal{X}[2\ell]_\Delta^{(0;2j)}}(v,u,\tfrac{1}{w}),
$$

(5.11)

with $F_{\mathbf{r}}^{\mathbf{r}'}$ the crossing matrix (6j symbol) defined in appendix C. A more explicit form of (5.10) is given in (D.3).

---

[23]However, for theories without the $\mathbb{Z}_2$ symmetry, the crossing equation (5.7) cannot constrain the linear combination of the OPE coefficients

$$
\lambda^2_{\mathrm{C},\mathrm{C},\mathcal{X}[2\ell]_\Delta^{(2j;0)}} - \lambda^2_{\mathrm{H},\mathrm{H},\mathcal{X}[2\ell]_\Delta^{(0;2j)}}, \qquad \text{for} \quad j \neq 0. \tag{5.9}
$$

When $G_C = G_H$, and when the $\mathbb{Z}_2$ outer automorphism that exchanges $G_C$ and $G_H$ is a global symmetry, the crossing equations (5.10) reduce to

$$
v \sum_{\mathcal{X}} \sum_{\Delta,\ell} \left( \lambda^2_{C,C,\mathcal{X}[2\ell]^{(0;0)}_{\Delta,+},\mathbf{1}} - \lambda^2_{C,C,\mathcal{X}[2\ell]^{(0;0)}_{\Delta,-},\mathbf{1}} \right) \mathcal{A}^s_{\mathcal{X}[2\ell]^{(0;0)}_{\Delta}}(u,v)
$$

$$
= u \sum_{\Delta,\ell,j_C,j_H} \lambda^2_{C,H,\mathcal{L}[2\ell]^{(2j_C;2j_H)}_{\Delta}} \mathcal{A}^t_{\Delta,\ell,j_C,j_H}(v,u),
$$

$$
\sum_{\mathcal{X}} \sum_{\Delta,\ell} \sum_{j_C \neq 0} \sum_{\mathbf{r}'_C \in \mathbf{adj}_C \otimes \mathbf{adj}_C} \lambda^2_{C,C,\mathcal{X}[2\ell]^{(2j_C;0)}_{\Delta},\mathbf{r}'_C} \mathcal{F}_{\mathcal{X}[2\ell]^{2j_C}_{\Delta},\mathbf{r}_C}{}^{\mathbf{r}'_C}(u,v,w)
$$

$$
+ \sum_{\mathcal{X}} \sum_{\Delta,\ell} \sum_{\mathbf{r}'_C \in \mathbf{adj}_C \otimes \mathbf{adj}_C} \left( \lambda^2_{C,C,\mathcal{X}[2\ell]^{(0;0)}_{\Delta,+},\mathbf{r}'_C} + \lambda^2_{C,C,\mathcal{X}[2\ell]^{(0;0)}_{\Delta,-},\mathbf{r}'_C} \right) \mathcal{F}_{\mathcal{X}[2\ell]^{2j_C}_{\Delta},\mathbf{r}_C}{}^{\mathbf{r}'_C}(u,v,w) = 0 \,.
$$

$$(5.12)$$

A more explicit form of (5.12) is given in (D.4).

Nonetheless, these crossing equations can also constrain theories without the $\mathbb{Z}_2$ symmetry, as the discussion in Section 5.1.2 for the $U(1) \times U(1)$ case also applies to the general case: any solution to the $\mathbb{Z}_2$ symmetric (5.12) can be lifted to a solution to the general (5.10). Conversely, a solution to (5.10) induces a solution to (5.12), via (5.8). Thus, (5.12) constrains the combinations of OPE coefficients appearing on the right hand sides of (5.8).

## 5.2 The linear functional method

The linear functional method is a powerful tool for constraining and solving unitary conformal field theories. We give a schematic explanation of the method here, and refer the reader to earlier papers by some of the authors for more details.

We act a vector valued linear functional $\alpha$ on the bootstrap equations (5.12), and the result can be written schematically as

$$
0 = \sum_{\mathcal{X},\mathbf{r}} \lambda^2_{\mathcal{X},\mathbf{r}} \alpha[\mathcal{K}^{\mathbf{r}}_{\mathcal{X}}]. \tag{5.13}
$$

Here, $\mathcal{K}$ involves $\mathcal{A}$ or $\mathcal{F}$, $\mathcal{X}$ denotes the multiplet, and $\mathbf{r}$ denotes the flavor representation. A linear functional that satisfies

$$
\alpha[\mathcal{K}^{\mathbf{1}}_{B[0]^{(0;0)}_0}] = -1, \quad \alpha[\mathcal{K}^{\mathbf{r}}_{\mathcal{X}}] \geq 0 \quad \text{for all } \mathcal{X}, \mathbf{r} \tag{5.14}
$$

implies a bound on the OPE coefficients

$$
\lambda^2_{\mathcal{X},\mathbf{r}} \leq \frac{1}{\alpha[\mathcal{K}^{\mathbf{r}}_{\mathcal{X}}]}. \tag{5.15}
$$

If we maximize $\alpha[\mathcal{K}^{\mathcal{X},J}]$ while satisfying (5.14), we obtain the optimal upper bound on $\lambda^2_{\mathcal{X},J}$.

An *extremal functional* is one that maximizes $\alpha[\mathcal{K}^{\mathbf{r}}_{\mathcal{X}}]$, which we denote by $\alpha_{\mathcal{X},\mathbf{r}}$. If there exists a four-point function that saturates the bound (5.15), then the OPE coefficients satisfy

$$
0 = \sum_{(\mathcal{X}',\mathbf{r}') \neq (\mathcal{X},\mathbf{r})} \lambda^2_{\mathcal{X}',\mathbf{r}'} \alpha_{\mathcal{X},J}[\mathcal{K}^{\mathbf{r}'}_{\mathcal{X}'}]. \tag{5.16}
$$

In light of (5.14), this means that the spectrum can be read off from the zeros of $\alpha_{\mathcal{X},\mathbf{r}}[\mathcal{K}^{\mathbf{r}'}_{\mathcal{X}'}]$.[24]

In practice, we perform the bootstrap in the following basis of linear functionals. Define $z$ and $\bar{z}$ by

$$
u = z\bar{z}, \quad v = (1-z)(1-\bar{z}), \tag{5.17}
$$

---

[24]While it is often assumed that the extremal correlator is unique, there are situations where we explicitly know this to be false. Nonetheless, we will assume that the spectrum is unique when the extremal theory is believed to be interacting.

so that crossing $u \leftrightarrow v$ is equivalent to $(z, \bar{z}) \leftrightarrow (1-z, 1-\bar{z})$, and consider the space of linear functionals at derivative order $\Lambda$:

$$\alpha = \sum_{m,n=0}^{\Lambda} \alpha_{m,n} \partial_z^m \partial_{\bar{z}}^n |_{z=\bar{z}=\frac{1}{2}}, \quad \alpha_{m,n} \in \mathbb{R}. \tag{5.18}$$

The optimal bound is obtained by extrapolation to infinite $\Lambda$.[25] The semidefinite programming computations are performed using the SDPB solver [80, 81].

## 5.3 Numerical bounds

We are interested in the conformal central charge $C_T$, the flavor central charge $C_J^{\mathrm{C}}$ of the Coulomb branch flavor group $G_{\mathrm{C}}$, and the flavor central charge $C_J^{\mathrm{H}}$ of the Higgs branch flavor group $G_{\mathrm{H}}$. They are related to the OPE coefficients by[26]

$$\lambda^2_{\mathrm{C,C},\mathcal{A}[0]^{(0;0)}_{1,-},\mathbf{1}} = \lambda^2_{\mathrm{H,H},\mathcal{A}[0]^{(0;0)}_{1,-},\mathbf{1}} = \frac{24}{C_T},$$

$$\lambda^2_{\mathrm{C,C},\mathcal{B}[0]^{(2;0)}_1,\mathbf{adj}_{\mathrm{C}}} = \frac{8h^{\vee}_{G_{\mathrm{C}}}}{C_J^{G_{\mathrm{C}}}}, \quad \lambda^2_{\mathrm{H,H},\mathcal{B}[0]^{(0;2)}_1,\mathbf{adj}_{\mathrm{H}}} = \frac{8h^{\vee}_{G_{\mathrm{H}}}}{C_J^{G_{\mathrm{H}}}}. \tag{5.19}$$

For numerical efficiency, we focus on the bootstrap equations with $\mathbb{Z}_2$ symmetry (5.7) and (5.12). One drawback is that we can only give bounds on the "average" flavor central charge $C_J^{\mathrm{avg}}$, given by

$$\frac{1}{C_J^{\mathrm{avg}}} = \frac{1}{2}\left(\frac{1}{C_J^{G_{\mathrm{C}}}} + \frac{1}{C_J^{G_{\mathrm{H}}}}\right). \tag{5.20}$$

### 5.3.1 Single branch

Let us first investigate the bootstrap bounds obtained from the crossing equation of flavor current multiplets with symmetry group $G$ in a single branch. Indeed, there are many interesting theories that only have Higgs branch flavor currents (charged under $SU(2)_{\mathrm{H}}$), but no Coulomb branch ones (charged under $SU(2)_{\mathrm{C}}$). The left side of Table 7 lists the universal lower bounds on $C_J$ and $C_T$ for various choices of $G$. The $G = A_n$ bounds on $C_J$ is saturated by ($\mathbb{Z}_2$-gauged) $n$ free hypers, as described in Section 2.3. The authors are unaware of candidate theories or correlators that saturate the other bounds.

A special class of theories satisfy the minimal nilpotent orbit condition, under which the flavor central charge $C_J$ is completely fixed by the minibootstrap program of Section 4.1. The right side of Table 7 provides a list of the minimal (smallest $C_T$) known such theories for various choices of $G$, and compares them with the corresponding bootstrap bounds on $C_T$ with the minimal nilpotent orbit condition imposed. We observe the following.

- Free theories with $n$ hypermultiplets saturate the minimal nilpotent $C_T$ bounds with flavor symmetry $C_n$.

- Interacting minimal theories are consistent with but do not saturate the nilpotent $C_T$ bounds. In fact, they have values of $C_T$ that lie very close to the bound, especially the $T_3$ theory in the case of $G = E_6$. Although these theories have no Coulomb branch moment

---

[25]We estimate the error of such an extrapolation by the the variation of results using different ansatze. We mark the error of the last significant digit in parentheses following standard convention, in Table 7, (5.21) and (5.22).

[26]As explained in Section 3.2, the $\mathcal{N} = 4$ superconformal primary of the stress tensor block is odd under the $\mathbb{Z}_2$ outer automorphism, though the stress tensor itself is even as is required by conformal Ward identities.

map operators ($\mathcal{N} = 4$ primaries of flavor current multiplets charged under $SU(2)_{\text{C}}$), they do have nontrivial Coulomb branches parameterized by the chiral operators of other $\mathcal{B}$-type multiplets, such as $\mathcal{B}[0]_3^{(4;0)}$. Therefore, in order to bootstrap these theories, it may be necessary to consider the mixed correlator bootstrap with these higher $\mathcal{B}$-type multiplets included as external operators.

- It would be interesting to see if the SQED and SQCD theories saturate the bootstrap bounds with $U(1) \times G_{\text{H}}$ flavor symmetry. We leave this for future work.

Table 7: Single correlator bootstrap lower bounds on the conformal and flavor central charges, both assuming and not assuming the minimal nilpotent orbit condition. Also listed are theories that saturate or are close to saturating the bounds on $(C_T)^{\text{min-nil}}$.

| $G$ | $(C_J)_{\text{min}}$ | $(C_T)_{\text{min}}$ | $(C_J)^{\text{min-nil}}$ | $(C_T)^{\text{min-nil}}_{\text{min}}$ | Reference theory | $C_T$ |
|---|---|---|---|---|---|---|
| $U(1)$ | | 6.02(3) | | 6.02(3) | | |
| $C_1 \cong A_1$ | 4.02(5) | 9.0(1) | $\geq 3.99(3)$ | 11.9(2) | Free hyper | 12 |
| $C_2 \cong B_2$ | 4.09(3) | 17.8(1) | 4 | 23.93(4) | Free hyper | 24 |
| $C_3$ | 4.04(4) | 28.3(3) | 4 | 35.7(2) | Free hyper | 36 |
| $A_2$ | 4.47(5) | 14.93(6) | 6 | 29.68(5) | SQED with 3 hypers | 34.5 |
| $A_3$ | 4.58(2) | 20.1(2) | $\frac{32}{5} = 6.4$ | 44.2(3) | SQED with 4 hypers | 46.3 |
| $B_3$ | 6.19(5) | 29.7(5) | 8 | 57.9(1) | $SU(2)$ SQCD with 7 **fund** half-hypers | 72.1 |
| $E_6$ | 13.66(4) | 102.0(9) | $\frac{192}{13} = 14.77$ | 155.6(3) | $T_3$ | 160.2 |
| $E_7$ | 19.32(2) | 167.6(6) | $\frac{384}{19} = 20.21$ | 239.1(4) | | |
| $E_8$ | 30.30(5) | 304(5) | $\frac{960}{31} = 30.97$ | 406.0(6) | | |
| $F_4$ | 10.68(5) | 70.1(9) | 12 | 113.9(2) | | |
| $G_2$ | 4.89(2) | 19.3(1) | $\frac{64}{9} = 7.11$ | 43.95(9) | | |

### 5.3.2 $G_{\text{C}} = G_{\text{H}} = U(1)$

For the mixed branch bootstrap, the simplest flavor symmetry to consider is $G_{\text{C}} = G_{\text{H}} = U(1)$. In this case, we cannot access $C_J$ (in the absence of a preferred normalization of the abelian current), but can only bound $C_T$. The result after extrapolating to infinite derivative order is

$$(C_T)_{\text{min}} = 12.0(2). \tag{5.21}$$

This value comes close to that of a single free hypermultiplet. However, a single free hypermultiplet has only $G_{\text{H}} = SU(2)$ and no Coulomb branch, $G_{\text{C}} = $ empty, so it is not a candidate for a $\mathbb{Z}_2$ symmetric four-point function; we cannot even apply the construction of adding $\mathbb{Z}_2$ images described in Section 5.1.2.

### 5.3.3 $G_{\text{C}} = G_{\text{H}} = SU(2)$

For $G_{\text{C}} = G_{\text{H}} = SU(2)$ flavor symmetry, the allowed region in the $C_T - C_J$ plane is shown in Figure 3, with and without imposing the minimal nilpotent orbit condition. For the former, the self-mirror $T[SU(2)]$ theory appears to sit at a (soft) corner. Certain BLG, ABJ, and ABJM theories with this flavor symmetry are also included in the figure.

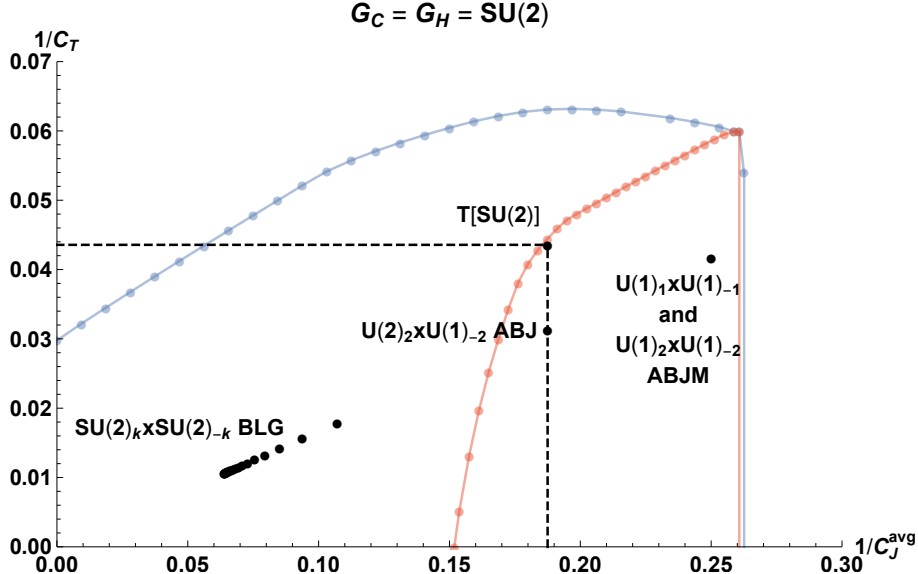

Figure 3: Allowed region in the $C_T - C_J$ plane for $G_C = G_H = SU(2)$ flavor symmetry, at derivative order $\Lambda = 32$. The stronger (red) bounds are with the minimal nilpotent orbit condition imposed, while the weaker (blue) bounds are without.

In Figure 4, we read off the extremal spectrum (in the OPE of moment map operators) from the extremal functional at derivative order $\Lambda = 32$. The $\mathbb{Z}_2$ even and odd sectors appear to have different spectra, providing evidence that the extremal four-point function has a genuine $\mathbb{Z}_2$ symmetry, per the discussion of Section 5.1.2. Moreover, it appears that the lightest multiplet is $\mathbb{Z}_2$ odd, the second lightest is a $\mathbb{Z}_2$ doublet.[27]

### 5.3.4 $G_C = G_H = SU(3)$

For $G_C = G_H = SU(3)$ flavor symmetry, the allowed regions in the $C_T - C_J$ plane are shown in Figure 5, with and without imposing the maximal nilpotent orbit condition. The minimal candidate SCFT with $G_C = G_H = SU(3)$ flavor symmetry is the $T[SU(3)]$ theory, whose $C_T$ and $C_J$ values are inside the allowed region, but do not appear to sit at the boundary. In Figure 6, we extrapolate our lower bounds on $C_T$ with $C_J = 8$ fixed to infinite derivative order, and find

$$(C_T)_{\min} = 51.7(4), \tag{5.22}$$

which is far from the value $C_T = 75.5$ of $T[SU(3)]$.

If we instead impose the minimal nilpotent orbit condition, then we learn from the mini-bootstrap of Section 4.1 that the flavor central charge is fixed to be $C_J = 6$. Fixing this value of $C_J$, we find that the universal lower bound on the conformal central charge at derivative order $\Lambda = 28$ is[28]

$$(C_T)_{\min}^{\min\text{-nil}} = 36.9. \tag{5.23}$$

---

[27]Since our data is insufficient for reliable precision study, we do not provide numerical values for the scaling dimensions of multiplets appearing in the OPE of moment map operators. Moreover, the lightest $\mathbb{Z}_2$ odd multiplet may not actually exist, if the corresponding zero of the extremal functional converges towards the unitarity bound as $\Lambda \to \infty$. If existent, then it would be interesting to study the deformation by this $\mathbb{Z}_2$-odd relevant operator.

[28]Since we do not have a reference theory/value in mind, we choose to present the rigorous bound at the highest derivative order that we have achieved in the bootstrap analysis, instead of the extrapolated result. Thus there is no extrapolation error here, and we keep three significant digits because the next digit changes between $\Lambda = 24$ and $\Lambda = 28$, and would likely change further for $\Lambda > 28$ giving stronger bounds.

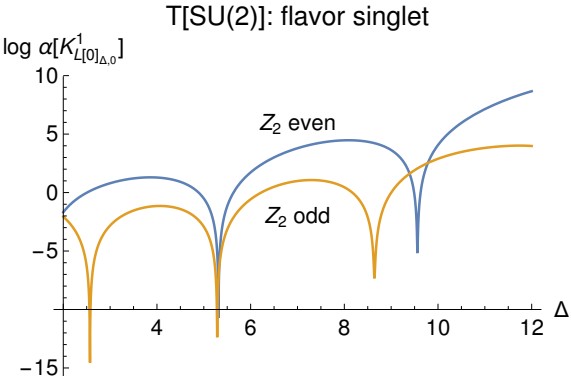

Figure 4: Extremal functional $\alpha[\mathcal{K}^{\mathbf{1}}_{\mathcal{L}[0]_{\Delta,0}}]$ for $G_{\mathrm{C}} = G_{\mathrm{H}} = SU(2)$ flavor symmetry with $C_J = \frac{16}{3}$ fixed and $C_T$ minimized at derivative order $\Lambda = 32$. Shown here are the sectors that are singlet under the flavor symmetry, and even or odd under the $\mathbb{Z}_2$ outer automorphism. The zeros correspond to the scaling dimensions appearing in the OPE of $\mathcal{B}[0]_0^{(2;0)}$ and $\mathcal{B}[0]_0^{(0;2)}$ in the $T[SU(2)]$ theory. The mismatch between the $\mathbb{Z}_2$ even and odd sectors indicates that the $\mathbb{Z}_2$ is a true global symmetry in the extremal theory.

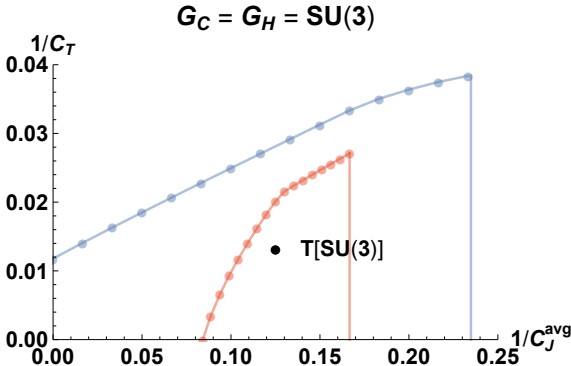

Figure 5: Allowed region in the $C_T - C_J$ plane for $G_{\mathrm{C}} = G_{\mathrm{H}} = SU(3)$ flavor symmetry, at derivative order $\Lambda = 28$. The stronger (red) bounds are with the maximal nilpotent orbit condition imposed, while the weaker (blue) bounds are without.

## 6 Conclusion

In this work we utilized the non-perturbative methods of the conformal bootstrap and supersymmetric localization, together with special properties of 3d $\mathcal{N} = 4$ such as mirror symmetry and a protected subsector described by topological quantum mechanics (TQM), to obtain universal constraints on 3d $\mathcal{N} = 4$ superconformal field theories. A key ingredient in the bootstrap analysis was the determination of the mixed-branch superconformal blocks. The main results are summarized as follows:

- We studied the single branch bootstrap with flavor symmetry $G = U(1)$, $G = ABC$ of low ranks, and $G = EFG$. Free theories with $n$ hypermultiplets appeared to saturate the universal lower bounds on $C_T$ with $G = C_n$ and the minimal nilpotent orbit condition imposed. Other theories that came relatively close to saturating the bounds were the SQED with $n + 1$ hypers to $G = A_n$ and the $T_3$ theory to $G = E_6$. We highlight these values below:

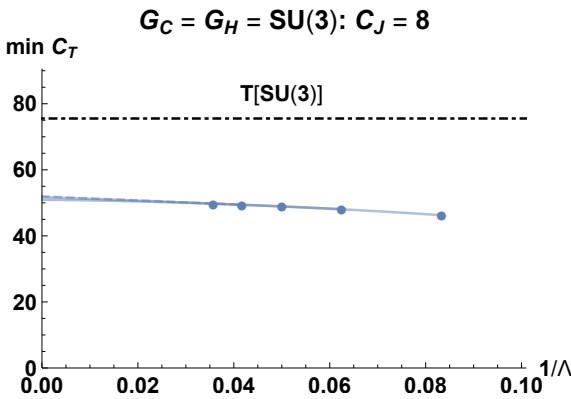

Figure 6: Lower bounds on $C_T$ with $C_J = 8$ fixed to be the value for the $T[SU(3)]$ theory, for derivative order $12 \leq \Lambda \leq 28$, and extrapolated to infinite $\Lambda$. Also shown is the actual value of $C_T = 75.5$ for $T[SU(3)]$.

| $G$ | $(C_T)^{\text{min-nil}}_{\text{min}}$ | Reference theory | $C_T$ |
|---|---|---|---|
| $C_1 \cong A_1$ | 11.9(2) | Free hyper | 12 |
| $C_2 \cong B_2$ | 23.93(4) | Free hyper | 24 |
| $C_3$ | 35.7(2) | Free hyper | 36 |
| $A_2$ | 29.68(5) | SQED with 3 hypers | 34.5 |
| $A_3$ | 44.2(3) | SQED with 4 hypers | 46.3 |
| $E_6$ | 155.6(3) | $T_3$ | 160.2 |

- For the mixed branch bootstrap with flavor symmetry $G_{\text{C}} = G_{\text{H}} = SU(2)$, we found that the $T[SU(2)]$ theory sits at a corner in the allowed region in the $C_T - C_J$ plane. The spectrum is read off from the extremal functional, and the mismatch between the even and odd sectors indicated that the $\mathbb{Z}_2$ outer automorphism is a true global symmetry in $T[SU(2)]$ (it cannot be reproduced by certain spurious $\mathbb{Z}_2$ symmetric solutions to the crossing equations).

- For the mixed branch bootstrap with flavor symmetry $G_{\text{C}} = G_{\text{H}} = SU(3)$, we found that the $T[SU(3)]$ theory sits in the interior of the allowed region in the $C_T - C_J$ plane.

The framework developed here can readily be applied to a wider range of flavor symmetries and assumptions, including self-mirror theories beyond $SU(2)$ and $SU(3)$ flavor factors, and non-self-mirrors situations with $G_{\text{C}} \neq G_{\text{H}}$ or $G_{\text{C}} = G_{\text{H}}$ but different flavor central charges. In particular, many of the known theories that came close to (but not quite) saturating our single-branch bootstrap bounds have at least $U(1)$ flavor symmetry in the other branch, whose incorporation into bootstrap may bring the theories (closer) to actual saturation.

Another promising direction is to explore bounds on certain protected OPE coefficients beyond the TQM sectors.[29] For example, the OPE coefficient for the $\mathcal{B}[0]^{(2;2)}_2$ operator that appears in the OPE of one Higgs branch and one Coulomb branch moment map operator is not captured by the Higgs branch or Coulomb branch TQM sector, but is part of the chiral ring data of the theory viewed as an $\mathcal{N} = 2$ SCFT. In particular, when the SCFT has a UV gauge theory description, this OPE coefficient may accessed from taking multiple derivatives of the (appropriately deformed) $S^3$ partition function with respect to mass and FI parameters at the same time. This provides additional input to refine our exploration of mirror symmetry.

---

[29]We thank Silviu Pufu for an interesting comment on this point.

Futhermore, the application of the analytic bootstrap [82–85] or the OPE inversion formula [86–88] can shed light on interesting limits of 3d $\mathcal{N} = 4$ superconformal field theory, that may be relevant for AdS/CFT.

One can also consider the four-point functions of chiral operators beyond moment map operators, to study theories without continuous flavor symmetry. For non-self-mirror theories such as the 3d Minahan-Nemeschansky theories, the incorporation of the Coulomb branch protected algebra (which no longer involve moment map operators) will be important to improve the bootstrap bounds. Another obvious avenue for future exploration is the bootstrap analysis of the four-point function of the stress-tensor multiplet. In particular, the most minimal known theories [89] have no chiral operators, and are inaccessible by the study of such four-point functions. Some superconformal blocks that are necessary in the bootstrap of these more general four-point functions have been computed in [45, 90].

# Acknowledgments

We are grateful to Silviu Pufu for interesting comments on the draft. The computations in this paper are performed on the Helios computing cluster supported by the School of Natural Sciences Computing Staff at the Institute for Advanced Study. C.C. is supported in part by the U.S. Department of Energy grant DE-SC0009999. The work of M.F. is supported by an SNS fellowship P400P2-180740, the Princeton physics department, the JSPS Grant-In-Aid for Scientific Research Wakate(A) 17H04837, the WPI Initiative, MEXT, Japan at IPMU, the University of Tokyo, and the David and Ellen Lee Postdoctoral Scholarship. Y.L. is supported by the Sherman Fairchild Foundation, and both M.F. and Y.L. by the U.S. Department of Energy, Office of Science, Office of High Energy Physics, under Award Number DE-SC0011632. The work of S.H.S. is supported by the National Science Foundation grant PHY-1606531, the Roger Dashen Membership, and a grant from the Simons Foundation/SFARI (651444, NS). The work of Y.W. is supported in part by the US NSF under Grant No. PHY-1620059 and by the Simons Foundation Grant No. 488653. This research was supported in part by Perimeter Institute for Theoretical Physics. Research at Perimeter Institute is supported by the Government of Canada through the Department of Innovation, Science and Economic Development and by the Province of Ontario through the Ministry of Research and Innovation. C.C., M.F., S.H.S. and Y.W. thank the Aspen Center for Physics, which is supported by National Science Foundation grant PHY-1607611, for hospitality during the finishing stages of this paper.

# A  Localization computation of the central charges

In this appendix, we provide detailed formulae for computing the conformal and flavor central charges, $C_T$ and $C_J$, of 3d $\mathcal{N} = 4$ superconformal field theories, derived from the matrix models obtained by supersymmetric localization [21, 52–57]. Using these formulae, we obtain explicit analytic expressions for these central charges in certain classes of theories.

## A.1  Double-sine functions and derivatives

We first start by introducing the main players, the double sine functions, and their various properties. The double sine functions were introduced in [91, 92], and are closely related to Barnes' double-gamma functions [93] (see also [94]). They are most straightforwardly defined

using their infinite product form,

$$S_2(z|\omega_1, \omega_2) = \prod_{n_1, n_2 = 0}^{\infty} \frac{z + n_1\omega_1 + n_2\omega_2}{-z + (n_1 + 1)\omega_1 + (n_2 + 1)\omega_2}. \tag{A.1}$$

However, for our purposes in the following, the integral expression is more useful: for $0 < \mathrm{Re}\,\omega_j$ and $0 < \mathrm{Re}\,z < \mathrm{Re}\,|\omega_1 + \omega_2|$,

$$S_2(z|\omega_1, \omega_2) = \exp\left( \frac{\pi i}{2} B_{2,2}(z|\omega_1, \omega_2) + \int_{\mathcal{C}} \frac{d\ell}{\ell} \frac{e^{z\ell}}{(e^{\omega_1\ell} - 1)(e^{\omega_2\ell} - 1)} \right). \tag{A.2}$$

The integration contour $\mathcal{C}$ is along the real axis except near the (essential) singularity at the origin where it runs along an infinitesimal half-circle in the upper half plane, and the multiple Bernoulli function $B_{2,2}(z|\omega_i)$ is given by

$$B_{2,2}(z|\omega_1, \omega_2) = \frac{z^2}{\omega_1\omega_2} - \frac{\omega_1 + \omega_2}{\omega_1\omega_2}z + \frac{\omega_1^2 + \omega_2^2 + 3\omega_1\omega_2}{6\omega_1\omega_2}. \tag{A.3}$$

Some useful identities are

$$\begin{aligned} S_2(cz|c\omega_1, c\omega_2) &= S_2(z|\omega_i), \\ S_2(x|\omega_1, \omega_2) &= S_2(\omega_1 + \omega_2 - x|\omega_1, \omega_2)^{-1}, \\ S_2(x|\omega_1, \omega_2)S_2(-x|\omega_1, \omega_2) &= -4\sin\frac{\pi x}{\omega_1}\sin\frac{\pi x}{\omega_2}. \end{aligned} \tag{A.4}$$

In the (round) limit $\omega_i \to 1$, we have

$$S_2(\pm ix + 1/2|1, 1) \equiv S_2(ix + 1/2|1, 1)S_2(-ix + 1/2|1, 1) = 2\cosh\pi x. \tag{A.5}$$

Throughout this paper, $S_2(\pm z)$ is understood as the product $S_2(z)S_2(-z)$.

As we are interested in computing the conformal central charges from the squashed $S^3$ partition function, we shall define

$$S_2(z|b) \equiv S_2(z|b, b^{-1}), \quad \text{and} \quad Q \equiv b + b^{-1}. \tag{A.6}$$

Furthermore, we are required to take derivatives of the double sine functions with respect to the squashing parameter $b$. This is most conveniently done using the explicit integral expression (A.2), and one can explicitly compute

$$\partial_b|_{b=1} S_2(z|b) = 0, \qquad \partial_b|_{b=1} S_2(z + Q/4|b) = 0. \tag{A.7}$$

The non-trivial contributions will come from $\partial_b^2$ acting on $S_2(z|b)$, namely,

$$\begin{aligned} \frac{\partial_b^2|_{b=1} S_2(z|b)}{S_2(z|1)} &= \frac{\pi\left(2\pi(z-2)(z-1)z + (3z-2)\sin 2\pi z\right)}{6\sin^2 \pi z}, \\ \frac{\partial_b^2|_{b=1} S_2(z + Q/4|b)}{S_2(z + Q/4|1)} &= \frac{\pi^2\left(8z^3 - 12z^2 - 2z + 3\right) - \pi(6z+1)\sin 2\pi z}{24\cos 2\pi z}. \end{aligned} \tag{A.8}$$

For instance, we can now immediately conclude

$$2\frac{\partial_b^2|_{b=1} S_2(Q/4|b)}{S_2(Q/4|1)} = \frac{\pi^2}{4}, \tag{A.9}$$

leading to $C_T = 12$ for the free hypermultiple using the formula (2.15). The generalization to $n$ free hypermultiplets follows immediately, $C_T = 12n$.

Finally, we remark that the matrix model integrals possess a $z \to -z$ symmetry, so the following combinations are useful:

$$
\begin{aligned}
\frac{\partial_b^2\big|_{b=1} S_2(z|b)}{S_2(z|1)} + \frac{\partial_b^2\big|_{b=1} S_2(-z|b)}{S_2(-z|1)} &= \frac{\pi z(-2\pi z + \sin 2\pi z)}{\sin^2 \pi z} \\
\frac{\partial_b^2\big|_{b=1} S_2(z+Q/4|b)}{S_2(z+Q/4|1)} + \frac{\partial_b^2\big|_{b=1} S_2(-z+Q/4|b)}{S_2(-z+Q/4|1)} &= \frac{\pi(\pi - 4\pi z^2 - 2z \sin 2\pi z)}{4\cos^2 \pi z} .
\end{aligned}
\tag{A.10}
$$

## A.2  Squashed $S^3$ partition function

Before we proceed with the evaluation of the conformal and flavor central charges, let us briefly recall the explicit localization results for $\mathcal{N} = 4$ (Lagrangian) theories on the squashed $S^3$ background (2.28) [21,52–57] (see also [58]). The contribution to the matrix model arising from $\mathcal{N} = 4$ vector multiplets associated to a gauge group $H$ is given by

$$
Z_b^{\text{vector}} = \frac{1}{|\mathcal{W}|} \int_{\mathfrak{t}} d\sigma \prod_{\alpha \in \Delta^+ \cup \Delta^-} S_2(i\alpha(\sigma)|b) \equiv \frac{1}{|\mathcal{W}|} \int_{\mathfrak{t}} d\sigma \prod_{\alpha \in \Delta^+} 4\sinh(\pi b\alpha(\sigma))\sinh(\pi b^{-1}\alpha(\sigma)),
\tag{A.11}
$$

where we integrate over the Cartan subalgebra $\mathfrak{t} \subset \mathfrak{h} = \text{Lie}\,H$, $\Delta^{\pm}$ is the set of all positive/negative roots of $H$, $\mathcal{W}$ is the Weyl group of $H$, and in the second equality we used the third identity in (A.4).

Additionally, the contribution from $\mathcal{N} = 4$ hypermultiplets in a representation $(R_{G_{\text{UV}}}, R_H)$ of the maximal subgroup $G_{\text{UV}} \times H \subset USp(2n)$ to the localized path integral is given by

$$
Z_b^{\text{hyper}} = \prod_{\rho \in R_{G_{\text{UV}}}} \prod_{\hat{\rho} \in R_H} \frac{1}{S_2(\pm i\rho(m) \pm i\hat{\rho}(\sigma) + Q/4|b)},
\tag{A.12}
$$

where the products are over the weights. Here $G_{\text{UV}}$ is the flavor symmetry realized by the gauged hypermultiplets and the dependence on $m$ encodes $C_J(G_{\text{UV}})$.

As will be used in the following, we remark that in the round limit $b \to 1$, the hypermultiplet contribution reduces to

$$
Z_{b=1}^{\text{hyper}} = \prod_{\rho \in R_{G_{\text{UV}}}} \prod_{\hat{\rho} \in R_H} \frac{1}{2\cosh \pi(\rho(m) \pm \hat{\rho}(\sigma))},
\tag{A.13}
$$

where we used the identity in (A.5).

Finally, we may add Fayet-Iliopoulos parameters to our theory which serve as mass parameters for topological $U(1)$ symmetries. Namely, for a theory of gauge group $G$, we may introduce

$$
Z_b^{FI} = e^{-2\pi i \sum_a \eta_a \sigma_a},
\tag{A.14}
$$

where the sum is taken to run over each (UV) abelian factor in $\mathfrak{h}$.

To get the full partition function of a Lagrangian $\mathcal{N} = 4$ theory with masses for flavor symmetries, we put the above expressions together and integrate over the gauge group according to (A.11).

## A.3  Computation of $C_T$

We now proceed to provide some details on the explicit evaluation of the conformal central charge $C_T$ for various $\mathcal{N} = 4$ SCFTs by using its relation to derivatives with respect to the squashing parameter of the squashed $S^3$ partition function as expressed in the formula (2.15).

**SQED with $k$ unit charge hypermultiplets**

The squashed $S^3$ partition function of the $\mathcal{N} = 4$ SQED theory, with quiver

$$U(1) \!-\!\!-\!\!-\! \boxed{k} \tag{A.15}$$

is given by

$$Z_b^{\text{SQED}_k} = \int_{\mathbb{R}} d\sigma \frac{1}{[S_2(\pm i\sigma + Q/4|b)]^k}. \tag{A.16}$$

Thus, using the central charge formula (2.15) as well as the relation (A.8), we get

$$
\begin{aligned}
C_T(\text{SQED}_k) &= -\frac{48}{\pi^2} \frac{1}{Z_b^{\text{SQED}_k}} \frac{\partial^2 Z_b^{\text{SQED}_k}}{\partial b^2}\bigg|_{b=1} \\
&= \frac{48}{\pi^2} \frac{k}{Z_{b=1}^{\text{SQED}_k}} \int_{\mathbb{R}} \frac{d\sigma}{(2\cosh\pi\sigma)^k} \frac{\pi(\pi + 4\pi\sigma^2 + 2\sigma\sinh 2\pi\sigma)}{4\cosh^2\pi\sigma}.
\end{aligned}
\tag{A.17}
$$

Now, we briefly recall the result for $Z_{b=1}^{\text{SQED}_k}$ [68], which can be computed for example by summing over poles in the upper half-plane,

$$Z_{b=1}^{\text{SQED}_k}(m_\alpha) = \int_{\mathbb{R}} d\sigma \frac{e^{2\pi i\eta\sigma}}{\prod_\alpha 2\cosh\pi(\sigma - m_\alpha)} = \frac{1}{i^{n-1}(e^{\pi\eta} - (-1)^n e^{-\pi\eta})} \sum_{\alpha=1}^n \frac{e^{2\pi i m_\alpha\eta}}{\prod_{\beta\neq\alpha} 2\sinh\pi(m_{\alpha\beta})}, \tag{A.18}$$

where we turned on mass parameters $m_\alpha$ for when we compute $C_J$ and an FI parameter $\eta$ which acts as a regulator. Then, together with following integration identities,

$$
\begin{aligned}
\int_{\mathbb{R}} \frac{2\pi\sigma\sinh\pi\sigma \, d\sigma}{(2\cosh\pi\sigma)^{k+1}} &= \int_{\mathbb{R}} \frac{d\sigma}{k(2\cosh\pi\sigma)^k}, \\
\int_{\mathbb{R}} \frac{d\sigma}{(2\cosh\pi\sigma)^k} &= \frac{\Gamma\left(\frac{k}{2}\right)}{\sqrt{\pi} 2^k \Gamma\left(\frac{k+1}{2}\right)}, \\
\int_{\mathbb{R}} \frac{(2\pi\sigma)^2 d\sigma}{(2\cosh\pi\sigma)^{k+2}} &= \frac{4}{\pi(k+2)} \frac{\Gamma(2 + \frac{k}{2})^2 \Gamma(1, \frac{k+2}{2})}{\Gamma(3+k)},
\end{aligned}
\tag{A.19}
$$

we find

$$C_T(\text{SQED}_k) = \frac{12\left(2k^2\psi^{(1)}\left(\frac{k}{2}+1\right) + \left(\pi^2 k + 4\right)k + 4\right)}{\pi^2(k+1)}, \tag{A.20}$$

where $\psi^{(1)}(z) \equiv \frac{d}{dz}\frac{\Gamma'(z)}{\Gamma(z)}$ is the $z$-derivative of the digamma function.

**$SU(2)$ SYM with $k$ fundamental half-hypermultiplets**

We now compute the conformal central charge for the 3d $\mathcal{N} = 4$ $SU(2)$ theory with $k > 4$ fundamental half-hypermultiplets, *i.e.*

$$SU(2) \!-\!\!-\!\!-\! \boxed{k} \tag{A.21}$$

This theory has an $SO(k)$ global symmetry. Given the general rules outlined in the previous subsection, the localized squashed sphere partition function of that theory reads

$$Z_b^{\text{SQCD}_k} = \int_{\mathbb{R}} \frac{d\sigma}{2} \frac{4\sinh^2(2\pi\sigma)}{[S_2(\pm i\sigma + Q/4|b)]^k}. \tag{A.22}$$

Using the central charge fomula (2.15), and recalling the general insertion rule in (A.10) for computing derivatives, we end up with the following expression

$$C_T(\text{SQCD}_k) = \frac{48}{\pi^2} \frac{1}{Z_{b=1}^{\text{SQCD}_k}} \int_{\mathbb{R}} \frac{d\sigma}{2} \frac{4\sinh^2(2\pi\sigma)}{(2\cosh\pi\sigma)^k} \left( k \frac{\pi(\pi + 4\pi\sigma^2 + 2\sigma\sinh 2\pi\sigma)}{4\cosh^2\pi\sigma} \right. $$
$$\left. + \frac{\pi\sigma(2\pi\sigma - \sinh 2\pi\sigma)}{\sinh^2\pi\sigma} \right). \tag{A.23}$$

Evaluating the round sphere partition function yields

$$Z_{b=1}^{\text{SQCD}_k} = \frac{2^{2-k}\Gamma\left(\frac{k}{2} - 2\right)}{\sqrt{\pi}\Gamma\left(\frac{k-1}{2}\right)}, \tag{A.24}$$

and repeatedly applying the integral identities in equation (A.19), we end up with

$$C_T(\text{SQCD}_k) = \frac{12}{\pi^2(k-4)(k-2)(k-1)} \left[ n\left(\pi^2(k-4)(k-2) + 12k - 20\right)(k-4) \right.$$
$$\left. + 2(k-2)(3k-2)(k-4)^2\psi^{(1)}\left(\frac{k-2}{2}\right) + 48 \right]. \tag{A.25}$$

### $T[SU(3)]$ from gauging $T[SU(2)]$

As discussed in Section 2.4, the $T[SU(3)]$ theory can be described by the quiver [41, 43, 65]

$$\boxed{U(1) \!\!-\!\! U(2) \!\!-\!\! \boxed{3}} \tag{A.26}$$

Thus, it is obtained from gauging the $SU(2) \times U(1)$ flavor symmetry of the $T[SU(2)]$ theory (*i.e.* $\text{SQED}_{k=2}$) and three additional hypermultiplets in the doublet representation of $U(2)$. The localized squashed $S^3$ partition function is then given by

$$Z_b^{T[SU(3)]} = \int_{\mathbb{R}} \frac{du_1 du_2}{2!} \prod_{j<\ell} S_2(\pm i u_{j\ell}|b) \frac{Z_b^{T[SU(2)]}(u_{12}/2)}{\prod_{j=1}^2 S_2(\pm i u_j + Q/4|b)^3}, \tag{A.27}$$

where

$$Z_b^{T[SU(2)]}(u) = \int_{\mathbb{R}} d\sigma \frac{1}{S_2(\pm i\sigma + iu + Q/4|b)S_2(\pm i\sigma - iu + Q/4|b)}, \tag{A.28}$$

and where we used the shorthand $u_{12} = u_1 - u_2$. Then, using the $C_T$ formula (2.15) and the identities in (A.10), we get the following expression for the central charge

$$C_T(T[SU(3)]) = \frac{48}{\pi^2} \frac{1}{Z_{b=1}^{T[SU(3)]}} \int_{\mathbb{R}} \frac{du_1 du_2}{2!} \frac{4\sinh^2(\pi u_{12})}{\prod_j (2\cosh\pi u_j)^3} Z_{b=1}^{T[SU(2)]}(u_{12}/2)$$
$$\times \left[ 3\sum_j \frac{\pi(\pi + 4\pi u_j^2 + 2u_j\sinh 2\pi u_j)}{4\cosh^2\pi u_j} - \frac{\pi u_{12}(-2\pi u_{12} + \sinh 2\pi u_{12})}{\sinh^2\pi u_{12}} \right.$$
$$\left. - \frac{1}{Z_{b=1}^{T[SU(2)]}} \frac{\partial^2 Z_{b=1}^{T[SU(2)]}}{\partial b^2}\bigg|_{b=1} \right], \tag{A.29}$$

where we can by the same methods evaluate

$$-\frac{\partial^2 Z_b^{T[SU(2)]}(u)}{\partial b^2}\bigg|_{b=1} = \frac{\pi}{6}\frac{(1+6u^2)\sinh 4\pi u - 4\pi u(1+4u^2)}{\sinh^3 2\pi u}, \tag{A.30}$$

and we further have the solutions for the round sphere (see *e.g.* [68,69])

$$Z_{b=1}^{T[SU(2)]}(u) = \frac{u}{\sinh 2\pi u}, \qquad Z_{b=1}^{T[SU(3)]} = \frac{1}{16\pi^3}. \tag{A.31}$$

Putting the pieces together, we can numerically evaluate the the pieces in (A.29) to get[30]

$$C_T(T[SU(3)]) = 75.5329. \tag{A.32}$$

### $T_3$ from gauging three $T[SU(3)]$ theories

The (mirror of the) 3d $T_3$ theory can be obtained by gauging three $T[SU(3)]$ theories [95] (see also [96]), *i.e.*[31]

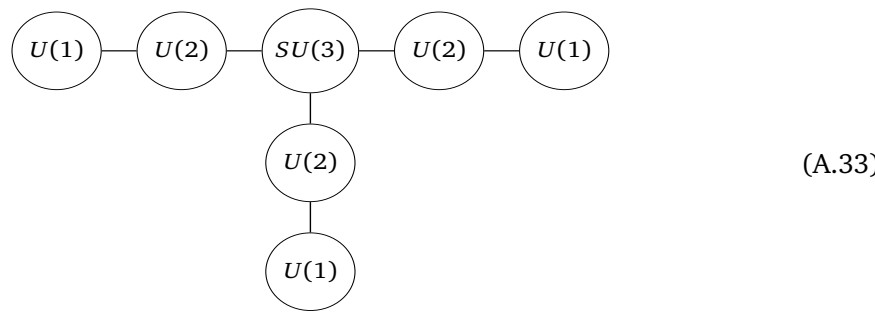

$$\tag{A.33}$$

Thus, the localized squashed $S^3$ partition function of the $T_3$ (mirror) theory is given by

$$Z_b^{T_3} = \int_{\mathbb{R}} \frac{\prod_{j=1}^3 dv_j}{3!}\delta\Big(\sum_j v_j\Big)\prod_{k<\ell} 4\sinh^2(\pi v_{k\ell})\Big(Z_b^{T[SU(3)]}(\vec{v})\Big)^3, \tag{A.34}$$

with notation as above and where $Z_b^{T[SU(3)]}(v_j)$ is the (Higgs branch) mass deformed squashed $S^3$ $T[SU(3)]$ partition function, *i.e.*

$$Z_b^{T[SU(3)]}(\vec{v}) = \int_{\mathbb{R}} \frac{du_1 du_2}{2!}\prod_{\ell<k} S_2(\pm iu_{\ell k}|b)\frac{Z_b^{T[SU(2)]}(u_{12}/2)}{\prod_{\ell,k=1}^3 S_2(\pm i(u_\ell + v_k)+\frac{Q}{4}|b)}. \tag{A.35}$$

In the round limit, this is evaluates to [68,69]

$$Z_{b=1}^{T[SU(3)]}(\vec{v}) = \frac{v_{12}v_{13}v_{23}}{16\sinh(\pi v_{12})\sinh(\pi v_{13})\sinh(\pi v_{23})}. \tag{A.36}$$

Again, using our trusty $C_T$ formula (2.15) and the identities in (A.10), we find

$$C_T(T_3) = -\frac{48}{\pi^2}\frac{1}{Z_{b=1}^{T_3}}\int_{\mathbb{R}} \frac{\prod_{j=1}^3 dv_j}{3!}\delta\Big(\sum_j v_j\Big)\prod_{k<\ell} 4\sinh^2(\pi v_{k\ell})\Big(Z_{b=1}^{T[SU(3)]}(\vec{v})\Big)^3$$
$$\times\left(\sum_{k<\ell}\frac{\pi v_{k\ell}(\sinh(2\pi v_{k\ell})-2\pi v_{k\ell})}{\sinh^2 \pi v_{k\ell}} + \frac{3}{Z_{b=1}^{T[SU(3)]}(\vec{v})}\frac{\partial^2 Z_b^{T[SU(3)]}(\vec{v})}{\partial b^2}\bigg|_{b=1}\right), \tag{A.37}$$

---

[30]All the numerical integrals throughout this paper are evaluated using `Mathematica` with 6 significant digits.

[31]There are alternative constructions; for example the proposals in [97,98] suggest $\mathcal{N}=1$ Lagrangian descriptions.

where (similar to before) we can write

$$
\begin{aligned}
-\frac{\partial^2 Z_b^{T[SU(3)]}(\vec{v})}{\partial b^2}\bigg|_{b=1} &= \int_{\mathbb{R}} \frac{du_1 du_2}{2!} \frac{4\sinh^2(\pi u_{12}) Z_{b=1}^{T[SU(2)]}(u_{12}/2)}{\prod_{k,\ell} 2\cosh \pi(u_k + v_\ell)} \\
&\times \Bigg( \sum_{k,\ell} \frac{\pi(\pi + 4\pi(u_k + v_\ell)^2 + 2(u_k + v_\ell)\sinh 2\pi(u_k + v_\ell))}{4\cosh^2 \pi(u_k + v_\ell)} \\
&- \frac{\pi u_{12}(\sinh 2\pi u_{12} - 2\pi u_{12})}{\sinh^2 \pi u_{12}} + \frac{\pi\big((3u_{12}^2/2 + 1)\sinh(2\pi u_{12}) - 2\pi u_{12}(u_{12}^2 + 1)\big)}{3u_{12}\sinh^2 \pi u_{12}} \Bigg).
\end{aligned}
\tag{A.38}
$$

We can now numerically evaluate the required integrals. However, we can do better; namely, we can explicitly evaluate the second and third term in (A.38). To do so, let us first define the following quantity

$$
F(\eta) = \int_{\mathbb{R}} dy_1 dy_2 \frac{e^{2\pi i \eta y_{12}}}{2\sinh \pi y_{12} \prod_{i=1}^{2} \prod_{\alpha=1}^{n} 2\cosh \pi(y_i - m_\alpha)},
\tag{A.39}
$$

which is convergent for generic $\eta \neq 0$ and (mildly) divergent at $\eta = 0$ ($F'(\eta)$ is well-defined everywhere). Thus, we may use the Fourier transform of the distribution

$$
\frac{1}{\sinh \pi x} = i \int dy\, e^{-2\pi i x a} 2\tanh \pi a,
\tag{A.40}
$$

and write

$$
F(\eta) = \frac{i}{2} \int dy_1 dy_2 da \frac{e^{2\pi i(\eta - a)y_{12}} 2\tanh \pi a}{\prod_{i=1}^{2} \prod_{\alpha=1}^{n} 2\cosh \pi(y_i - m_\alpha)}.
\tag{A.41}
$$

This can be thought of as an analytic continuation (analogous to Principal Value regularization) of (A.39) which is now well-defined for arbitrary $\eta$. We remark that the integrand contains two factors of the SQED$_n$ integral in equation (A.18), and thus (after a change of variables) we may explicitly evaluate those contributions to find

$$
F(\eta) = \frac{i}{8\sinh^2 \pi \eta} \sum_{\alpha,\beta} \frac{m_{\alpha\beta}\sinh(2\pi\eta) - \sin(2\pi m_{\alpha\beta}\eta)}{\sinh \pi m_{\alpha\beta} \prod_{i\neq\alpha} 2\sinh \pi m_{\alpha i} \prod_{j\neq\beta} 2\sinh \pi m_{\beta j}}.
\tag{A.42}
$$

With the quantity $F(\eta)$ in hand, the integral of the second and third terms in equation (A.38) can be expressed in terms of the function (A.39) as follows

$$
\frac{\pi}{3}\big[F(-i) - F(i) + 2iF'(0)\big] + \frac{1}{8\pi}\big[F''(-i) - F''(i)\big] + \frac{i}{3\pi}F'''(0).
\tag{A.43}
$$

Then, the explicit numerical evaluation of (A.37) gives

$$
C_T(T_3) = 160.246.
\tag{A.44}
$$

### Type $\mathrm{II}_k(n+1)$ Chern-Simons matter theories

So far we focused on 3d SCFTs which have UV descriptions that involve no Chern-Simons couplings. Now, we consider the $U(1)_k \times U(1)_{-k}$ Chern-Simons matter theories of type $\mathrm{II}_k(n+1)$ [59], which in $\mathcal{N} = 4$ language are described by one vector multiplet and $n$ unit-charge hypermultiplets, one twisted vector multiplet and one unit-charge twisted hypermultiplet, with a BF coupling (between the vector and twisted vector multiplets) of even level $k$.

For illustration, we focus on the $\text{II}_k(3)$ theories, *i.e.* we fix $n = 2$. The squashed $S^3$ partition function is

$$
\begin{aligned}
Z_b^{\text{II}_k(3)}(m) &= \int_{\mathbb{R}} d\sigma d\tau \frac{e^{-ki\pi\sigma\tau}}{S_2(\pm i\sigma + Q/4|b)^2 S_2(\pm i\tau + Q/4|b)} \\
&= \int_{\mathbb{R}} d\sigma \frac{1}{S_2(\pm i\sigma + Q/4|b)^2 S_2(\pm ik\sigma/2 + Q/4|b)},
\end{aligned}
\tag{A.45}
$$

where in the second equality above we have used the Fourier transformation identity for the double-sine functions in [99]. Using the central charge formula (2.15) and the relation (A.8), we find

$$
\begin{aligned}
C_T(\text{II}_k(3)) &= -\frac{48}{\pi^2} \frac{1}{Z_b^{\text{CS}}} \frac{\partial^2 Z_b^{\text{CS}}}{\partial b^2}\bigg|_{b=1} \\
&= \frac{48}{\pi^2} \frac{1}{Z_{b=1}^{\text{CS}}} \int_{\mathbb{R}} \frac{d\sigma}{(2\cosh\pi\sigma)^2 (2\cosh\frac{\pi k\sigma}{2})} \left( \frac{2\pi(\pi + 4\pi\sigma^2 + 2\sigma\sinh 2\pi\sigma)}{4\cosh^2\pi\sigma} + \right. \\
&\qquad \left. + \frac{\pi(\pi + k^2\pi\sigma^2 + k\sigma\sinh\pi k\sigma)}{4\cosh^2\frac{\pi k\sigma}{2}} \right) \\
&= 34.5463, \, 34.7619, \, 35.0577, \, 35.2887, \, \ldots
\end{aligned}
\tag{A.46}
$$

for $k = 2, 4, 6, 8, \ldots$.

## A.4 Computation of $C_J$

Now, let us turn to the evaluation of the flavor central charges $C_J$ for a variety of examples. We shall use the formula (2.46), which relates mass deformations of the round $S^3$ partition function to the flavor central charge.

**Free hypermultiplets**

Let us start by considering $k$ free hypermultiplets, for which we turn on the following mass matrix $\mathcal{M} = \text{Diag}(m_1, m_2, \ldots, m_k, -m_k, \ldots, -m_2, -m_1) \in USp(2k)$. Then, it follows immediately from (A.13), that

$$
F(\mathcal{M}) = \sum_{i=1}^{k} \log 2\cosh(\pi m_i),
\tag{A.47}
$$

and therefore, using (2.46), we conclude

$$
C_J^{USp(2k)}(\text{hyper}_k) = 4.
\tag{A.48}
$$

In case we are interested in their $SU(k) \subset USp(2k)$ flavor symmetry, then the minimal achievable $C_J$ for free hypermultiplets is

$$
C_J^{SU(k)}(\text{hyper}_k) = \begin{cases} C_J^{USp(4)}(\text{hyper}_2) = 4, & k = 2, \\ I_{SU(k) \hookrightarrow USp(2k)} C_J^{USp(2k)}(\text{hyper}_k) = 8, & k > 2, \end{cases}
\tag{A.49}
$$

where as before we denote by $I$ the embedding index.

**$\mathbb{Z}_2$ gauged $k$ hypermultiplets**

The $USp(2k)$ 1-instanton moduli space is $\mathbb{C}^{2k}/\mathbb{Z}_2$, and can be realized by $k$ $\mathbb{Z}_2$ gauged hypermultiplets. However, since discrete gauging does not affect local correlators of $\mathbb{Z}_2$ invariant operators, we get

$$
C_J^{USp(2k)}(\text{hyper}_k/\mathbb{Z}_2) = 4,
\tag{A.50}
$$

*i.e.*, the free field value.

**SQED with $k$ unit charge hypermultiplets**

Let us now turn to the next example: the 3d $\mathcal{N} = 4$ SQED with $k$ unit charge hypermultiplets. The mass-deformed $S^3$ partition function is already computed in (A.18), where now the mass matrix is given by $\mathcal{M} = \mathrm{Diag}\,(m_1, \ldots, m_k, -m_k, \ldots, -m_1) \in USp(2k)$ subject to $\sum_{i=1}^{k} m_i = 0$ (note that $k \geq 2$ to satisfy this constraint). The formula (2.46) then gives

$$C_J^{SU(k)}(\mathrm{SQED}_k) = \frac{8k}{k+1}. \tag{A.51}$$

Notice that in the large $k$ limit, this formula reproduces (A.48). Moreover, for given flavor symmetry $SU(k)$, the interacting theory obtained in the infrared of the SQED has smaller $C_J$ than the one realizable by free theories. This means that the bootstrap gives us access to interacting theories despite the fact that we cannot exclude higher spin conserved currents from the onset.

**$SU(2)$ SYM with $k$ fundamental hypermultiplets**

Now, let us consider 3d $N = 4$ $SU(2)$ gauge theories with $k$ fundamental hypermultiplets; their Higgs branches have $SO(2k)$ flavor symmetry and are described by the 1-instanton moduli space of $SO(2k)$. The mass deformed $S^3$ partition function is then given by

$$Z_{b=1}^{\mathrm{SQCD}_{2k}}(m) = \int_{\mathbb{R}} d\sigma \frac{\sinh^2(2\pi\sigma)}{\prod_{i=1}^{k} 4 \cosh \pi(\pm\sigma + m_i)}, \tag{A.52}$$

where $\mathcal{M}_{SO(2k)} = \mathrm{Diag}\,(m_1, \ldots, m_k, -m_k, \ldots, -m_1)$. The formula (2.46) then gives

$$C_J^{SO(2k)}(\mathrm{SQCD}_{2k}) = \frac{32(k-2)}{2k-1}. \tag{A.53}$$

Notice that here we take $k \geq 3$, otherwise the quiver is not good (ugly or bad in the sense of [43]) and the naive $S^3$ partition function diverges.

**$SU(2)$ SYM with $k$ fundamental hypermultiplets and an additional half-hypermultiplet**

If we include an additional half-hypermultiplet in the doublet representation to the gauge theory described in the previous example, we obtain a Higgs branch with $SO(2k+1)$ flavor symmetry, described by the 1-instanton moduli space of $SO(2k+1)$.

Then, the $SO(2k+1)$ mass deformed $S^3$ partition function for this theory reads

$$Z_{b=1}^{\mathrm{SQCD}_{2k+1}} = \int_{\mathbb{R}} d\sigma \frac{\sinh^2(2\pi\sigma)}{2 \cosh(\pi\sigma) \prod_{i=1}^{k} 4 \cosh \pi(\pm\sigma + m_i)}, \tag{A.54}$$

where $\mathcal{M}_{SO(2k+1)} = \mathrm{Diag}\,(m_1, \ldots, m_k, 0, -m_k, \ldots, -m_1)$. Thus, the formula (2.46) gives

$$C_J^{SO(2k+1)}(\mathrm{SQCD}_{2k+1}) = \frac{16(2k-3)}{2k}. \tag{A.55}$$

Notice that here we take $k \geq 2$, since otherwise the naive $S^3$ partition function diverges. Thus, together we find that for 3d $\mathcal{N} = 4$ $SU(2)$ SYM with $n$ half-hypermultiplets in the doublet representation, the flavor central charge is given by[32]

$$C_J^{SO(n)}(\mathrm{SQCD}_n) = \frac{16(n-4)}{n-1}. \tag{A.56}$$

---

[32]It is also easy to see that the central charge $C_J$ of the $SU(k) \subset SO(2k)$ subgroup is bigger than the ones in (2.52).

**Affine quiver gauge theories**

Finally, let us turn to the $T_3$ theory; The mass-deformed partition function was computed in [68,69], and for convenience we shall recall it here,

$$Z_{b=1}^{T_3}\left(\vec{m}^{(1)}, \vec{m}^{(2)}, \vec{m}^{(3)}\right) = \frac{1}{3! \prod_{\alpha=1}^{3} \prod_{\ell<j} 2\sinh \pi m_{\ell j}^{(\alpha)}}$$
$$\times \sum_{\rho^{(\alpha)}\in S_3} (-1)^{\sum_\alpha \rho^{(\alpha)}} \left(\sum_\alpha m_{\rho^{(\alpha)}(1)}^{(\alpha)}\right) \coth\left(\sum_\alpha m_{\rho^{(\alpha)}(1)}^{(\alpha)}\right) \coth\left(\sum_\alpha m_{\rho^{(\alpha)}(3)}^{(\alpha)}\right),$$
(A.57)

where $S_3$ is the symmetric group and the mass parameters are turned on for $SU(3)^3 \subset E_6$. Then, we immediately find

$$F\big|_{\mathcal{M}^2} = \frac{12}{13}\pi^2 \sum_{i,\alpha=1}^{3} \left(m_i^{(\alpha)}\right)^2.$$
(A.58)

Then, using the formula (2.46), we conclude that the $SU(3)$ central charge is

$$C_J^{SU(3)}(T[SU(3)]) = \frac{192}{13}.$$
(A.59)

The $E_6$ representation, **27**, decomposes into $SU(3)$ representations as follows

$$\mathbf{27} \to (\mathbf{3,1,3}) \oplus (\mathbf{\bar{3},\bar{3},1}) \oplus (\mathbf{1,3,\bar{3}}).$$
(A.60)

Furthermore, we have the following quadratic indices of the representation (see *e.g.* [100])

$$T_{E_6}(\mathbf{27}) = 3, \quad T_{SU(3)}(\mathbf{3}) = \frac{1}{2}, \quad T_{SU(3)}(\mathbf{1}) = 0.$$
(A.61)

Then, the embedding index is given by

$$I_{SU(3)\hookrightarrow E_6} = \frac{6T_{SU(3)}(\mathbf{3}) + 9T_{SU(3)}(\mathbf{1})}{T_{E_6}(\mathbf{27})} = 1,$$
(A.62)

and we conclude that

$$C_J^{E_6}(T_3) = C_J^{SU(3)}(T[SU(3)]) = \frac{192}{13}.$$
(A.63)

**Type $\mathrm{II}_k(n+1)$ Chern-Simons matter theories**

The $U(1)_k \times U(1)_{-k}$ Chern-Simons matter theory of type $\mathrm{II}_k(n+1)$ [59] with even $k$ has a Higgs branch of quaternionic dimension $n$ from the hypermultiplets, and a Coulomb branch given by $\mathbb{C}^2/\mathbb{Z}_{k/2+n}$ from the twisted hypermultiplets. The Higgs and Coulomb branch flavor symmetries are $SU(n) \times U(1)$ and $U(1)$, respectively. For $k = 2$, the Higgs branch symmetry is enhanced to $SU(n+1)$ and the theory is known to flow to the same IR SCFT as the SQED with $n+1$ flavors.

Here, we consider the type $\mathrm{II}_k(3)$ Chern-Simons matter theories with $G_\mathrm{H} = SU(2) \times U(1)$ and focus on the $SU(2)$ factor. The $SU(2)$-mass deformed $S^3$ partition function is

$$Z_{b=1}^{\mathrm{II}_k(3)}(m) = \int_\mathbb{R} d\sigma d\tau \frac{e^{-ki\pi\sigma\tau}}{8\cosh\pi(\pm\sigma+m)\cosh(\pi\tau)}$$
$$= \int_\mathbb{R} d\sigma \frac{1}{8\cosh\pi(\pm\sigma+m)\cosh(\pi k\sigma/2)},$$
(A.64)

giving rise to

$$C_J^{SU(2)}(\mathrm{II}_k(3)) = \frac{8\int dx\, \mathrm{sech}^4(\pi x)\,\mathrm{sech}(\pi kx/2)}{\int dx\, \mathrm{sech}^2(\pi x)\,\mathrm{sech}(\pi kx/2)} = 6, 6.65633, 7.06667, 7.3258, \ldots \quad \text{(A.65)}$$

for $k = 2, 4, 6, 8\ldots$.

# B Derivation of the superconformal Casimir equations

In this appendix we detail the derivation of the superconformal Casimir equations for the $s$- and $t$-channels of the mixed Coulomb and Higgs branch four-point functions.

## B.1 $s$-channel

We first consider the four-point function

$$\langle \mathcal{O}^{\mathrm{C}}(x_1, Y_1)\mathcal{O}^{\mathrm{C}}(x_2, Y_2)\mathcal{O}^{\mathrm{H}}(x, \widetilde{Y}_3)\mathcal{O}^{\mathrm{H}}(x, \widetilde{Y}_4)\rangle = \left(\frac{(Y_1 \cdot Y_2)(\widetilde{Y}_3 \cdot \widetilde{Y}_4)}{|x_{12}||x_{34}|}\right)^2 G_{\mathrm{mixed}}(u, v), \qquad \text{(B.1)}$$

where the operators $\mathcal{O}^{\mathrm{C}}(x, Y)$ and $\mathcal{O}^{\mathrm{H}}(x, \widetilde{Y})$ are defined as

$$\mathcal{O}^{\mathrm{C}}(x, Y) = Y^{A_1}\cdots Y^{A_k}\mathcal{O}^{\mathrm{C}}_{A_1\cdots A_k}(x), \quad \mathcal{O}^{\mathrm{H}}(x, \widetilde{Y}) = \widetilde{Y}^{\dot{A}_1}\cdots\widetilde{Y}^{\dot{A}_k}\mathcal{O}^{\mathrm{H}}_{\dot{A}_1\cdots\dot{A}_k}(x), \qquad \text{(B.2)}$$

where $\mathcal{O}^{\mathrm{C}}_{A_1\cdots A_k}(x)$ and $\mathcal{O}^{\mathrm{H}}_{\dot{A}_1\cdots\dot{A}_k}(x)$ are Coulomb branch and Higgs branch operators, respectively. For $k = 2$, we have

$$\langle \mathcal{O}^{\mathrm{C}}_{++}(x_1)\mathcal{O}^{\mathrm{C}}_{--}(x_2)\mathcal{O}^{\mathrm{H}}_{\dot{+}\dot{+}}(x_3)\mathcal{O}^{\mathrm{H}}_{\dot{-}\dot{-}}(x_4)\rangle = \frac{1}{|x_{12}|^2|x_{34}|^2} G_{\mathrm{mixed}}(u, v), \qquad \text{(B.3)}$$

with the following cross ratios

$$u = \frac{|x_{12}|^2|x_{34}|^2}{|x_{13}|^2|x_{24}|^2}, \quad v = \frac{|x_{14}|^2|x_{23}|^2}{|x_{13}|^2|x_{24}|^2}. \qquad \text{(B.4)}$$

Now, the Casimir operator acts on the four-point function as a differential operator, *i.e.*

$$\langle [C, \mathcal{O}^{\mathrm{C}}_{++}(x_1)\mathcal{O}^{\mathrm{C}}_{--}(x_2)]\mathcal{O}^{\mathrm{H}}_{\dot{+}\dot{+}}(x_3)\mathcal{O}^{\mathrm{H}}_{\dot{-}\dot{-}}(x_4)\rangle = \frac{1}{|x_{12}|^2|x_{34}|^2}\mathcal{C}^s G_{\mathrm{mixed}}(u, v), \qquad \text{(B.5)}$$

and we may solve for the superconformal blocks via the (eigenvalue) equation

$$\mathcal{C}^s \mathcal{A}^s_{\Delta, \ell, j_{\mathrm{C}}, j_{\mathrm{H}}}(u, v) = \lambda_{\mathrm{C}} \mathcal{A}^s_{\Delta, \ell, j_{\mathrm{C}}, j_{\mathrm{H}}}(u, v). \qquad \text{(B.6)}$$

We can generically write the differential operator $\mathcal{C}^s$ as

$$\mathcal{C}^s = \mathcal{C}^s_b + \mathcal{C}^s_{SQ} + \mathcal{C}^s_R, \qquad \text{(B.7)}$$

where by the results by Dolan and Osborn [101], we can read off the piece from the bosonic subalgebra,

$$\mathcal{C}^s_b = 2z^2(1-z)\partial^2 + 2\bar{z}^2(1-\bar{z})\bar{\partial}^2 - 2(z^2\partial + \bar{z}^2\bar{\partial}) + 2\frac{z\bar{z}}{z-\bar{z}}[(1-z)\partial - (1-\bar{z})\bar{\partial}]. \qquad \text{(B.8)}$$

Furthermore, for our purposes, the intermediate operators should be R-symmetry singlets, and thus we have

$$\mathcal{C}^s_R = 0. \qquad \text{(B.9)}$$

Thus, it remains to compute $\mathcal{C}_{SQ}$. The BPS conditions for the moment map operators are

$$\begin{aligned}
[S^{\alpha A\dot{A}}, \mathcal{O}^{\mathrm{C}}_{AB}(0)] &= [S^{\alpha A\dot{A}}, \mathcal{O}^{\mathrm{H}}_{\dot{A}\dot{B}}(0)] &= 0, \\
[Q_{\alpha+\dot{A}}, \mathcal{O}^{\mathrm{C}}_{++}(x)] &= [Q_{\alpha-\dot{A}}, \mathcal{O}^{\mathrm{C}}_{--}(x)] &= 0, \\
[Q_{\alpha A\dot{+}}, \mathcal{O}^{\mathrm{H}}_{\dot{+}\dot{+}}(x)] &= [Q_{\alpha A\dot{-}}, \mathcal{O}^{\mathrm{H}}_{\dot{-}\dot{-}}(x)] &= 0,
\end{aligned} \qquad \text{(B.10)}$$

and the superconformal $S$-supercharge acts on a superconformal primary at position $x$ as follows

$$[S^{\alpha A \dot{A}}, \mathcal{O}(x)] = -ix_\mu \epsilon^{AB} \epsilon^{\dot{A}\dot{B}} \sigma_\mu^{\alpha\beta} [Q_{\beta B \dot{B}}, \mathcal{O}(x)], \tag{B.11}$$

where we have used the commutators in (3.1) as well as

$$\mathcal{O}(x) = e^{ix \cdot P} \mathcal{O}(0) e^{-ix \cdot P}. \tag{B.12}$$

To compute the differential operator $\mathcal{C}_{SQ}$, we consider

$$\frac{1}{2}[S^{\alpha A \dot{A}}, Q_{\alpha A \dot{A}}] \mathcal{O}^{\mathrm{C}}_{++}(x_1) \mathcal{O}^{\mathrm{C}}_{--}(x_2)|0\rangle$$
$$= \left\{ -ix_{12,\mu} \epsilon^{\dot{A}\dot{B}} \sigma_\mu^{\alpha\beta} [Q_{\beta -\dot{B}}, \mathcal{O}^{\mathrm{C}}_{++}(x_1)][Q_{\alpha +\dot{A}}, \mathcal{O}^{\mathrm{C}}_{--}(x_2)] + 8\mathcal{O}^{\mathrm{C}}_{++}(x_1)\mathcal{O}^{\mathrm{C}}_{--}(x_2) \right\}|0\rangle. \tag{B.13}$$

Now, given the following Ward identities

$$0 = \langle \{Q_{\alpha++}, [Q_{\beta--}, \mathcal{O}^{\mathrm{C}}_{++}(x_1)]\mathcal{O}^{\mathrm{C}}_{--}(x_2)\mathcal{O}^{\mathrm{H}}_{++}(x_3)\mathcal{O}^{\mathrm{H}}_{--}(x_4)\} \rangle,$$
$$0 = \langle \{Q_{\beta-+}, \mathcal{O}^{\mathrm{C}}_{++}(x_1)[Q_{\alpha+-}, \mathcal{O}^{\mathrm{C}}_{--}(x_2)]\mathcal{O}^{\mathrm{H}}_{++}(x_3)\mathcal{O}^{\mathrm{H}}_{--}(x_4)\} \rangle, \tag{B.14}$$

we find

$$\langle [Q_{\beta--}, \mathcal{O}^{\mathrm{C}}_{++}(x_1)][Q_{\alpha++}, \mathcal{O}^{\mathrm{C}}_{--}(x_2)]\mathcal{O}^{\mathrm{H}}_{++}(x_3)\mathcal{O}^{\mathrm{H}}_{--}(x_4) \rangle$$
$$+ \langle [Q_{\beta--}, \mathcal{O}^{\mathrm{C}}_{++}(x_1)]\mathcal{O}^{\mathrm{C}}_{--}(x_2)\mathcal{O}^{\mathrm{H}}_{++}(x_3)[Q_{\alpha++}, \mathcal{O}^{\mathrm{H}}_{--}(x_4)] \rangle \tag{B.15}$$
$$= -i(\sigma^\mu)_{\alpha\beta} \partial_{x_{1,\mu}} \langle \mathcal{O}^{\mathrm{C}}_{++}(x_1)\mathcal{O}^{\mathrm{C}}_{--}(x_2)\mathcal{O}^{\mathrm{H}}_{++}(x_3)\mathcal{O}^{\mathrm{H}}_{--}(x_4) \rangle,$$

as well as

$$\langle [Q_{\beta-+}, \mathcal{O}^{\mathrm{C}}_{++}(x_1)][Q_{\alpha+-}, \mathcal{O}^{\mathrm{C}}_{--}(x_2)]\mathcal{O}^{\mathrm{H}}_{++}(x_3)\mathcal{O}^{\mathrm{H}}_{--}(x_4) \rangle$$
$$- \langle \mathcal{O}^{\mathrm{C}}_{++}(x_1)[Q_{\alpha+-}, \mathcal{O}^{\mathrm{C}}_{--}(x_2)]\mathcal{O}^{\mathrm{H}}_{++}(x_3)[Q_{\beta-+}, \mathcal{O}^{\mathrm{H}}_{--}(x_4)] \rangle \tag{B.16}$$
$$= -i(\sigma^\mu)_{\alpha\beta} \partial_{x_{2,\mu}} \langle \mathcal{O}^{\mathrm{C}}_{++}(x_1)\mathcal{O}^{\mathrm{C}}_{--}(x_2)\mathcal{O}^{\mathrm{H}}_{++}(x_3)\mathcal{O}^{\mathrm{H}}_{--}(x_4) \rangle.$$

Therefore, taking the $x_4 \to \infty$ limit we obtain

$$-ix_{12,\mu} \sigma_\mu^{\alpha\beta} \langle [Q_{\beta--}, \mathcal{O}^{\mathrm{C}}_{++}(x_1)][Q_{\alpha++}, \mathcal{O}^{\mathrm{C}}_{--}(x_2)]\mathcal{O}^{\mathrm{H}}_{++}(x_3)\mathcal{O}^{\mathrm{H}}_{--}(x_4) \rangle$$
$$\sim 2x_{12,\mu} \partial_{x_{1,\mu}} \langle \mathcal{O}^{\mathrm{C}}_{++}(x_1)\mathcal{O}^{\mathrm{C}}_{--}(x_2)\mathcal{O}^{\mathrm{H}}_{++}(x_3)\mathcal{O}^{\mathrm{H}}_{--}(x_4) \rangle, \tag{B.17}$$

and

$$ix_{12,\mu} \sigma_\mu^{\alpha\beta} \langle [Q_{\beta-+}, \mathcal{O}^{\mathrm{C}}_{++}(x_1)][Q_{\alpha+-}, \mathcal{O}^{\mathrm{C}}_{--}(x_2)]\mathcal{O}^{\mathrm{H}}_{++}(x_3)\mathcal{O}^{\mathrm{H}}_{--}(x_4) \rangle$$
$$\sim 2x_{21,\mu} \partial_{x_{2,\mu}} \langle \mathcal{O}^{\mathrm{C}}_{++}(x_1)\mathcal{O}^{\mathrm{C}}_{--}(x_2)\mathcal{O}^{\mathrm{H}}_{++}(x_3)\mathcal{O}^{\mathrm{H}}_{--}(x_4) \rangle. \tag{B.18}$$

On the one hand, for the case in equation (B.17), we consider the parametrization

$$x_1^\mu = \left( 1 + \frac{1}{2}\left( \frac{z}{1-z} + \frac{\bar{z}}{1-\bar{z}} \right), \frac{1}{2}\left( \frac{z}{1-z} - \frac{\bar{z}}{1-\bar{z}} \right), 0 \right),$$
$$x_2^\mu = (1, 0, 0), \qquad x_3^\mu = \frac{x_4^\mu}{|x_4|^2} = (0, 0, 0), \tag{B.19}$$

and thus, we have

$$u = z\bar{z}, \quad v = (1-z)(1-\bar{z}),$$
$$x_{12}^\mu \frac{\partial}{\partial x_{12}^\mu} = z(1-z)\frac{\partial}{\partial z} + \bar{z}(1-\bar{z})\frac{\partial}{\partial \bar{z}}. \tag{B.20}$$

On the other hand, for the case in (B.18), we consider the parametrization

$$x_1^\mu = (1, 0, 0), \quad x_2^\mu = \left( 1 - \frac{1}{2}(z+\bar{z}), \frac{1}{2}(z-\bar{z}), 0 \right), \quad x_3^\mu = \frac{x_4^\mu}{|x_4|^2} = (0, 0, 0), \tag{B.21}$$

and thus, we have

$$u = z\bar{z}, \quad v = (1-z)(1-\bar{z}),$$
$$x_{21}^{\mu} \frac{\partial}{\partial x_2^{\mu}} = z \frac{\partial}{\partial z} + \bar{z} \frac{\partial}{\partial \bar{z}}. \tag{B.22}$$

Putting all the pieces together, we end up with the following expression for $\mathcal{C}_{SQ}^s$,

$$\mathcal{C}_{SQ}^s = 2 \left[ z(1-z)\frac{\partial}{\partial z} + \bar{z}(1-\bar{z})\frac{\partial}{\partial \bar{z}} \right] + 2 \left[ z\frac{\partial}{\partial z} + \bar{z}\frac{\partial}{\partial \bar{z}} \right]. \tag{B.23}$$

### B.2   $t$-channel

We now proceed by computing the $t$-channel Casimir operator, *i.e.* we are deriving $\mathcal{C}^t$ in

$$\langle [C, \mathcal{O}_{++}^H(x_1)\mathcal{O}_{--}^C(x_2)]\mathcal{O}_{++}^C(x_3)\mathcal{O}_{--}^H(x_4)\rangle = \frac{1}{|x_{12}|^2|x_{34}|^2} \mathcal{C}^t \left[ \frac{u}{v} G_{\text{mixed}}(v, u) \right]. \tag{B.24}$$

Again, we can write the differential operator $\mathcal{C}^t$ as follows

$$\mathcal{C}^t = \mathcal{C}_b^t + \mathcal{C}_{SQ}^t + \mathcal{C}_R^t, \tag{B.25}$$

and as before, given [101], we may read off

$$\mathcal{C}_b^t = 2z^2(1-z)\partial^2 + 2\bar{z}^2(1-\bar{z})\bar{\partial}^2 - 2(z^2\partial + \bar{z}^2\bar{\partial}) + 2\frac{z\bar{z}}{z-\bar{z}}[(1-z)\partial - (1-\bar{z})\bar{\partial}]. \tag{B.26}$$

However, now the intermediate operator are in the representation $(2j_C, 2j_H) = (2, 2)$ of the $SU(2)_C \times SU(2)_H$ R-symmetry, and so we have

$$\mathcal{C}_R^t = -2 - 2 = -4. \tag{B.27}$$

Finally, we compute the remnant piece, $\mathcal{C}_{SQ}^t$. To do so, we consider

$$\frac{1}{2}[S^{\alpha A\dot{A}}, Q_{\alpha A\dot{A}}]\mathcal{O}_{++}^H(x_1)\mathcal{O}_{--}^C(x_2)|0\rangle$$
$$= \left\{ -ix_{12,\mu}\sigma_\mu^{\alpha\beta}[Q_{\beta--}, \mathcal{O}_{++}^H(x_1)][Q_{\alpha++}, \mathcal{O}_{--}^C(x_2)] + 8\mathcal{O}_{++}^H(x_1)\mathcal{O}_{--}^C(x_2) \right\}|0\rangle. \tag{B.28}$$

Now, the Ward identity

$$0 = \langle \{Q_{\alpha++}, [Q_{\beta--}, \mathcal{O}_{++}^H(x_1)]\mathcal{O}_{--}^C(x_2)\mathcal{O}_{++}^C(x_3)\mathcal{O}_{--}^H(x_4)\} \rangle, \tag{B.29}$$

gives

$$\langle [Q_{\beta--}, \mathcal{O}_{++}^H(x_1)][Q_{\alpha++}, \mathcal{O}_{--}^C(x_2)]\mathcal{O}_{++}^C(x_3)\mathcal{O}_{--}^H(x_4)\rangle$$
$$+ \langle [Q_{\beta--}, \mathcal{O}_{++}^H(x_1)]\mathcal{O}_{--}^C(x_2)\mathcal{O}_{++}^C(x_3)[Q_{\alpha++}, \mathcal{O}_{--}^H(x_4)]\rangle \tag{B.30}$$
$$= -i(\sigma^\mu)_{\alpha\beta}\partial_{x_{1,\mu}}\langle \mathcal{O}_{++}^H(x_1)\mathcal{O}_{--}^C(x_2)\mathcal{O}_{++}^C(x_3)\mathcal{O}_{--}^H(x_4)\rangle.$$

Therefore, in the $x_4 \to \infty$ limit we obtain

$$-ix_{12,\mu}\sigma_\mu^{\alpha\beta}\langle [Q_{\beta--}, \mathcal{O}_{++}^H(x_1)][Q_{\alpha++}, \mathcal{O}_{--}^C(x_2)]\mathcal{O}_{++}^C(x_3)\mathcal{O}_{--}^H(x_4)\rangle$$
$$\sim 2x_{12,\mu}\partial_{x_{1,\mu}}\langle \mathcal{O}_{++}^H(x_1)\mathcal{O}_{--}^C(x_2)\mathcal{O}_{++}^C(x_3)\mathcal{O}_{--}^H(x_4)\rangle. \tag{B.31}$$

Putting everything together, we end up with

$$\mathcal{C}_{SQ}^t = 2 \left[ z(1-z)\frac{\partial}{\partial z} + \bar{z}(1-\bar{z})\frac{\partial}{\partial \bar{z}} \right] + 4. \tag{B.32}$$

## C  Crossing matrices (6j symbols)

Consider a simple Lie group $G$, the tensor product of two adjoint representations decomposes as

$$(\mathbf{adj} \otimes \mathbf{adj})_S = \mathbf{1} \oplus \mathbf{2adj} \oplus \bigoplus_i \mathcal{R}_{(S,i)},$$

$$(\mathbf{adj} \otimes \mathbf{adj})_A = \mathbf{adj} \oplus \bigoplus_i \mathcal{R}_{(A,i)}, \tag{C.1}$$

where $\mathbf{2adj}$ denotes the representation whose Dynkin label is twice the Dynkin label of the adjoint representation, and $\mathcal{R}_{(S,i)}$ ($\mathcal{R}_{(A,i)}$) are representations that appear in the symmetric (antisymmetric) tensor product and which are not $\mathbf{1}$, $\mathbf{adj}$ and $\mathbf{2adj}$.

Let us denote the Clebsch-Gordan coefficients by

$$(\Gamma_{\mathbf{1}})_{ab} = \frac{1}{\sqrt{|G|}} \delta_{ab}, \quad (\Gamma_{\mathbf{adj}})_{ab}^c = \frac{1}{\sqrt{2h^\vee}} f_{abc}, \quad (\Gamma_{\mathbf{2adj}})_{ab}^\alpha, \quad (\Gamma_{\mathcal{R}_{(S,i)}})_{ab}^{I_{(S,i)}}, \quad (\Gamma_{\mathcal{R}_{(A,i)}})_{ab}^{I_{(A,i)}}, \tag{C.2}$$

where $a = 1, \cdots, \dim(G)$, $I_{(S,i)} = 1, \cdots, \dim(\mathcal{R}_{(S,i)})$, $I_{(A,i)} = 1, \cdots, \dim(\mathcal{R}_{(A,i)})$, and $f^{abc}$ is the structure constant of $G$, which is normalized by

$$f^{aed} f^{bde} = 2h^\vee \delta^{ab}. \tag{C.3}$$

The other Clebsch-Gordan coefficients are normalized by

$$(\Gamma_{\mathbf{2adj}})_{ab}^\alpha (\Gamma_{\mathbf{2adj}})_{ab}^\beta = \delta^{\alpha\beta}, \quad (\Gamma_{\mathcal{R}_{(S,i)}})_{ab}^{I_{(S,i)}} (\Gamma_{\mathcal{R}_{(S,i)}})_{ab}^{J_{(S,i)}} = \delta^{I_{(S,i)} J_{(S,i)}},$$

$$(\Gamma_{\mathcal{R}_{(A,i)}})_{ab}^{I_{(A,i)}} (\Gamma_{\mathcal{R}_{(A,i)}})_{ab}^{J_{(A,i)}} = \delta^{I_{(A,i)} J_{(A,i)}}. \tag{C.4}$$

The projection matrices are defined as

$$P_{\mathcal{R}}^{abcd} = (\Gamma_{\mathcal{R}})_{I_{\mathcal{R}}}^{ab} (\Gamma_{\mathcal{R}})_{J_{\mathcal{R}}}^{cd} \delta^{I_{\mathcal{R}} J_{\mathcal{R}}}, \tag{C.5}$$

and the crossing matrix (6j symbol) is defined as follows

$$F_i{}^j = \frac{1}{\dim(\mathcal{R}_j)} P_{\mathcal{R}_i}^{dabc} P_{\mathcal{R}_j}^{abcd}. \tag{C.6}$$

## D  Crossing equations in matrix form

In this appendix, we rewrite the crossing equations (5.4), (5.7), (5.10), and (5.12) in forms that are more explicit and readily usable for semidefinite programming.

The crossing equations for theories with $U(1) \times U(1)$ flavor symmetry (5.4) can be rewritten

as

$$
0 = \begin{pmatrix} 2v\mathcal{A}^s_{\mathcal{B}[0]^{(0;0)}_0}(u,v) \\ \mathcal{F}_{\mathcal{B}[0]^{(0;0)}_0}(u,v,w) \\ \mathcal{F}_{\mathcal{B}[0]^{(0;0)}_0}(u,v,\widetilde{w}) \end{pmatrix} - \sum_{\Delta,\ell} \lambda^2_{\mathrm{C,H},(\mathcal{L},\Delta,\ell)} \begin{pmatrix} 2u\mathcal{A}^t_{\Delta,\ell,0,0}(v,u) \\ 0 \\ 0 \end{pmatrix}
$$

$$
+ \sum_{\Delta,\ell} \sum_i \begin{pmatrix} \lambda^i_{\mathrm{C,C},\mathcal{L}[2\ell]^{(0;0)}_\Delta} & \lambda^i_{\mathrm{H,H},\mathcal{L}[2\ell]^{(0;0)}_\Delta} \end{pmatrix} \begin{pmatrix} \begin{pmatrix} 0 & v\mathcal{A}^s_{\mathcal{L}[2\ell]^{(0;0)}_\Delta}(u,v) \\ v\mathcal{A}^s_{\mathcal{L}[2\ell]^{(0;0)}_\Delta}(u,v) & 0 \end{pmatrix} \\ \begin{pmatrix} \mathcal{F}_{\mathcal{L}[2\ell]^{(0;0)}_\Delta}(u,v,w) & 0 \\ 0 & 0 \end{pmatrix} \\ \begin{pmatrix} 0 & 0 \\ 0 & \mathcal{F}_{\mathcal{L}[2\ell]^{(0;0)}_\Delta}(u,v,\widetilde{w}) \end{pmatrix} \end{pmatrix} \begin{pmatrix} \lambda^i_{\mathrm{C,C},\mathcal{L}[2\ell]^{(0;0)}_\Delta} \\ \lambda^i_{\mathrm{H,H},\mathcal{L}[2\ell]^{(0;0)}_\Delta} \end{pmatrix}
$$

$$
+ \sum_\ell \left[ \lambda^2_{\mathrm{C,C},\mathcal{A}[2\ell]^{(2;0)}_{\ell+2}} \begin{pmatrix} 0 \\ \mathcal{F}_{\mathcal{A}[2\ell]^{(2;0)}_{\ell+2}}(u,v,w) \\ 0 \end{pmatrix} + \lambda^2_{\mathrm{H,H},\mathcal{A}[2\ell]^{(0;2)}_{\ell+2}} \begin{pmatrix} 0 \\ 0 \\ \mathcal{F}_{\mathcal{A}[2\ell]^{(0;2)}_{\ell+2}}(u,v,\widetilde{w}) \end{pmatrix} \right]
$$

$$
+ \lambda^2_{\mathrm{C,C},\mathcal{B}[0]^{(4;0)}_2} \begin{pmatrix} 0 \\ \mathcal{F}_{\mathcal{B}[0]^{(4;0)}_2}(u,v,w) \\ 0 \end{pmatrix} + \lambda^2_{\mathrm{H,H},\mathcal{B}[0]^{(0;4)}_2} \begin{pmatrix} 0 \\ 0 \\ \mathcal{F}_{\mathcal{B}[0]^{(0;4)}_2}(u,v,\widetilde{w}) \end{pmatrix}.
$$

$$(D.1)$$

Here, the sum over $\mathcal{A}[2\ell]^{(0;0)}_{\ell+1}$ is omitted because these blocks are smooth $\Delta \to 1$ limits of the $\mathcal{L}[2\ell]^{(0;0)}_\Delta$ blocks. In the second line, we explicitly included a sum over degenerate multiplets labeled by $i$ with the same $\Delta, \ell$. By contrast, in the other lines, we simply defined $\lambda^2$ to be the sum of degenerate contributions.

For the $(U(1) \times U(1)) \rtimes \mathbb{Z}_2$ flavor symmetry, the crossing equations (5.7) can be rewritten as

$$
0 = \begin{pmatrix} 2v\mathcal{A}^s_{\mathcal{B}[0]^{(0;0)}_0}(u,v) \\ \mathcal{F}_{\mathcal{B}[0]^{(0;0)}_0}(u,v,w) \end{pmatrix} - \sum_{\Delta,\ell} \lambda^2_{\mathrm{C,H},(\mathcal{L},\Delta,\ell)} \begin{pmatrix} 2u\mathcal{A}^t_{\Delta,\ell,0,0}(v,u) \\ 0 \end{pmatrix}
$$

$$
+ \sum_{\Delta,\ell} \lambda^2_{\mathrm{C,C},\mathcal{L}[2\ell]^{(0;0)}_\Delta,+} \begin{pmatrix} 2v\mathcal{A}^s_{\mathcal{L}[2\ell]^{(0;0)}_\Delta}(u,v) \\ \mathcal{F}_{\mathcal{L}[2\ell]^{(0;0)}_\Delta}(u,v,w) \end{pmatrix} + \sum_{\Delta,\ell} \lambda^2_{\mathrm{C,C},\mathcal{L}[2\ell]^{(0;0)}_\Delta,-} \begin{pmatrix} -2v\mathcal{A}^s_{\mathcal{L}[2\ell]^{(0;0)}_\Delta}(u,v) \\ \mathcal{F}_{\mathcal{L}[2\ell]^{(0;0)}_\Delta}(u,v,w) \end{pmatrix} \quad (D.2)
$$

$$
+ \sum_\ell \lambda^2_{\mathrm{C,C},\mathcal{A}[2\ell]^{(2;0)}_{\ell+2}} \begin{pmatrix} 0 \\ \mathcal{F}_{\mathcal{A}[2\ell]^{(2;0)}_{\ell+2}}(u,v,w) \end{pmatrix} + \lambda^2_{\mathrm{C,C},\mathcal{B}[0]^{(4;0)}_2} \begin{pmatrix} 0 \\ \mathcal{F}_{\mathcal{B}[0]^{(4;0)}_2}(u,v,w) \end{pmatrix}.
$$

For the most general $G_\mathrm{C} \times G_\mathrm{H}$ flavor symmetry, the crossing equations (5.10) can be rewrit-

ten as

$$
0 = \begin{pmatrix} 2v\mathcal{A}^s_{\mathcal{B}[0]_0^{(0;0)}}(u,v) \\ \mathcal{F}_{\mathcal{B}[0]_0^{(0;0)},*}{}^{\mathbf{1}}(u,v,w) \\ \mathcal{F}_{\mathcal{B}[0]_0^{(0;0)},*}{}^{\mathbf{1}}(u,v,\widetilde{w}) \end{pmatrix} + \lambda^2_{\mathrm{C,C},\mathcal{A}[0]_1^{(0;0)},\mathbf{1}} \begin{pmatrix} -2v\mathcal{A}^s_{\mathcal{A}[0]_1^{(0;0)}}(u,v) \\ \mathcal{F}_{\mathcal{A}[0]_1^{(0;0)},*}{}^{\mathbf{1}}(u,v,w) \\ \mathcal{F}_{\mathcal{A}[0]_1^{(0;0)},*}{}^{\mathbf{1}}(u,v,\widetilde{w}) \end{pmatrix}
$$

$$
+ \sum_{\Delta,\ell} \begin{pmatrix} \lambda_{\mathrm{C,C},\mathcal{L}[2\ell]_\Delta^{(0;0)},\mathbf{1}} & \lambda_{\mathrm{H,H},\mathcal{L}[2\ell]_\Delta^{(0;0)},\mathbf{1}} \end{pmatrix} \left( \begin{pmatrix} \begin{pmatrix} 0 & v\mathcal{A}^s_{\mathcal{L}[2\ell]_\Delta^{(0;0)}}(u,v) \\ v\mathcal{A}^s_{\mathcal{L}[2\ell]_\Delta^{(0;0)}}(u,v) & 0 \end{pmatrix} \\ \begin{pmatrix} \mathcal{F}_{\mathcal{L}[2\ell]_\Delta^{(0;0)},*}{}^{\mathbf{1}}(u,v,w) & 0 \\ 0 & 0 \end{pmatrix} \\ \begin{pmatrix} 0 & 0 \\ 0 & \mathcal{F}_{\mathcal{L}[2\ell]_\Delta^{(0;0)},*}{}^{\mathbf{1}}(u,v,\widetilde{w}) \end{pmatrix} \end{pmatrix} \right) \begin{pmatrix} \lambda_{\mathrm{C,C},\mathcal{L}[2\ell]_\Delta^{(0;0)},\mathbf{1}} \\ \lambda_{\mathrm{H,H},\mathcal{L}[2\ell]_\Delta^{(0;0)},\mathbf{1}} \end{pmatrix}
$$

$$
+ \lambda^2_{\mathrm{C,C},\mathcal{B}[0]_1^{(2;0)},\mathbf{adj}_\mathrm{C}} \begin{pmatrix} 0 \\ \mathcal{F}_{\mathcal{B}[0]_1^{(2;0)},*}{}^{\mathbf{adj}_\mathrm{C}}(u,v,w) \\ 0_* \end{pmatrix} + \lambda^2_{\mathrm{H,H},\mathcal{B}[0]_1^{(0;2)},\mathbf{adj}_\mathrm{H}} \begin{pmatrix} 0 \\ 0_* \\ \mathcal{F}_{\mathcal{B}[0]_1^{(0;2)},*}{}^{\mathbf{adj}_\mathrm{H}}(u,v,\widetilde{w}) \end{pmatrix}
$$

$$
+ \left[ \sum_{\mathbf{r}_\mathrm{C}} \lambda^2_{\mathrm{C,C},\mathcal{B}[0]_2^{(4;0)},\mathbf{r}_\mathrm{C}} \begin{pmatrix} 0 \\ \mathcal{F}_{\mathcal{B}[0]_2^{(4;0)},*}{}^{\mathbf{r}_\mathrm{C}}(u,v,w) \\ 0_* \end{pmatrix} + \sum_{\mathbf{r}_\mathrm{H}} \lambda^2_{\mathrm{H,H},\mathcal{B}[0]_2^{(0;4)},\mathbf{r}_\mathrm{H}} \begin{pmatrix} 0 \\ 0_* \\ \mathcal{F}_{\mathcal{B}[0]_2^{(0;4)},*}{}^{\mathbf{r}_\mathrm{H}}(u,v,\widetilde{w}) \end{pmatrix} \right]
$$

$$
+ \sum_\ell \left[ \sum_{\mathbf{r}_\mathrm{C}} \lambda^2_{\mathrm{C,C},\mathcal{A}[2\ell]_{\ell+2}^{(2;0)},\mathbf{r}_\mathrm{C}} \begin{pmatrix} 0 \\ \mathcal{F}_{\mathcal{A}[2\ell]_{\ell+2}^{(2;0)},*}{}^{\mathbf{r}_\mathrm{C}}(u,v,w) \\ 0_* \end{pmatrix} + \sum_{\mathbf{r}_\mathrm{H}} \lambda^2_{\mathrm{H,H},\mathcal{A}[2\ell]_{\ell+2}^{(0;2)},\mathbf{r}_\mathrm{H}} \begin{pmatrix} 0 \\ 0_* \\ \mathcal{F}_{\mathcal{A}[2\ell]_{\ell+2}^{(0;2)},*}{}^{\mathbf{r}_\mathrm{H}}(u,v,\widetilde{w}) \end{pmatrix} \right]
$$

$$
+ \sum_{\Delta,\ell} \left[ \sum_{\mathbf{r}_\mathrm{C}\neq\mathbf{1}} \lambda^2_{\mathrm{C,C},\mathcal{L}[2\ell]_\Delta^{(0;0)},\mathbf{r}_\mathrm{C}} \begin{pmatrix} 0 \\ \mathcal{F}_{\mathcal{L}[2\ell]_\Delta^{(0;0)},*}{}^{\mathbf{r}_\mathrm{C}}(u,v,w) \\ 0_* \end{pmatrix} + \sum_{\mathbf{r}_\mathrm{H}\neq\mathbf{1}} \lambda^2_{\mathrm{H,H},\mathcal{L}[2\ell]_\Delta^{(0;0)},\mathbf{r}_\mathrm{H}} \begin{pmatrix} 0 \\ 0_* \\ \mathcal{F}_{\mathcal{L}[2\ell]_\Delta^{(0;0)},*}{}^{\mathbf{r}_\mathrm{H}}(u,v,\widetilde{w}) \end{pmatrix} \right]
$$

$$
- \sum_{\Delta,\ell} \lambda^2_{\mathrm{C,H},(\mathcal{L},\Delta,\ell)} \begin{pmatrix} 2u\mathcal{A}^t_{\Delta,\ell,0,0}(v,u) \\ 0_* \\ 0_* \end{pmatrix},
$$

$$
\tag{D.3}
$$

where $*$ represents a column vector over representations in $\mathbf{adj}_\mathrm{C} \otimes \mathbf{adj}_\mathrm{H}$ for the respective $G_\mathrm{C}$ or $G_\mathrm{H}$. When $G_\mathrm{C}$ (resp. $G_\mathrm{H}$) is empty, one simply omits the crossing equations involving the first and second (resp. third) entries.

Consider $G_\mathrm{C} = G_\mathrm{H}$, for the $(G_\mathrm{C} \times G_\mathrm{H}) \rtimes \mathbb{Z}_2$ flavor symmetry, the crossing equations (5.12)

can be rewritten as

$$
\begin{aligned}
0 = &\begin{pmatrix} v\mathcal{A}^{s}_{\mathcal{B}[0]^{(0;0)}_0}(u,v) \\ (\mathcal{F}_{\mathcal{B}[0]^{(0;0)}_0})^{\mathbf{1}}_{*}(u,v,w) \end{pmatrix} - \sum_{\Delta,\ell}\lambda^2_{\mathrm{C,H,}(\mathcal{L},\Delta,\ell)}\begin{pmatrix} u\mathcal{A}^{t}_{\Delta,\ell,0,0}(v,u) \\ 0 \end{pmatrix} \\
&+ \sum_{\Delta,\ell}\lambda^2_{\mathrm{C,C,}\mathcal{L}[2\ell]^{(0;0)}_\Delta,\mathbf{1},+}\begin{pmatrix} v\mathcal{A}^{s}_{\mathcal{L}[2\ell]^{(0;0)}_\Delta}(u,v) \\ (\mathcal{F}_{\mathcal{L}[2\ell]^{(0;0)}_\Delta})^{\mathbf{1}}_{*}(u,v,w) \end{pmatrix} + \sum_{\Delta,\ell}\lambda^2_{\mathrm{C,C,}\mathcal{L}[2\ell]^{(0;0)}_\Delta,\mathbf{1},-}\begin{pmatrix} -v\mathcal{A}^{s}_{\mathcal{L}[2\ell]^{(0;0)}_\Delta}(u,v) \\ (\mathcal{F}_{\mathcal{L}[2\ell]^{(0;0)}_\Delta})^{\mathbf{1}}_{*}(u,v,w) \end{pmatrix} \\
&+ \sum_{\Delta,\ell,\mathbf{r}\neq\mathbf{1}}\lambda^2_{\mathrm{C,C,}\mathcal{L}[2\ell]^{(0;0)}_\Delta,\mathbf{r}}\begin{pmatrix} 0 \\ (\mathcal{F}_{\mathcal{L}[2\ell]^{(0;0)}_\Delta})^{\mathbf{r}}_{*}(u,v,w) \end{pmatrix} \\
&+ \sum_{\ell,\mathbf{r}}\lambda^2_{\mathrm{C,C,}\mathcal{A}[2\ell]^{(2;0)}_{\ell+2},\mathbf{r}}\begin{pmatrix} 0 \\ (\mathcal{F}_{\mathcal{A}[2\ell]^{2}_{\ell+2}})^{\mathbf{r}}_{*}(u,v,w) \end{pmatrix} + \sum_{\mathbf{r}}\lambda^2_{\mathrm{C,C,}\mathcal{B}[0]^{(4;0)}_2,\mathbf{r}}\begin{pmatrix} 0 \\ (\mathcal{F}_{\mathcal{B}[0]^{4}_2})^{\mathbf{r}}_{*}(u,v,w) \end{pmatrix} \\
&+ \lambda^2_{\mathrm{C,C,}\mathcal{B}[0]^{(2;0)}_1,\mathbf{adj}_\mathrm{C}}\begin{pmatrix} 0 \\ (\mathcal{F}_{\mathcal{B}[0]^{2}_1})^{\mathbf{adj}}_{*}(u,v,w) \end{pmatrix},
\end{aligned}
$$
(D.4)

where $*$ represents a column vector over representations in $\mathbf{adj}\otimes\mathbf{adj}$.

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
