# Peer review of "d N=4 Bootstrap and Mirror Symmetry"

_SciPost Physics, doi:SciPost Phys. 10, 097 (2021)_

## Round 1 · Referee Report · Anonymous (Referee 1) · 2021-3-11

Report

I am happy with the changes made. The manuscript reads better and can be accepted.

---

## Round 1 · Referee Report · Anonymous (Referee 2) · 2021-4-1

Strengths

1. The paper adds meaningfully to the very interesting and topical subject of understanding mirror symmetry in three dimensions. The analysis of the four point functions of the moment map operators using self-mirror symmetry constraints and minimality criteria are novel and definitely very interesting. These approaches provide interesting venues for further future investigations.

2. The paper integrates many available approaches to study N=4 3d gauge theories. These include sphere partition functions, conformal bootstrap, and exploitation of topological sectors of the theory.

3. The paper is very clearly written. Given a reader has some familiarity with some subsets of the techniques used, the paper will be very useful to learn in detail the facets of the story less familiar to the reader.

Weaknesses

The paper has no real weaknesses. I would say it is a bit too long but this is as the authors invest a lot of space into detailed review and analysis of the needed background material.

Report

In my opinion the paper definitely meets the criteria for publications in SciPost and I recommend it to be published.

Requested changes

A very minor suggestion: When the authors discuss the mirror of the $E_6$ MN theory above A.33 (footnote 31) I would recommend also mentioning for completeness the ${\cal N}=1$ 4d description of it in 4d derived in 1912.09348.

  • validity: top
  • significance: high
  • originality: high
  • clarity: high
  • formatting: excellent
  • grammar: excellent

Author:  Chi-Ming Chang  on 2021-04-03  [id 1342]

(in reply to Report 2 on 2021-04-01)

We thank the referee for the comments, and we have added the reference suggested by the referee in our new version.

---

## Round 1 · Author Response

We thank the referee for the comments., and we have made corrections and adjustments accordingly, please see our new version. Our responses are below.

1- Please see the footnote 30 on page 56 of the new version. 2- Please see the footnote 25 on page 43 and the footnote 28 on page 48 of the new version. 3- We added a brief summary of the main results of this paper in the introduction section of the new version.

---

## Round 1 · List of Changes

1- A summary of the main results has been added in the introduction section.
2- Footnote 25 on page 43, footnote 28 on page 48, and footnote 30 on page 56 have been added.
3- The notation for the flavor central charge is changed, for example in (2.7) on page 8.

---

## Round 2 · Author Response

We have added the reference suggested by the referee.

---

## Round 2 · List of Changes

Added reference 98 on page 57.

---

## Editorial Decision

published